# Importance of ocean dynamics in the onset and persistence of the 2013-15 and 2019-20 northeast Pacific marine heatwaves

Yu Long [1,2] ✉, Xinyu Guo[3], Neil J. Holbrook [4,5], Yunwei Yan[6], Zenghong Liu [1,2] ✉ & Xiao-Hua Zhu[1,2,7]

Large-scale marine heatwaves in the mid-latitude northeast Pacific have garnered significant attention due to their vast spatial extent, intensity, prolonged duration, and detrimental impacts on marine ecosystems and fisheries. Contrary to the conventional understanding that surface heat fluxes drive their formation, here we show that the two severe and impactful multi-season marine heatwave events in 2013-15 and 2019-20 were caused by the large-scale northward displacement of warm subtropical waters into the typically colder subarctic region. These oceanic changes are mainly explained by wind-driven circulation changes and Sverdrup balance adjustment. The marine heatwave decay phase corresponds with anomalous northwesterly winds which transport cold, dry air, enhancing latent heat loss and leading to ocean surface cooling. The physical driver is linked to the Tropical/Northern Hemisphere (TNH) teleconnection pattern, which aligns with previous studies. Collectively, the characteristic interannual timescale of the oceanic dynamic response and TNH explains the multi-season persistence of these extreme events.

Over the past decade, two large-scale marine heatwaves in the northeast Pacific—colloquially known as the 'Blob'—have attracted considerable scientific attention due to their unprecedented duration, spatial extent, and ecological impacts[1–10]. 'Blob 1.0' (2013–2015) emerged in the boreal winter (September–February)[1], while 'Blob 2.0' (2019–2020) intensified in the boreal summer (May–August)[6]. Although interannual climate variability has been proposed as the key driver[2,11], the reasons for their distinct onset month[6,12] and multi-year persistence[2,11,13] remain debated, highlighting gaps in our understanding of regional air-sea interactions.

Traditional mixed-layer heat budget analyses attribute the formation of these 'Blob' events primarily to reduced ocean heat loss to the atmosphere, with secondary contributions from oceanic advection and subsurface downward heat fluxes[1,6,14–19]. Subsequent numerical

and observational studies have explored the influences of atmospheric forcing on surface heat flux and upper-ocean mixing[6,18,19], highlighting mixed-layer depth variability[16] and subsurface freshening[20] as important mechanisms. However, we have identified that certain assumptions within these heat budget frameworks may obscure the important causal mechanisms.

Sea surface temperature anomalies (SSTA) in the northeast Pacific region exhibit significant interannual variability from 1980 to 2019 (Fig. 2c in ref. 2; Fig. 1b in ref. 6; Fig. 1a in ref. 12). This implies while marine heatwaves are typically defined as extreme high temperatures exceeding a threshold for at least five consecutive days[21], understanding these northeast Pacific marine heatwaves requires focusing on interannual SSTA rather than raw, unfiltered data (Supplementary Note 1). On the other hand, atmospheric changes occur at higher

[1]State Key Laboratory of Satellite Ocean Environment Dynamics, Second Institute of Oceanography, Ministry of Natural Resources, Hangzhou, China. [2]Southern Marine Science and Engineering Guangdong Laboratory (Zhuhai), Zhuhai, China. [3]Center for Marine Environmental Study, Ehime University, Matsuyama, Japan. [4]Institute for Marine and Antarctic Studies, University of Tasmania, Hobart, TAS, Australia. [5]Australian Research Council Centre of Excellence for the Weather of the 21st Century, University of Tasmania, Hobart, TAS, Australia. [6]College of Oceanography, Hohai University, Nanjing, China. [7]School of Oceanology, Shanghai Jiao Tong University, Shanghai, China. ✉e-mail: longyu@sio.org.cn; zliu@sio.org.cn

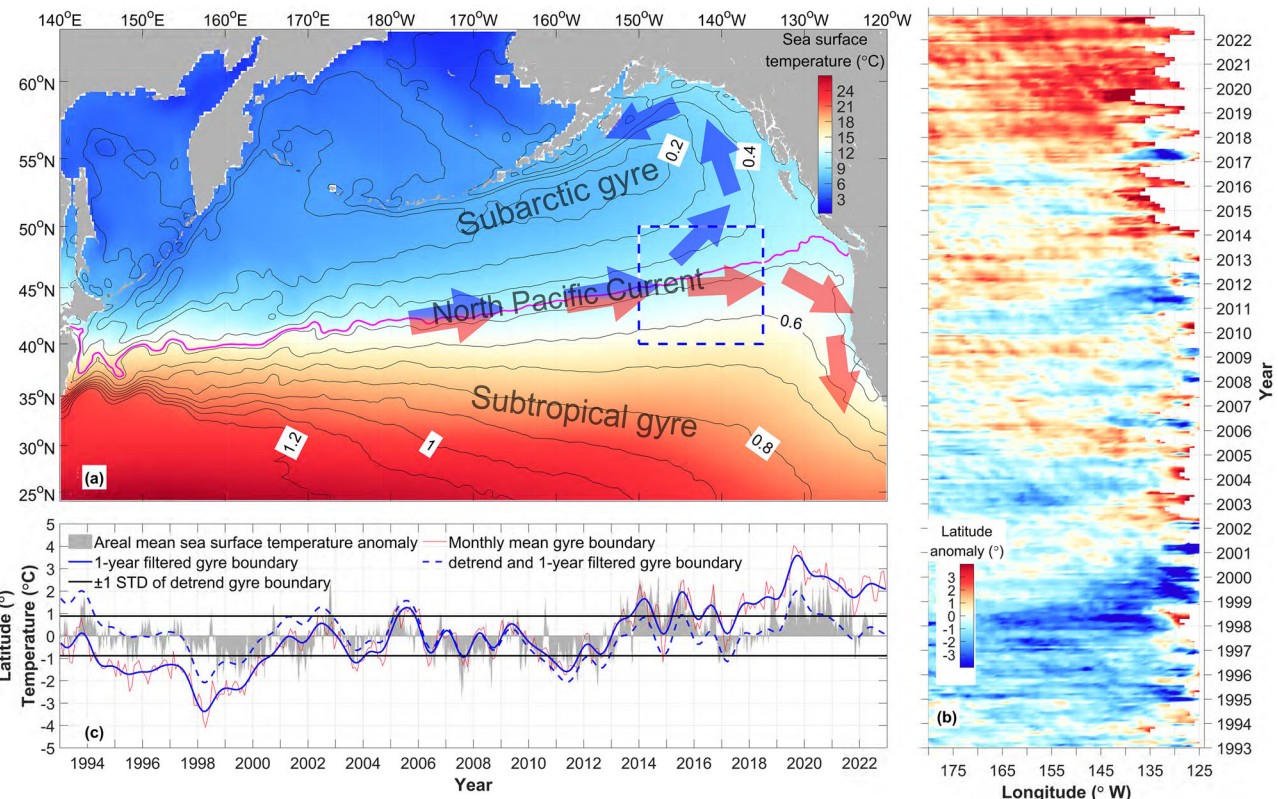

**Fig. 1 | Variation of the gyre boundary and sea surface temperature anomaly (SSTA). a** Annual mean sea surface temperature (color shading) and sea level (black lines). The magenta line represents the North Pacific Current flow axis along the +0.47 m sea level contour. **b** Hovmöller plot of the gyre boundary latitude anomaly (Represented by the location of the +0.47 m contour of monthly mean sea level) as a function of time and longitude. **c** Zonally averaged (150°W–130°W) latitude anomaly of the gyre boundary. Gray shading: the areal mean temperature anomaly (°C) within the box denoted by the blue rectangle in (**a**). For gyre boundary values, the red line is the monthly mean, the blue solid line is 1-year filtered, the blue dashed line is 1-year filtered and then detrended, and the black horizontal lines identify one standard deviation (STD) of the blue dashed line. A 5th-order Butterworth filter has been applied to the data in (**c**).

frequencies, meaning that temperature tendencies driven by surface heat flux dominate over slower oceanic processes. Therefore, analyzing unfiltered SSTA without accounting for the timescales most relevant to extreme events can potentially bias toward the atmospheric processes.

We note that classic mixed-layer heat budget equations may inadequately assess oceanic contributions. For example, advection terms often represent heat redistribution within a domain rather than bulk water parcel movement[22]. Additionally, combining lateral and vertical advection into entrainment can obscure key oceanic advection effects (see "Methods"). Oceanic processes are interdependent: geostrophy influences horizontal heat transport and modulates water column thickness, affecting stratification and vertical mixing. These nonlinear interactions complicate mechanistic interpretations of linearly decomposed heat budget terms. While the standard surface mixed-layer heat budget framework (temperature tendency = surface heat flux + temperature advection + residual) is widely used to understand local temperature change, the individual components of the residual term may contain critical insights when examining larger-scale dynamics.

Notably, the spatiotemporal scales of the 2013–15 and 2019–20 northeast Pacific marine heatwaves align with changes in the large-scale ocean dynamics. Although initial sea level anomalies were proposed as precursors[23], their underlying dynamics remain unclear. The North Pacific Current (NPC), a confluence of the North Pacific subarctic and subtropical gyres, traverses regions with SSTA exceeding +2 °C[24], marking the major temperature front in the northeast Pacific. We hypothesize that large-scale ocean circulation variations—particularly meridional excursions of the NPC—play a key role.

Our analysis demonstrates that large-scale interannual SSTA variations in the northeast Pacific—specifically, the dominant component of unfiltered SSTA—is primarily governed by large-scale ocean dynamics, a critical mechanism previously underestimated in the literature. Using a mixed-layer heat budget and a box model, we decompose oceanic contributions into lateral heat advection and a residual term (mainly vertical heat flux). The initiation of these two major northeast Pacific marine heatwave events, arises from anomalous wind stress curl driving northward subtropical warm water advection, whereas its decay is mechanistically linked to northwesterly wind anomalies that induce cold air advection and amplify latent heat loss. Collectively, these processes identify the Tropical/Northern Hemisphere (TNH) teleconnection pattern as the dominant driver of northeast Pacific marine heatwave dynamics[25].

## Result

### Gyre boundary variation in the northeast Pacific

The North Pacific subtropical gyre, characterized by warm and saline anticyclonic (clockwise) circulation, converges with the cool and fresh cyclonic (anticlockwise) subarctic gyre, resulting in a northeastward flow. The eastern segment of this flow is recognized as the North Pacific Current, whose axis marks both the gyre boundary and a basin scale temperature front (Fig. 1a).

The gyre boundary (see Methods) exhibited a southward displacement during the periods of 1994–2001 and 2010–2012, while a northward shifted from 2002–2007 and after 2013, coinciding with the occurrences of the 'Blob 1.0' and 'Blob 2.0' events (Fig. 1b). At the beginning of 2013, the gyre boundary was situated near its mean position (150 °W–130 °W, Fig. 1c, blue solid line), subsequently

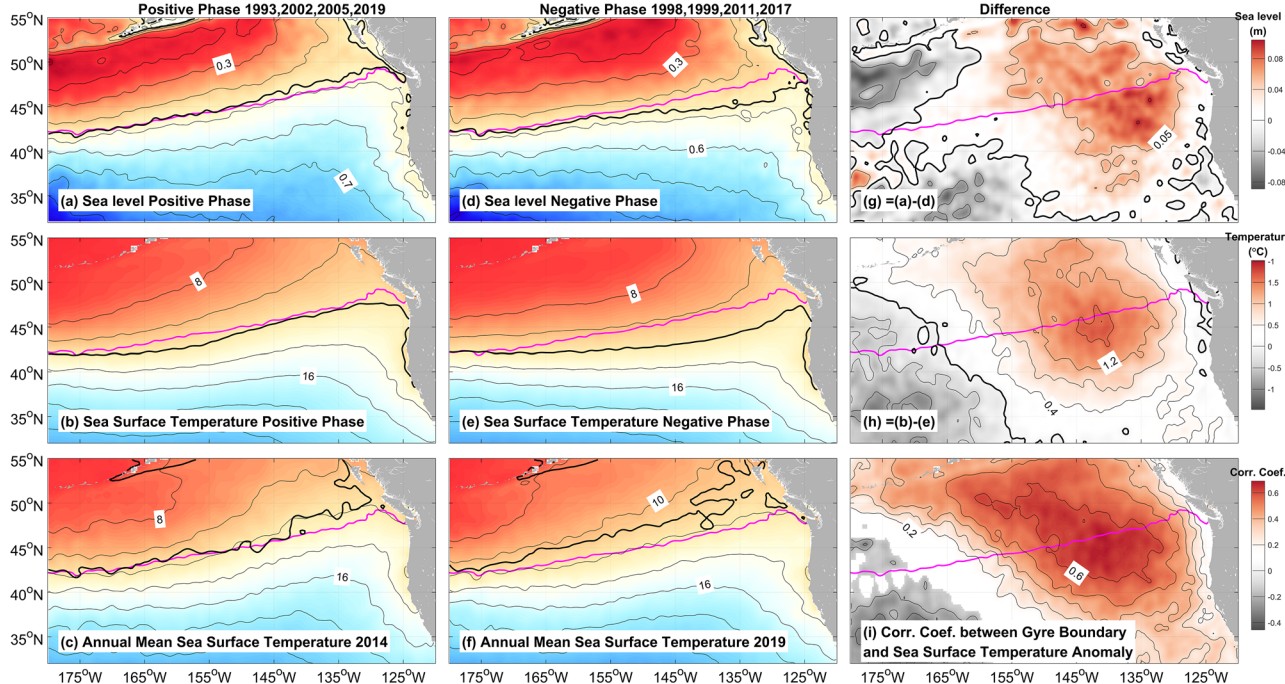

**Fig. 2 | Composite maps of sea level (SL) and sea surface temperature (SST).** SL (**a**, **d**) and SST (**b**, **e**) during the peak positive phase years of 1993, 2002, 2005, and 2019 (**a**, **b**), peak negative phase years of 1998, 1999, 2011, and 2017 (**d**, **e**), and the peak phase difference (**g** is for sea level anomaly (SLA), **h** is for sea surface temperature anomaly (SSTA)). Annual mean SST (color shading) and gyre boundary (black line) in 2014 (**c**) and 2019 (**f**). **i** Correlation coefficient between the gyre boundary (GB, blue dash line in Fig. 1c) and detrended SSTA, with regions where $p > 0.05$ are left blank. The thick black line in (**a**, **c**) is the +0.47 m contour of sea level and denotes the gyre boundary and flow axis of the North Pacific Current. The intervals for **g** and **h** are set at 0.05 m and 0.4 °C, respectively, while the magenta lines indicate the gyre boundary in the mean state.

displacing northward from 2014 to 2016. Following a brief southward displacement in 2017, the boundary reached a northern latitude of approximately 54°N during the summer of 2019, a position that was unprecedented within the satellite altimeter record from 1993 to 2022. This shift coincided with the documented intensification of 'Blob 2.0'.

The interannual displacement of the gyre boundary from its mean latitudinal position dominates its annual cycle (Fig. 1c, red line), with a poleward trend of +1.33° latitude per decade (Fig. 1c, blue solid line). The daily SSTA (Fig. 1c, gray shading) is consistent with interannual variations of the gyre boundary. Moreover, the zero-lag correlation coefficient was found to be +0.7 ($p < 0.05$) between the 1-year filtered, detrended monthly SSTA and the latitude displacement anomaly of the gyre boundary.

In order to conduct a more thorough examination of the potential impact of gyre boundary displacements on the SSTA in the northeast Pacific, which are linked to variations in the basin-scale ocean circulation, we created composite maps depicting sea level and SST for the years when the gyre boundary reached its northernmost positions (1993, 2002, 2005, and 2019; henceforth referred to as the 'positive phase') and its southernmost positions (1998, 1999, 2011, and 2017; henceforth referred to as the 'negative phase') (Fig. 2).

During the positive phase, the subtropical gyre located east of 175°W expands northward, leading to a southwest-northeast oriented tilt of the gyre boundary (Fig. 2a). In contrast, during the negative phase, the gyre boundary shifts southward and presents an almost zonal orientation (Fig. 2d). The composite maps of SST exhibit a pattern that closely resembles that of sea level (Fig. 2b, e). Given that the subtropical gyre is characterized by higher sea level and SST, while the subarctic gyre is associated with lower values, the northward displacement of the subtropical gyre into the region typically occupied by the subarctic gyre results in positive anomalies in both sea level and SST. Consequently, the phase difference of sea level (Fig. 2g) shows a similar spatial pattern to that of SST (Fig. 2h).

The orientations of the gyre boundary during the 'Blob 1.0' event (Fig. 2c, black line) and 'Blob 2.0' event (Fig. 2f, black line) exhibit characteristics like those observed during the positive phase (Fig. 2a). This correspondence also extends to spatial patterns of SST (Comparison of the color shading in Fig. 2c, f with that in Fig. 2b). In addition, the correlation coefficient (Fig. 2i) between the latitude displacement of the gyre boundary (Fig. 1c, blue dashed line) and the SSTA (also detrended) aligns with the spatial pattern of the SST difference between the positive and negative phases (Fig. 2h). Notably, a high correlation value (>0.6, $p < 0.05$, Fig. 2i, black thick line) is observed in regions traversed by the gyre boundary.

Collectively, these findings imply that the poleward shifts of the gyre boundary substantially influence SSTA in the midlatitude northeast Pacific through lateral advection. This interpretation challenges the existing paradigm, highlighting the need to analyze the mixed layer heat budget in greater detail to understand the reason for this inconsistency.

## Mixed-layer heat budget
As stated in the Introduction, it is imperative to select an appropriate timescale to undertake mixed layer heat budget diagnostic analyses (Fig. 3a). Our analysis shows the interannual component of the mixed layer temperature anomalies (MLTA) accounted for 80% of the variations observed in the unfiltered MLTA, while the ratio of their respective daily tendencies represented only 4% (see Methods). This discrepancy is attributed to the prolonged duration and consistent sign of the interannual component of the MLTA tendency. Specifically, if a 90-day low-pass filtered MLTA time series is selected for the analysis, the tendency associated with variations spanning from 90 to 365 days may obscure the tendency linked to interannual variations, potentially resulting in a misinterpretation of the prevailing physical processes (Comparison of the magnitude between Fig. 3b, c).

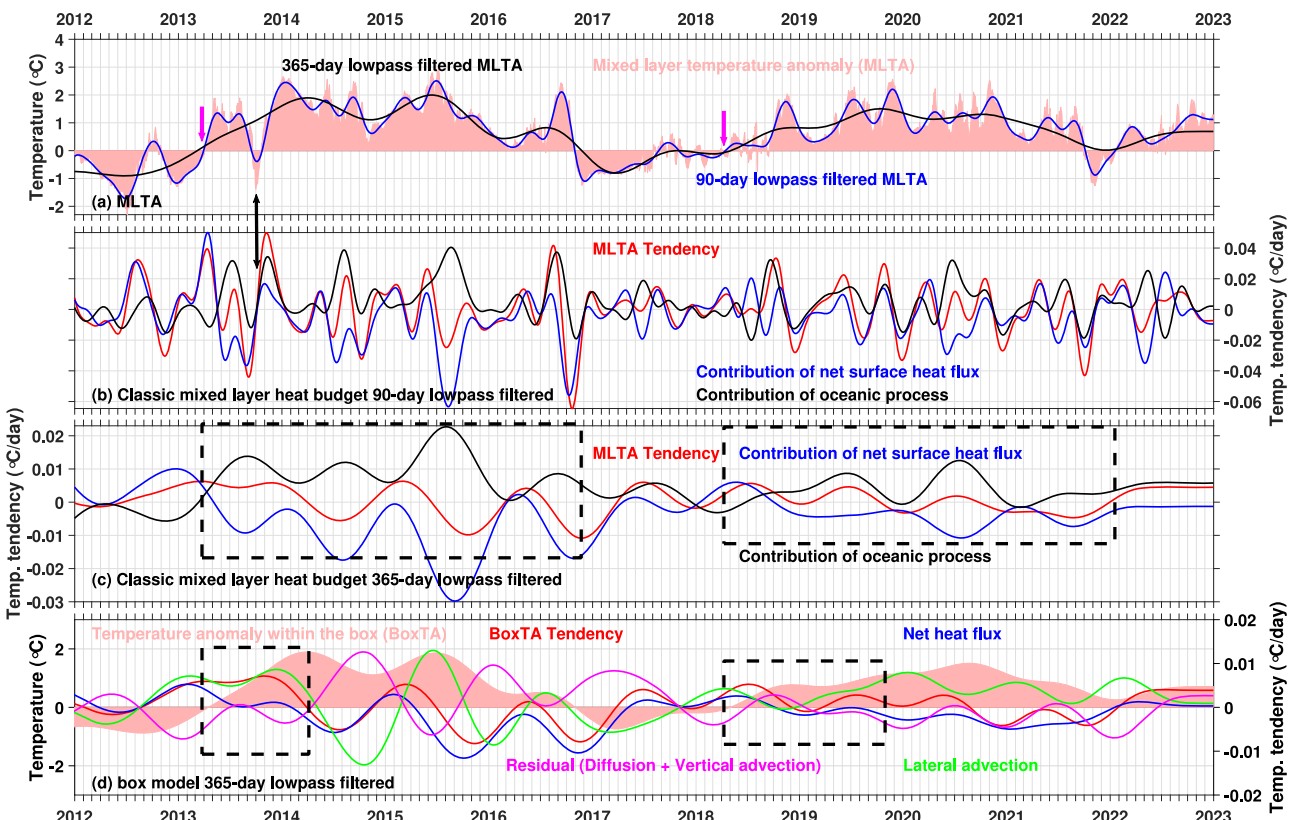

**Fig. 3 | The mixed-layer heat budget. a** Mixed-layer temperature anomalies (MLTA, °C, red shading) within the region 135–150°W, 40–50 °N, as denoted by the blue rectangle in Fig. 1a; also shown are the MLTA low-pass filtered by a 90-day cutoff (blue line) and MLTA low-pass filtered by a 365-day cutoff (black line). **b** Mixed layer temperature tendencies (°C/day) as a function of time, including net surface heat flux and integrated ocean process contributions from the tendency budget, derived from daily resolution data, with the annual cycle removed and 90-day lowpass filtered. **c** Same as (**b**) but 365-day lowpass filtered. **d** The heat budget for a box model, and 365-day lowpass filtered. The magenta arrows in (**a**) denote the neutral point. The black arrow denotes the cooling event in 2017. The black boxes in (**c**) denote the periods of the positive SSTAs associated with the 2013–15 and 2019–20 northeast Pacific marine heatwaves. The black boxes in (**d**) denote the period of the positive MLTA tendency that involved the development of the 2013–15 and 2019–20 northeast Pacific marine heatwaves.

Collectively, understanding the physical mechanisms that govern these large-scale northeast Pacific marine heatwaves require explaining their interannual SSTA variations. Our framework reveals significant discrepancies with previous studies. For example, there was a warm anomaly in the unfiltered MLTA (Fig. 3a, red shading) from 1 June 2013 to 15 September 2013, and a transient cool anomaly from 15 September 2013 to 23 October 2013, which again turned to a warm anomaly after November 2013. The cool anomaly lasted for about 40 days and was induced by the negative surface heat flux anomaly (Fig. 3b, blue line, black double-head arrow). The previous studies did not take this short cool anomaly into account and focused on the heat budget in November 2013. Since the interannual component of the MLTA is dominant, its neutral point (the time point when the cool anomaly shifts to the warm anomaly) should be regarded as a more reliable reference point for analyzing the underlying physical mechanisms.

This approach shifts the reference point of SSTA increase for 'Blob 1.0' from November 2013 to March 2013, and identifies May 2018 as the reference for 'Blob 2.0' (Fig. 3a, black line, magenta arrow), contrasting with the previous diagnosis of May 2019[6]. This adjustment affects the interpretation of the underlying physical drivers, particularly given the differing atmospheric conditions between March–May and November[2,19].

Our interannual-scale mixed layer heat budget analysis[6,26,27] partitions MLTA tendencies into atmospheric (surface heat flux) and oceanic (MLTA tendency minus net surface heat flux term) contributions (Fig. 3b, c). Our results show persistent positive oceanic contributions (Fig. 3c, black line in dashed rectangles) and negative

atmospheric contributions during the formation of these marine heatwaves, indicating that oceanic processes predominantly drive the onset while surface fluxes govern decay.

To address potential limitations in using mixed layer heat budget analyses in representing the oceanic dynamics (see Methods), we employed a box model (Fig. 3d) that decomposes the oceanic term into lateral advection and a residual (primarily vertical processes). The result demonstrates the dominant role of lateral advective heat flux in the formation of these two large-scale northeast Pacific marine heatwaves, with minimal surface flux influence (Fig. 3d, black dashed rectangle), supporting our advection hypothesis. These findings raise fundamental questions: What factors cause the interannual variations in ocean circulation and net heat flux?

## Wind stress curl anomalies drive gyre boundary poleward displacements

Variations in the northeast Pacific Ocean large-scale circulation are predominantly controlled by the wind stress curl[28]. To analyze this relationship, we generated composite wind stress curl maps by using ERA5 data during extreme northward or southward displacements of the gyre boundary, corresponding to its positive and negative phases (Fig. 4a, b). The zero wind stress curl contour closely coincides with the gyre boundary positions (magenta lines). Positive wind stress curl anomalies in the southwest and negative anomalies in the northeast of the region, delineated along a diagonal axis from 125°W-40°N to 160°W-55°N (Fig. 4c), correspond with northward displacements of the gyre. Conversely, the negative phase features a weaker and spatially reversed anomaly pattern (Fig. 4d). As expected, the annual mean wind

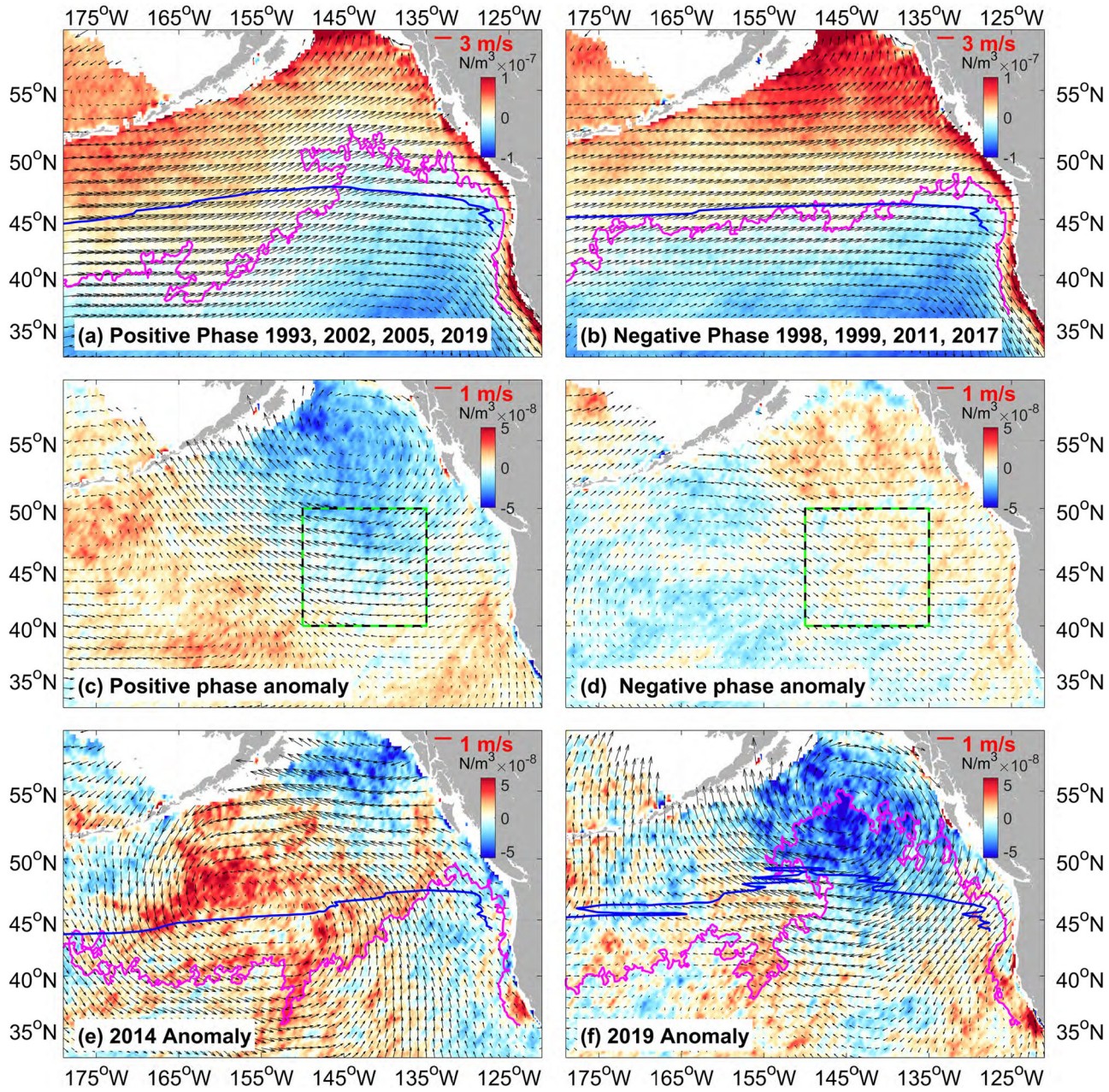

**Fig. 4 | Composite maps of the wind stress curl (N/m³, color shading) and wind vectors (black arrows).** During the (**a**) peak positive phase years of 1993, 2002, 2005, and 2019, **b** peak negative phase years of 1998, 1999, 2011, and 2017, **c** positive phase anomaly, **d** negative phase anomaly, **e** anomaly in 2014, and **f** 2019. The magenta lines denote the zero contour of the wind stress curl. The blue lines denote the zero contour of the meridional mass transport, as derived from the Sverdrup relation.

stress curl anomalies in 2014 (Fig. 4e) and 2019 (Fig. 4f) closely mirror the pattern associated with northward gyre boundary shifts, suggesting changes in the wind stress curl were a critical driver of these large-scale northeast Pacific marine heatwave events.

The North Pacific subtropical (subarctic) gyre responds to the large-scale negative (positive) wind stress curl through Sverdrup dynamics[24]: $\beta M_y = curl_Z(\boldsymbol{\tau})$, where $\beta$ represents the latitudinal gradient of Coriolis parameter, $curl_Z(\boldsymbol{\tau})$ the vertical component of wind stress curl, and $M_y$ the meridional mass transport. Negative wind stress curl anomalies enhance the subtropical anticyclonic circulation while weakening the cyclonic subarctic gyre. Although the Sverdrup relation represents the steady-state solution, our analysis does not concentrate on the transition of ocean circulation between its positive and negative phases (Fig. 2a, d), which pertains to time-varying solutions. Rather,

the Sverdrup relation here explains the ocean circulation during these two phases.

Positive-phase wind stress curl anomalies induce a northeast-tilted gyre boundary displacement east of 175°W, contrasting with zonal boundary orientations during negative phases. Sverdrup-derived mass transport streamfunction zero contours (blue lines) align closely with the observed gyre boundaries (magenta lines) in both phases (Fig. 4).

Sea level alone inadequately represents the geostrophic stream-function due to steric height contributions (sea level changes caused by thermal expansion and saline contraction). Composite analyses of the 2000 m-referenced geostrophic streamfunction and temperature reveal northward (39.1°N) and southward (36.4°N) surface displacements of the gyre boundary during the positive and negative phases

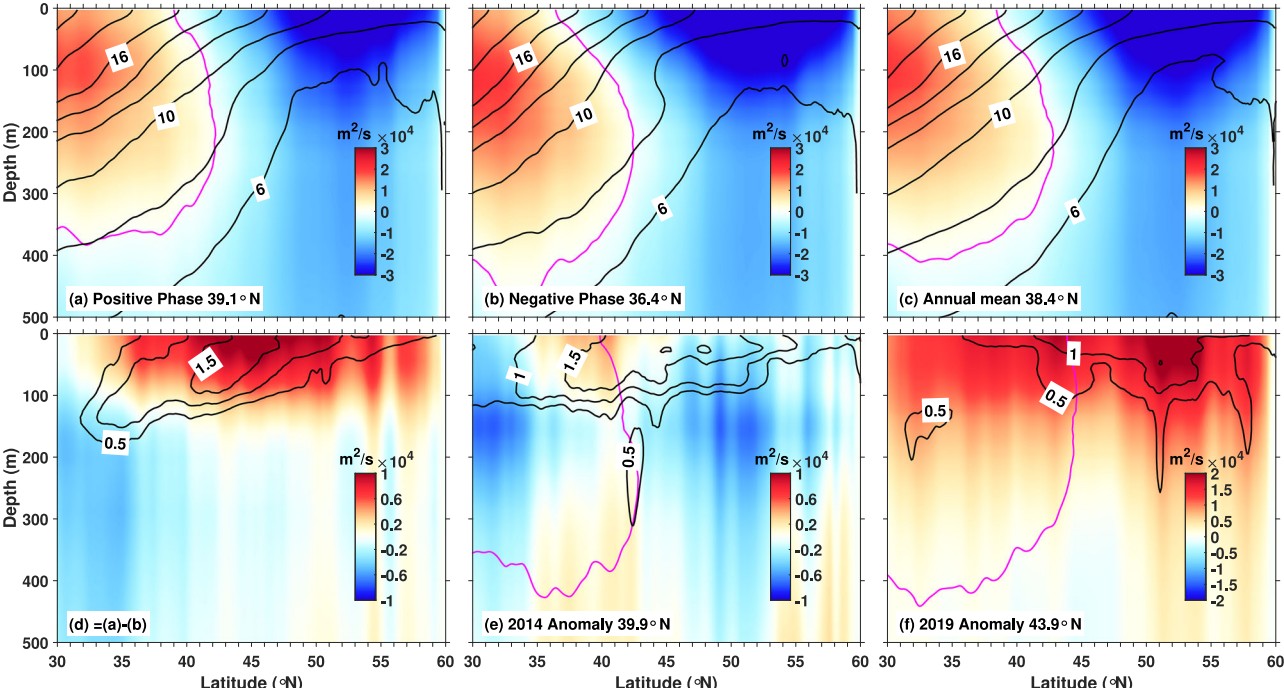

**Fig. 5 | Zonally averaged (145°W-135°W) geostrophic stream function (m²/s, color shading) and temperature (°C, black line).** Relative to a reference depth of 2000 m, for the **a** positive phase, **b** negative phase, and (**c**) annual mean. **d** is the phase difference between **a** and **b**. **e**, **f** are the anomaly for 2014 and 2019 from (**c**). The magenta line in (**a**, **b**, **c**, **e**, **f**) denotes the zero contour of the geostrophic stream function. Positive (negative) values of the geostrophic stream function in (**d**–**f**) represent anticyclonic (cyclonic) rotation (or change). The latitude in the legend is the position of the zero contour of the geostrophic stream function on the sea surface.

respectively, creating a 2.7° latitudinal displacement (Fig. 5a, b). Upper 200 m streamfunction anomalies indicate subtropical gyre expansion and subarctic gyre contraction during the positive phases (Fig. 5d). The exceptional 2019 northward shift to 43.9°N reflects intensified anticyclonic anomalies across 30-60°N (Fig. 5f).

The spatial expansion of the subtropical gyre results in two key consequences: the warm subtropical water moves poleward, and the local thermocline deepens, causing warming within the latitude band of 30–60 °C. Additionally, the deepening of the thermocline accounts for the presence of the warm anomaly beneath the mixed layer[16,29] and indicates that advection plays a more important role in the subsurface warming.

Phase-dependent wind vector anomalies (Fig. 4c, d), though not directly incorporated into turbulent heat flux calculations, influence airmass sources over the northeast Pacific region. Given the role of surface heat flux in the decay of these two events, these wind pattern variations motivate investigation of the atmospheric conditions that caused the heat flux anomalies.

**Anomalous northwesterly wind explains marine heatwave decay**

The net heat flux term in the mixed layer heat budget (Fig. 3c, blue line) exhibits similar variations with the surface net heat flux (Fig. 6a, black line) during the negative peaks of SSTA during the decay phases of the 2013–15 and 2019–20 northeast Pacific marine heatwaves (Fig. 6, vertical magenta lines). Building on prior comprehensive analysis[16] of mixed layer depth impacts on surface heat flux components, we focus on clarifying the role of the surface heat flux in the dynamics of these two marine heatwave events.

Latent heat flux dominates the net surface heat flux (Fig. 6a, blue line), with secondary contributions from sensible heat flux and long-wave radiation (Fig. 6a, red and green lines). The primary driver of the latent heat flux anomalies—calculated as wind speed multiplied by the specific humidity difference between saturation and 2 m air—was

decomposed into wind speed variations (Fig. 6b, blue line) and humidity difference variations (Fig. 6b, red line).

Although oceanic heat advection played a critical role in the evolution of these two marine heatwave events, weakened surface heat loss occurred in 2013 and around 2018 (Fig. 6a). Latent heat flux decomposition reveals that specific humidity differences predominantly governed variations in early 2013 and late 2017 (Fig. 6b), while wind speed reductions (Fig. 6e) gained comparable importance during late 2013 and early 2018 (Fig. 6b). Throughout the 2016–2017 and 2020–2021 northeast Pacific marine heatwave decay periods, specific humidity differences remained the principal factor.

Saturation-specific humidity (Fig. 6c, red shading) and 2 m air humidity (Fig. 6c, blue shading) display consistent variations, with their difference (Fig. 6c, black line) aligning closely with the humidity difference component in latent flux decomposition (Fig. 6b, red line). However, spatial patterns of wind vector anomalies (Fig. 4c, d) and directional shifts (Fig. 6d) suggest wind direction variations critically influenced both wind speed (Fig. 6e) and humidity differences.

The initiation of these two large-scale northeast Pacific marine heatwave events coincided with anomalous southeasterlies (Fig. 4c), characterized by positive anomaly of wind direction (Fig. 6e), driving warm air advection toward lower-SST regions. This induced airmass compression (reducing wind speeds, Fig. 6e), while warm, moist southern air traversing low SSTA region generated negative humidity difference anomalies (Fig. 6c), collectively suppressing oceanic heat loss. On the other hand, the decay of these two marine heatwaves featured a negative anomaly of the wind direction, marked by anomalous northwesterlies (Fig. 4d), with subsequent wind speed increases attributable to cool air advection toward higher-SST areas, causing airmass expansion. These cold winds enhanced the ocean surface latent heat flux in the northeast Pacific, accelerating SSTA reduction through intensified ocean heat loss.

## Schematic view and physical driver

During the onset phase of the 2013–15 and 2019–20 northeast Pacific marine heatwaves, wind stress curl anomalies drove a northward migration of the gyre boundary that served as the primary contributor to increased SSTA. In contrast, surface turbulent heat flux anomalies generated by wind direction anomalies emerged as the dominant driver of decreased SSTA during the subsequent decay phase. The distinct temporal responses of SSTA to heat advection versus diabatic cooling processes reveal a nonlinear relationship between the local upper ocean thermal response and the wind forcing (Fig. 7).

Specifically, these variations in the temperature tendency result from the interaction between oceanic advection and the surface heat flux. During the positive SSTA tendency (Trough→0→Ridge), southeasterly wind anomalies advect warm humid air into the northeast Pacific region. The condition of 'warm winds over low-SSTA' suppress ocean heat loss, while negative wind stress curl anomalies enable subtropical warm water intrusion, jointly elevating SSTA via advection and reduced latent heat flux. Once the SSTA exceeds the value of zero (0), the air-sea interactions transit to 'warm winds over high-SSTA', where advection is the dominant contributor to the amplification of the SSTA as latent flux stabilizes at the 0 anomaly (Fig. 3c). Conversely, northwesterly wind anomalies drove cold air southward in correspondence with the negative SSTA tendency (Ridge→0→Trough). Specifically, 'Cold winds over high-SSTA' enhanced turbulent heat loss, while positive wind stress curl anomalies caused warm water divergence out of the northeast Pacific region. Meanwhile, the eastward North Pacific Current continuously drove the inflow of warm water, causing the oceanic heat advection to remain positive. When the gyre boundary shifted past its mean position, 'cold winds over low-SSTA' restored the positive latent heat flux, as southward gyre displacements amplified the advective cooling.

Given the Sverdrup relation ($\beta M_y = curl_z(\tau)$) describes the linear response of the ocean circulation to the wind stress curl (Fig. 4a, b), we found that the wind stress curl over the 'Blob' region led gyre boundary latitude anomalies (Fig. 1c, blue dashed line) by 4 months at the maximum correlation coefficient (Fig. 8a). As theoretically anticipated, a negative correlation (peak $r < -0.6$, $p < 0.05$) manifested in the Alaskan gyre region (Fig. 8a, green box) north of the 'Blob' region, indicating that the northward displacement of the gyre boundary was predominantly governed by the subarctic gyre contraction. Importantly, we found positive correlations near 12°N (135°E-160°W) and negative correlations in the equatorial Pacific (0–3°N, 180–120°W), suggesting that wind fluctuations over the northeast Pacific region were most likely influenced by broader-scale atmospheric changes.

Within the Alaskan gyre region, the wind stress curl corresponds strongly and significantly with the sea level pressure anomalies (−0.81, $p < 0.05$, SLPAs, Fig. 8b). Lag-correlation analysis reveals that SLPA perturbations originate in the tropical eastern Pacific with 5-month lead before the Alaskan gyre intensified (Fig. 8c–h). The zero-lag correlation pattern exhibits positive SLPA values over the Gulf of Alaska and from the Gulf of Mexico northeastward across the western North Atlantic, contrasted by negative anomalies across eastern Canada. This spatial configuration bears strong resemblance to the positive phase of the Tropical/Northern Hemisphere teleconnection pattern (TNH, Fig. 8h)[25], which modulates the longitudinal extension of the Pacific jet stream and influences the intensity/position of the climatological Hudson Bay Low. Our findings align with previous conclusions[13] that persistent positive TNH phases facilitate prolonged 'Blob' marine heatwave events through sustained atmospheric forcing.

## Discussion

In this study, we investigated and further analyzed the causes of the persistent multi-year marine heatwaves in the mid-latitude northeast Pacific, also known as 'Blob 1.0' and 'Blob 2.0'. Our results demonstrate that the onset and emergence of these events stemmed from anomalous oceanic heat advection driven by wind stress curl changes through ocean dynamic responses. Conversely, the decrease of SSTA through their decay phase primarily resulted from enhanced latent heat losses to the atmosphere, associated with southward displacement of the gyre boundary and cold air advection via northwesterly wind anomalies. Collectively, the anomalous spatial pattern of the wind field constituted the dominant factor governing the formation and decay processes.

Through correlation analyses between the gyre boundary position anomalies and the wind stress curl, combined with lag-correlation analyses of the SLPA, we determined that interannual meridional displacements of the subtropical-subarctic gyre boundary and northeast Pacific ('Blob') region SSTA variations align with the Tropical Northern Hemisphere (TNH) teleconnection pattern. The key findings are organized as follows:

First, oceanic advection was found to be the principal driver of the formation of the 2013–15 and 2019–20 northeast Pacific marine heatwave events. Ocean circulation changes were manifested through poleward gyre boundary displacement, enabling subtropical warm water intrusion into the typical subarctic regions. Our analysis reveals that these extensive marine heatwaves were modulated by alternating southwest-northeast oriented wind stress curl anomalies across the northeast Pacific, consistent with interannual variations in the quasi-steady Sverdrup balance. This gyre boundary shift mechanism contrasts findings in previous studies outlining the dominance of the net surface heat flux in the development of these events. We demonstrate that standard mixed-layer heat budget advection terms do not adequately represent the lateral heat advection complexity, where entrainment components gain significance at interannual scales.

Based on our analyses, we revise the reference month for onset of 'Blob 1.0' to March 2013 (versus prior November 2013 designations) by defining initiation at the neutral point (cool-to-warm transition) of interannual SSTA. This approach eliminates high-frequency interference, particularly the 40-day cool anomaly in October 2013. Both 'Blob' events initiated during boreal spring (March-May), with this revised timeline focusing on physical mechanism rather than occurrence of temperature extremes.

Second, anomalous latent heat flux from cold air influx primarily drove the decay of the 2013-15 and 2019-20 northeast Pacific marine heatwave events. Mixed-layer heat budget analyses show the dominance of the net surface flux cooling, with latent heat flux loss paramount. Specific humidity differences exerted stronger influence on the latent fluxes than the wind speed. During the decay of the 2013-15 event, northwesterlies introduced dry cold air enhancing humidity differences, while the decline of the 2019-20 event involved sustained high SSTA and weak dry air advection. This represents a comprehensive analysis of local air-sea interactions during the decay of these two large-scale northeast Pacific marine heatwave events.

Third, we identify the TNH teleconnection pattern as the underlying physical driver. The gyre-boundary meridional displacement is characteristic of Sverdrup balance adjustment to changes in the wind stress curl, and accounts for the large-scale pattern of temperature change within the northeast Pacific region. The maximum correlation is in the Alaskan gyre region north of the 'Blob' region. This indicates subarctic gyre contraction links closely to northward boundary shifts, with SLPA correlation patterns further substantiating the role of the TNH as an important large-scale atmospheric driver.

We found that the persistence of these two large-scale northeast Pacific marine heatwaves can be explained by: (1) the large-scale ocean circulation response to the wind stress curl anomaly in the northeast Pacific which favors the interannual timescale, (2) the connection with the TNH teleconnection pattern on interannual and longer timescales[25], and (3) contributions from extratropical atmospheric teleconnections[19]. While these northeast Pacific marine heatwaves are caused by ocean-atmosphere processes acting locally, the modulating

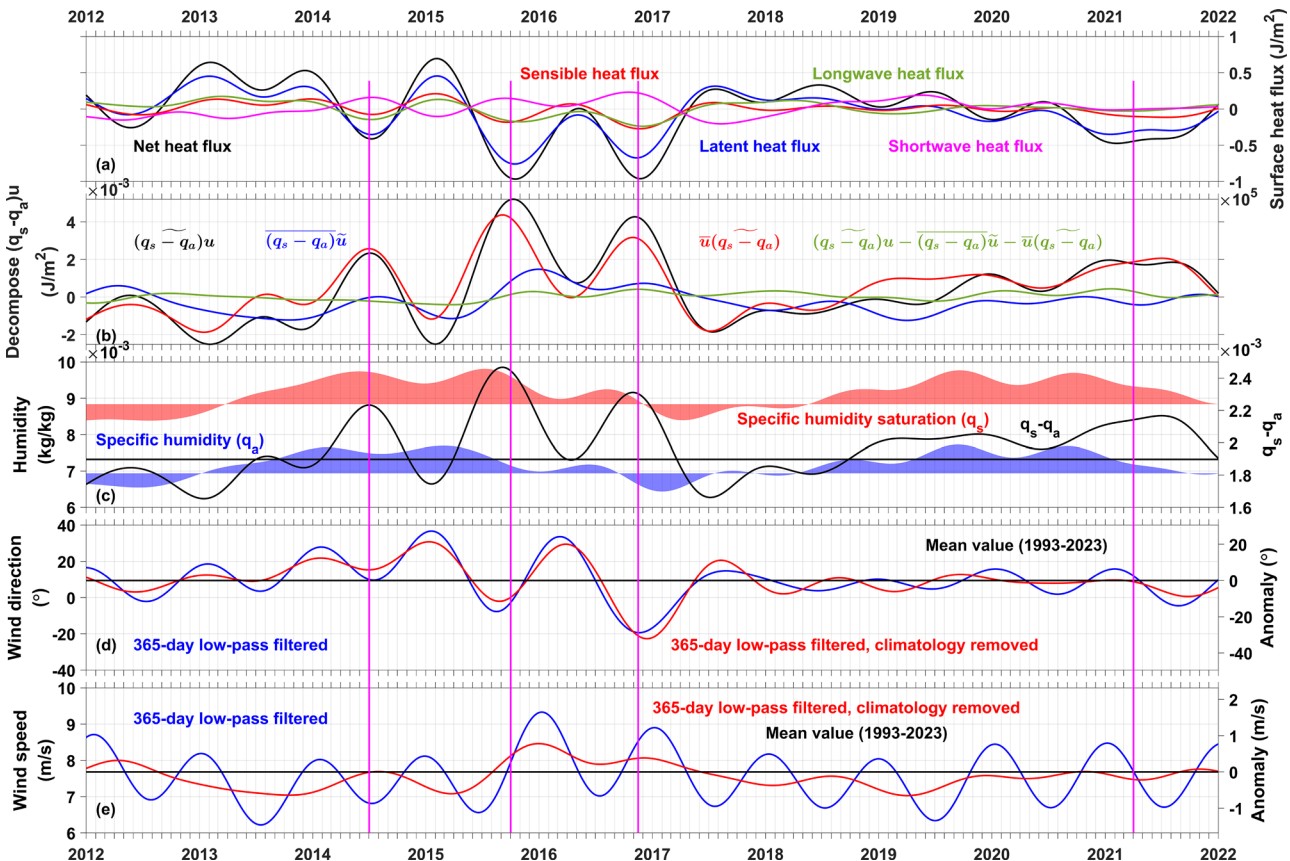

**Fig. 6 | Surface heat fluxes and latent heat flux decomposition in the northeast Pacific region.** As denoted by the blue rectangle in Fig. 1a, Anomalies of the (**a**) net heat flux (black line), latent heat flux (blue line), sensible heat flux (red line), longwave radiation flux (green line), and shortwave radiation flux (magenta line). **b** Decomposition of the latent heat flux (black line) into the contributions from the wind speed variations and annual mean humidity difference (blue line), and contributions from variations in the humidity difference and annual mean wind speed (red line). The residual (green line) term is small. **c** Specific humidity (blue shading), specific humidity at saturation (red shading) and their difference. **d**, **e** Direction and magnitude of the wind vector. The right y-axis of (**d**, **e**) is for the anomaly value, centered at the mean value denoted by the black line.

influence of the TNH via an atmospheric teleconnection was found to be important to the multi-month to multi-year persistence of these events, and which aligns with the much longer timescales in the ocean compared to those in and from the atmosphere[30].

Given the characteristic dynamics that underpin these large-scale northeast Pacific marine heatwave events, we call these 'gyre frontal marine heatwaves'. Although shifts in the location of fronts as a mechanism to increase ocean temperatures have been previously suggested[31], to our knowledge these 2013–15 and 2019–20 northeast Pacific events represent the first large-scale marine heatwaves diagnosed here of this type. First, we find that sea level anomaly (characterized by the surface geostrophic stream function) changes are important in understanding the SSTA. Second, shifts in the location of the gyre frontal boundary cause warm water to move into the northeast Pacific box (or 'Blob') region—an important dynamic mechanism found here that contrasts the local to regional diabatic heating from air-sea heat fluxes suggested in previous studies. This indicates that the broad-scale SSTA that evolved and characterized these northeast Pacific marine heatwaves are predominantly due to warm water advection into the region. In contrast, mixed layer heat budget analyses, that focus on fluxes into a defined box, do not capture the important contribution of the gyre-scale dynamics on these marine heatwave events. This calls for the importance of partitioning into the total ocean advection versus the heat flux.

The pronounced trend in the northward displacement of the gyre boundary over 1993–2023 (+1.33° latitude per decade, Fig. 1c) represents a major contributor to the rapid warming observed in the mid-

latitude northeast Pacific. This meridional displacement is attributable to the persistent intensification of SLPA-induced reduction in the positive wind stress curl anomalies (Fig. 8b)[32]. Through Sverdrup dynamics, these changes in atmospheric forcing drive a gradual contraction of the subarctic gyre constrained by the poleward migration of the subtropical-subarctic gyre boundary. The influence of this gyre boundary shift on SSTA is important. Assuming a characteristic SST gradient intensity of 2 °C/3° latitude to 2 °C/4° latitude in the northeast Pacific box ('Blob') region, the observed gyre boundary displacement could induce substantial warming of 0.67–0.89 °C per decade. This estimated warming rate considerably surpasses the actual observed warming trend in the 'Blob' region (0.17 °C per decade). The apparent discrepancy can be explained by the counteracting effect of the enhanced net heat flux from the ocean surface, which partially mitigates the warming tendency. On the other hand, the gyre boundary derived from the sea level data exhibits negative decadal anomalies in the early phase and positive anomalies in the most recent decade. This temporal variation suggests that the actual linear trend in the northward excursion of the gyre boundary may be influenced—potentially even overestimated—by its decadal variability.

The temperature front in the northeast Pacific between the cool, fresh subarctic gyre and the warm, saline subtropical gyre is important here. In the background of a warming climate, poleward shifts of the subtropical gyres result in warm water occupying regions where colder water previously existed[33]. This is analogous to these broad scale marine heatwave occurrences caused by episodic gyre boundary meridional shifts.

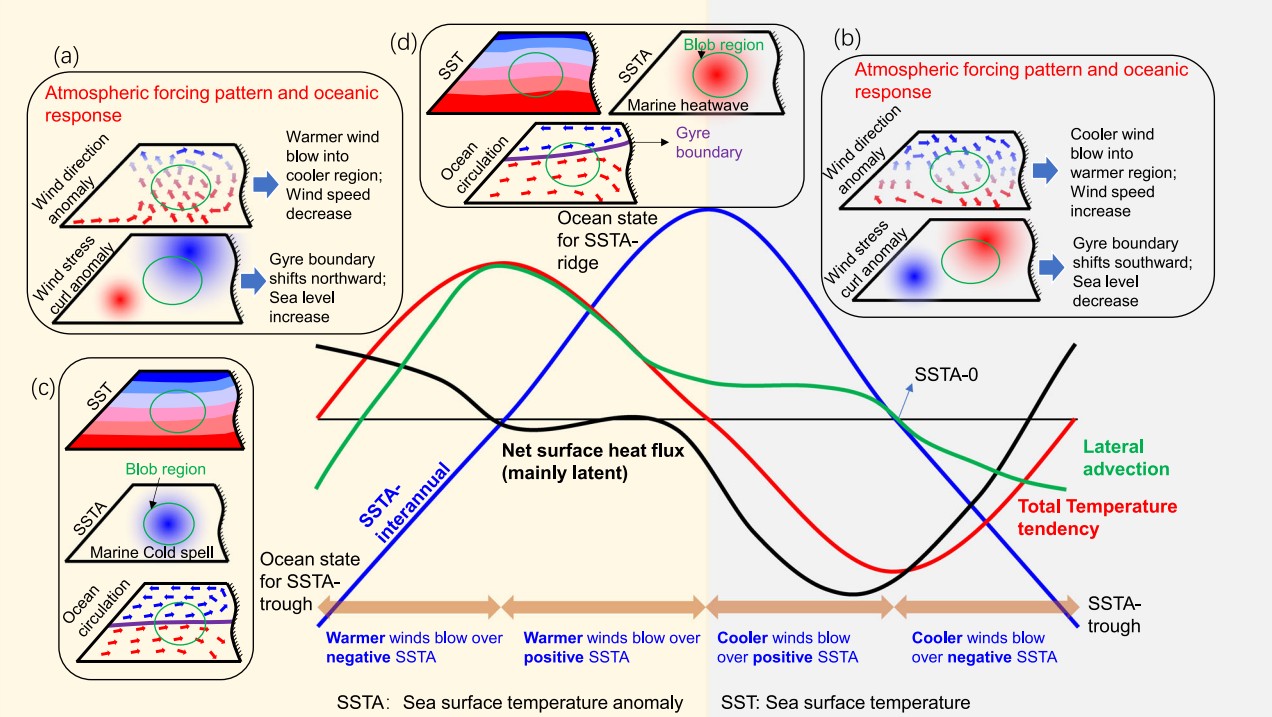

**Fig. 7 | Schematic of sea surface temperature anomaly (SSTA) and its tendency evolution in the northeast Pacific region.** Wind forcing pattern and oceanic response for the condition of SSTA increase (**a**) and decrease (**b**). Color shading of the wind vector indicates the air temperature. Sea surface temperature (SST), SSTA, and ocean circulation at the time point of the SSTA-trough (**c**) and peak (**d**). Red and blue shadings represent high and low temperatures. Blue and red arrows represent the subarctic gyre and the subtropical gyre, respectively. The brown double-headed arrows mark four typical air-sea interaction stages. Green circles represent the characteristic northeast Pacific 'Blob' region.

Moreover, it has been suggested that thermal displacements associated with marine heatwaves are important in understanding potential changes in marine species distributions[34]. In this case, isotherms within the latitude band of 30°N to 60°N were found to displace northward (Fig. 5d). The thermal displacements of isotherms increase with latitude from less than 100 kilometers at about 35°N to over 200 kilometers at about 52°N. Because the displacement of isotherms in the northeast Pacific region is mainly explained by contraction of the subarctic gyre, we speculate that the distribution of 'passive' marine species (i.e., non-swimmers or very slow swimmers) is also likely to be very much affected by lateral advection during these large-scale marine heatwave events.

## Methods

### Sea level and sea surface temperature data
The daily sea level data were obtained from the Copernicus Marine Service, with the Product ID of SEALEVEL_GLO_PHY_L4_MY_008_047. This 0.25° gridded product has global coverage. The data used here are from 1 January 1993 to 7 June 2023.

The U.S. National Oceanic and Atmospheric Administration Daily Optimum Interpolation Sea Surface Temperature (OISST V2.1) data[35] are used to show the sea surface temperature in Figs. 1 and 2. These SST data are optimally interpolated onto a 0.25° × 0.25° grid. The SSTA were obtained by removing the climatological daily mean values, calculated between January 1993 and December 2021, from the SST data.

### Wind and ocean reanalysis data
We also analyzed 10 m wind vectors above the sea surface based on data from the European Center for Medium Weather Forecasting (ECMWF) reanalysis, with the product name of 'ECMWF Reanalysis v5 hourly data on single levels from 1940 to present' (ERA5). The data used in the study are 6-hourly from 00:00, with a spatial resolution of 0.25° × 0.25°. The wind stresses are calculated following ref. 36.

The daily ocean reanalysis data used in this study are from the Bluelink ReANalysis 2020[37]. These data are 0.1° × 0.1° spatial resolution extending from 75°S-75°N. The data cover the period from January 1993–September 2023.

### Calculating the gyre boundary
The monthly sea level data within the region 180°W-116°W, 32°N-55°N are used to calculate the location of the gyre boundary. The longest +0.47 m contour is identified as the gyre boundary and North Pacific Current flow axis. For each month, the latitude of the gyre boundary was averaged from the original 0.25° resolution in longitude onto a 1° grid. Then, the climatological monthly mean value was removed to obtain the latitude anomaly of the gyre boundary from its mean position (color shading in Fig. 1b).

### Calculating the geostrophic stream function
The geostrophic stream function as a function of latitude and depth (the sectional view)—within the longitude band 145°W–135°W and latitude band 30°N–61°N where the west coast of Alaska is located—was calculated using daily seawater temperatures and salinities from the Bluelink reanalysis 2020 (BRAN2020) and validated by comparing with the result from the IAP gridded dataset (Supplementary Note 2).

For any day, the matrix of the temperature and salinity data is in the form of longitude × latitude × depth. According to the zonal resolution of the ocean reanalysis data, the temperature and salinity data were divided into meridional sections, each section in the form of latitude × depth. Then the zonal geostrophic velocity was calculated, based on a reference level of no motion of 2000 m, or the deepest grid level where water depth was shallower than 2000 m. Then the sectional geostrophic velocity was zonally averaged. The geostrophic stream function is the horizontal integral of the geostrophic velocity from the northernmost to the southernmost extent of the section.

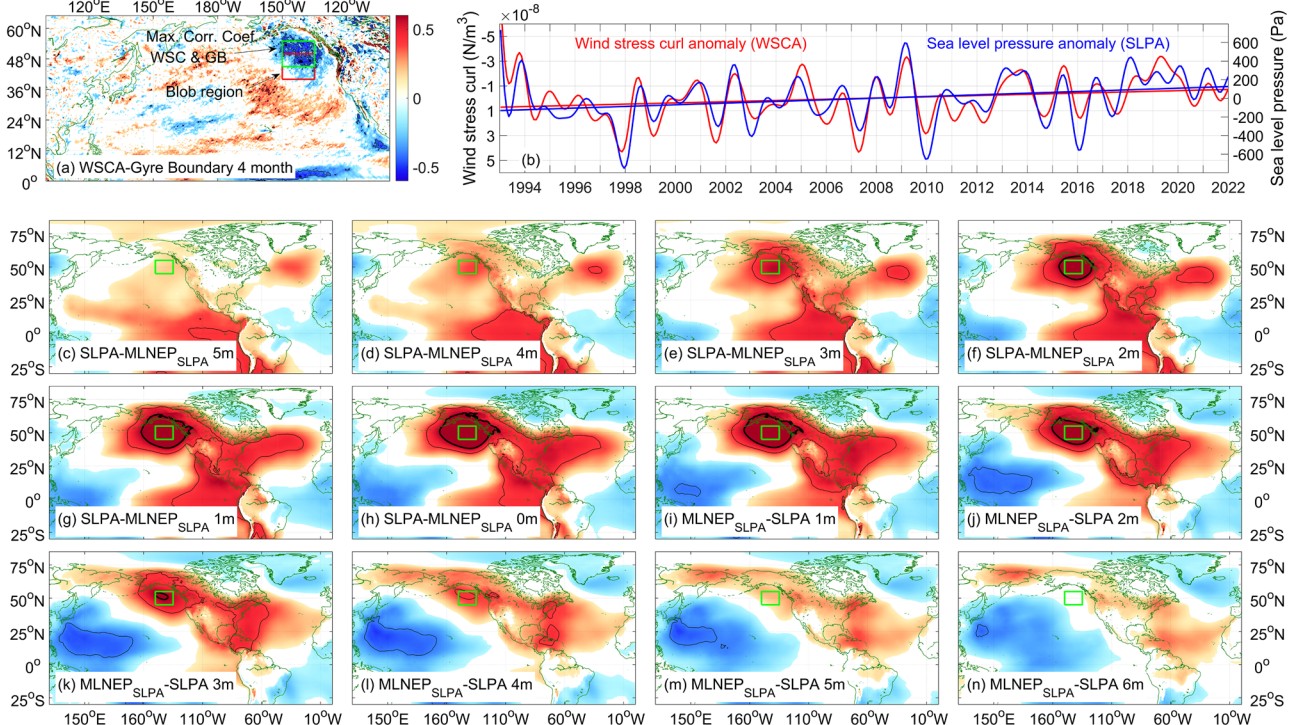

**Fig. 8 | Lag-correlation between sea level pressure anomaly (SLPA) at each grid point and areal mean SLPA in the Alaskan gyre region. a** Correlation coefficient pattern between the gyre boundary and wind stress curl (WSC led GB by 4-month), **b** areal mean wind stress curl anomaly (red line) and sea level pressure anomaly (blue line) over the region (green boxes, 45-55°N, 150-135°W) where the correlation coefficient between the gyre boundary latitude and wind stress curl anomaly reaches a maximum, and **c**–**n** lag correlation (in months) between the blue line in (**b**) and sea level pressure, color shadings represent the correlation coefficient, and the red box denotes the characteristic northeast Pacific 'Blob' region.

## Heat budget calculation

**The heat budget calculations.** The mixed layer heat budget equation applied here is in a simple form of:

$$\frac{\partial MLTA}{\partial t} = \frac{Q_{net} - Q_h}{\rho C_p h} + residual \qquad (1)$$

where $Q_{net}$ is the net heat flux (the sum of shortwave radiation, longwave radiation, latent and sensible heat flux), $Q_h$ is the penetrative shortwave radiation (based on the relationship of:

$$Q_h = Q_{sw}\left(0.58 e^{-\frac{h}{0.35}} + 0.42 e^{-\frac{h}{23}}\right) \qquad (2)$$

where $Q_{sw}$ is the shortwave radiation, $h$ is the time varying mixed layer depth, and $\rho$ and $C_p$ are the seawater density and specific heat capacity, respectively. Here, $\rho C_p = 4.088 \times 10^6 J/{}^\circ C^{-1} m^{-3}$, following ref. 16, and the residual term comprising the ocean dynamic contributions calculated as the difference between $\frac{\partial MLTA}{\partial t}$ and $\frac{Q_{net}}{\rho C_p h}$.

The temporally and spatially varying mixed layer depth, from the model output, is defined as the first depth where its buoyancy (per unit mass) increment from the surface exceeds $0.0003\ ms^{-2}$. The calculation is conducted at each data point of the ocean reanalysis data (BRAN2020), over the region of 150°W-135°W, 40°N – 50°N, and is then averaged by considering the area of each grid cell (areal mean value). The climatological daily mean value is removed by assuming the equation is balanced for the annual cycle.

The box model quantifies the total temperature tendency ($\frac{\partial BOXTA}{\partial t}$) through six interfaces of a predefined cubic control volume in the upper ocean. The model assumes the heat flux into the box is positive

and is expressed as:

$$\frac{\partial BOXTA}{\partial t} = \frac{Q_{net} - Q_H}{\rho C_p H} + \frac{\oint\left(\int_{-H}^0 \boldsymbol{u}_L T_L dz\right)dl}{\iiint_{-H}^0 dxdydz} + \frac{\iint wT_B dxdy}{\iiint_{-H}^0 dxdydz} + mixing \qquad (3)$$

where $H$ is the thickness of the box (in our case it is the total thickness of the upper 10 grids, with a value of about 61 m), $\boldsymbol{u}_L$ is the horizontal velocity normal to the boundaries of the computational domain, by assuming inflow is positive, $T_L$ is the seawater temperature at the boundaries. The accumulated horizontal advection through the four lateral interfaces (west, east, north, south) is calculated as:

$$\oint\left(\int_{-H}^0 \boldsymbol{u}_L T_L dz\right)dl = \iint u_w T_w dydz + \iint v_s T_s dydz$$
$$+ \iint -u_e T_e dydz + \iint -v_n T_n dydz \qquad (4)$$

with $u_w/v_s$ and $u_e/v_n$ representing zonal (east−west) and meridional (north−south) velocities at the corresponding interfaces, $T_w, T_e, T_s, T_n$ as temperatures. The vertical advection term is derived from $wT_B$ (vertical velocity × temperature at the bottom interface). Each term in the box model is represented as a net heat flux (velocity × temperature, e.g., $\left(\frac{\oint\left(\int_{-H}^0 \boldsymbol{u}_L T_L dz\right)dl}{\iiint_{-H}^0 dxdydz}\right.$ and $\left.\frac{\iint wT_B dxdy}{\iiint_{-H}^0 dxdydz}\right)$) at the boundaries, rather than an effective heat flux (e.g., velocity × temperature gradient, e.g., $-\bar{\boldsymbol{u}} \cdot \nabla \bar{T}$ and $-\left(\frac{\bar{T}-T_{-H}}{H}\right)\left(\frac{\partial H}{\partial t} + \boldsymbol{u}_{-H} \cdot \nabla H + w_{-H}\right)$).

The high correlation coefficients for the total temperature tendencies (+0.93, $p < 0.05$, between the red line in Fig. 3d and red shading in Fig. 3c), net heat flux terms (+0.84, $p < 0.05$, between the blue lines in Fig. 3d and Fig. 3c), and the respective residual terms that

represent the oceanic contribution (+0.75, $p < 0.05$) suggest that the decomposition of the box model reflects the same physical processes as that of the classic mixed layer heat budget.

**Examining the entrainment term.** We analyzed the complete form of the classic mixed-layer heat budget equation to show why this equation cannot fully capture the advection process.

The complete equation of the classic mixed layer heat budget is:

$$\frac{\partial MLTA}{\partial t} = \frac{Q_{net} - Q_h}{\rho C_p h} - \bar{\boldsymbol{u}} \cdot \nabla \bar{T} + Mixing - \left(\frac{\bar{T} - T_{-h}}{h}\right)\left(\frac{\partial h}{\partial t} + \boldsymbol{u}_{-h} \cdot \nabla h + w_{-h}\right) \quad (5)$$

The second term on the right-hand side represents advection, while the fourth term represents entrainment. If we take the entrainment term apart, we can get,

$$-\left(\frac{\bar{T} - T_{-h}}{h}\right)\left(\frac{\partial h}{\partial t} + \boldsymbol{u}_{-h} \cdot \nabla h + w_{-h}\right) = -\frac{\bar{T}}{h} w_{-h} \\ + \frac{T_{-h}}{h} w_{-h} - \left(\frac{\bar{T} - T_{-h}}{h}\right)\left(\frac{\partial h}{\partial t} + \boldsymbol{u}_{-h} \cdot \nabla h\right) \quad (6)$$

According to the continuity equation, the vertical velocity, $w_{-h}$ can be written as $h(\nabla \cdot \boldsymbol{u})$, which suggests the $-\frac{\bar{T}}{h} w_{-h}$ term can be written as $-\bar{T}\nabla \cdot \boldsymbol{u}$. This term represents the temperature tendency induced by the divergence or convergence of the horizontal velocity.

Moreover, the entrainment term in the classic mixed layer heat budget is in the form of the effective flux, whereby the temperature difference ($\bar{T} - T_{-h}$) multiplies vertical velocity (the term of $\frac{\partial h}{\partial t} + \boldsymbol{u}_{-h} \cdot \nabla h + w_{-h}$ has a unit of m/s). Given that the temporal variation of the mixed layer depth ($\frac{\partial h}{\partial t}$) may also be affected by the vertical motion, the entrainment term likely contains greater contribution from the lateral advection. Therefore, the advection term in this equation may underestimate the contribution of the advection process.

Another simple way to understand this issue is by applying the classic mixed layer heat budget equation to a cubic control volume. Consider a cubic control volume with fixed dimensions (width, length, and thickness), similar to a basic grid cell in a numerical model. Since the volume is time-invariant, the $\frac{\partial h}{\partial t}$ and $\nabla h$ terms vanish. This leads to:

$$\frac{\partial MLTA}{\partial t} - \frac{Q_{net} - Q_h}{\rho C_p h} = -\bar{\boldsymbol{u}} \cdot \nabla \bar{T} - \left(\frac{\bar{T} - T_{-h}}{h}\right) w_{-h} + Mixing \quad (7)$$

Expanding the second term in the right-hand side apart and rearranging, we have:

$$\frac{\partial MLTA}{\partial t} - \frac{Q_{net} - Q_h}{\rho C_p h} = -\bar{\boldsymbol{u}} \cdot \nabla \bar{T} - \frac{\bar{T}}{h} w_{-h} + \frac{T_{-h}}{h} w_{-h} + Mixing \quad (8)$$

where $-\bar{\boldsymbol{u}} \cdot \nabla \bar{T} - \frac{\bar{T}}{h} w_{-h}$ is the lateral advection and $\frac{T_{-h}}{h} w_{-h}$ represents the vertical heat advection.

A simple scale analysis could help to evaluate the relative importance of each term. In the 'Blob' region, the average horizontal velocity at the sea surface is 0.05 m/s, the maximum horizontal temperature gradient is $9*10^{-6}$ °C/m (1 °C/110 km* 0.001 km/m), the vertical velocity in the upper ocean is $10^{-4}$ m/s, the annual mean mixed layer depth is 50 m, SST = 12 °C, $T_{-H} = 12 - 0.3°C = 11.7°C$ (the mixed layer depth is commonly defined as the depth at which temperature first becomes 0.3 °C different from the surface), and suppose $\bar{T} = \frac{SST + T_{-H}}{2} = 11.85°C$. On the right-hand side of the above equation, the first term ($\bar{\boldsymbol{u}} \cdot \nabla \bar{T}$) is on the order of $4.5*10^{-7}$ °C/s, the second term ($\frac{\bar{T}}{H} w_{-H}$) is on the order of $2.37*10^{-5}$ °C/s, and the third term ($\frac{T_{-H}}{H} w_{-H}$) is on the order of $2.34*10^{-5}$ °C/s.

Therefore, both the lateral advection ($\frac{\bar{T}}{H} w_{-H}$) and the vertical advection at the bottom of the mixed layer ($\frac{T_{-H}}{H} w_{-H}$) play important roles in the heat budget equation. However, if we write them in the form of the effective flux ($-\frac{\bar{T}}{H} w_{-H} + \frac{T_{-H}}{H} w_{-H}$, temperature difference multiplying vertical velocity), the two terms cancel out and generate a small term that is on the order of $-3*10^{-7}$ °C/s ($-2.37*10^{-5}$ °C/ s + $2.34*10^{-5}$ °C/s), which is comparable to the term of $\bar{\boldsymbol{u}} \cdot \nabla \bar{T}$. This analysis shows that if the predominance of a particular term is assessed based on its magnitude, both the vertical and lateral processes may be undervalued when the entrainment term is represented as the effective flux.

**Decomposition of the latent heat flux.** The latent heat flux can be represented as:

$$Q_{LH} = \rho L_e C_e (q_s - q_a) W \quad (9)$$

where $L_e$ is the latent heat of vaporization, $C_e$ is the stability- and height-dependent turbulent exchange coefficients for latent heat. $W$ is the wind speed, $q_a$ is the specific humidity at a reference height of 2 m above the sea surface, and $q_s$ is the saturation humidity at sea surface temperature.

Suppose $(q_s - q_a)W$ is the major contributor to variations in $Q_{LH}$, we decompose the variation of $(q_s - q_a)W$ according to:

$$(\widetilde{q_s - q_a})W = \overline{(q_s - q_a)}\widetilde{W} + \bar{W}(\widetilde{q_s - q_a}) + residual \quad (10)$$

Where tilde denotes the anomaly while overline denotes the annual mean. The climatological mean was subtracted from each term obtained from the above equation. Subsequently, a 365-day low-pass filter was implemented.

**Interannual component contribution of the MLTA to the total MLTA.** The interannual component contribution of the MLTA to the total MLTA is defined as the standard deviation of the interannual component of the MLTA (black line in Fig. 3a) divided by the standard deviation of the total MLTA (red shading in Fig. 3a). The interannual component contribution of the MLTA tendency to the total MLTA tendency is the standard deviation of the interannual component MLTA tendency (red shading in Fig. 3c) divided by the standard deviation of the total MLTA tendency (tendency of red shading in Fig. 3a).

## Data availability
We have used publicly available data only; new data were not generated by this study. ADT: https://data.marine.copernicus.eu/product/SEALEVEL_GLO_PHY_CLIMATE_L4_MY_008_057/description. SST:https://www.ncei.noaa.gov/products/optimum-interpolation-sst. ECMWF ERA5 data:https://cds.climate.copernicus.eu/cdsapp#!/dataset/reanalysis-era5-single-levels?tab=overview. BRAN2020 Ocean reanalysis: https://research.csiro.au/bluelink/bran2020-data-released/

## Code availability
The code used to analyze these data and generate the results presented in the study can be obtained from Supplementary Dataset 1–3.

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

## Acknowledgements

The authors acknowledge the very constructive and valuable comments from three reviewers—Nathan Mantua and two anonymous reviewers. Y.L. acknowledges funding from the Southern Marine Science and Engineering Guang dong Laboratory, Zhuhai (No. SML2021SP102), the National Natural Science Foundation of China (42106035, 41920104006), the Project of State Key Laboratory of Satellite Ocean Environment Dynamics (SOEDZZ2530), Scientific Research Fund of Second Institute of Oceanography, MNR (grants JG2306), and the China Postdoctoral Science Foundation (grant 2022T150778). X.G. is supported by Grant-in-Aid for Scientific Research (MEXT KAKENHI Grant number 22H05206). N.J.H. acknowledges funding from the Australian Research Council Centre of Excellence for the Weather of the 21st Century (CE230100012).

## Author contributions

Y.L. designed the study, performed the data analysis, prepared all figures and drafted the manuscript. X.Y.G. reviewed and edited the manuscript. N.J.H. contributed to the overall thinking and helped critically revise the manuscript. Y.W.Y., Z.H.L., and X.H.Z. discussed the results and commented on the manuscript. Z.H.L. funded this study.

## Competing interests

The authors declare no competing interests.
