## [Transparent Peer Review file · Nature Communications]

Importance of ocean dynamics in the onset and persistence of the 2013-15 and 2019-20 northeast Pacific marine heatwaves

Corresponding Author: Dr Yu Long

Version 0:

Reviewer comments:

Reviewer #1

(Remarks to the Author)

This manuscript reports on an investigation of a novel hypothesis that a dynamic ocean response to wind stress curl forcing played a fundamental role in two recent multi-year NE Pacific marine heatwaves. The authors conclude that interannual variations in gyre scale frontal dynamics comprehensively explain the spatial characteristics and multi-year persistence of the two events. The article is interesting for offering up this idea and some provocative interpretations of the results of their analyses. However, I do not believe that the work detailed here supports such grand conclusions. As noted in my specific comments below, simply showing that these events featured meridional excursions of the subarctic frontal boundary falls short of demonstrating cause and effect. Moreover, there is really no analysis or theory offered for linking the proposed mechanisms to the observed time-scales involved in the multi-year marine heatwaves of interest. If the event duration times were determined by the atmospheric forcing, for instance, one would need to explain the observed evolution of the atmospheric forcing field (something that DiLorenzo and Mantua focused on for the 2014-15 period, and Amaya et al. focused on for summer 2019). This work has focused more on a time-varying but spatially invariant interpretation of the atmospheric forcing - for example, creating composite maps (Figs 2 and 3) for sea level, SST, and the surface wind stress fields during what are termed "positive" and "negative" phase years in their detrended and 1-yr filtered gyre boundary index (which has its maximum value in 1993, a year that is not used in the composites).

This work has raised some interesting questions for me that I think are worth addressing. These include:

- The monthly mean gyre boundary index shows a substantial trend over the 1993-2022 period - is there a corresponding trend in the surface wind stress field (for instance, in the zero-curl contour latitude in the NE Pacific)?
- How would the gyre strengths and positions vary in response to the kind of atmospheric forcing variations described by Di Lorenzo and Mantua (2016)? Would they include the expansion of the subtropical gyre in 2013 through winter 2014, or a slowing of both the subarctic and subtropical gyres in response to the atmospheric forcing in the "North Pacific Oscillation" pattern that DiLorenzo and collaborators have linked with the North Pacific Gyre Oscillation?
- Would the fall 2014/winter of 2015 feature an expansion of the subarctic gyre but contraction of the subtropical gyre? These SSH variations and their links to the gyre circulations and atmospheric forcing are basically those described by Cummins and Lagerlof (2004), your reference 20.
- Could you better evaluate the relative contributions of basin-scale ocean dynamics versus surface flux forcing through simulations that involved forced ocean responses in a slab-ocean (mixed layer only) versus a dynamically-responding ocean model?

In summary, I believe there is merit in the research directions this work reports on. However, I do not believe that the research reported here supports the conclusions that are offered. I recommend this manuscript be returned to the authors with a request for major revisions.

Nate Mantua

Specific comments:

40: I would add “and fisheries” after “marine ecosystems”

43-44: revise to “meridional expansion (contraction” of the subtropical (subarctic) gyre” ... current text doesn't include subarctic

45-47: this analysis does not isolate the contribution to gyre scale frontal dynamics from other processes, nor does it connect those dynamics to the time scale of the events, so it falls short of “comprehensively explain(ing) the spatial characteristics and multi-year persistence of these ... heatwaves”

60: reasons for multi-year persistence have been investigated and explained by others (e.g., DiLorenzo and Mantua 2016)

78-80: Isn't a dominant tendency term required to understand the origins of a broad-scale warming of the upper ocean that happens within a single season, hence the focus on atmosphere-ocean interactions?

83-86: This may be true in an ideal case, but this real-world case study does not involve a surface mixed layer having top and bottom interfaces that are adiabatic.

120-125: Causation may go either way, or there may be a common forcing for the SSTa and latitude of the gyre boundary. I also think that the trend over the 1994-2023 period of record is striking. Is there is a similar trend in atmospheric forcing? Why doesn't the daily SSTA time series share the longer-term trend with the gyre boundary index?

131-134: Why do you conclude that the subarctic gyre west of 150W expands southward? I can't see the southward expansion of the subarctic gyre west of 150 from the SLH or SST difference maps (panels e and f) – do you mean the negative SSH differences between 40-50N and 180-170W?. I don't see this in Fig 1b either (I would expect to see a dipole centered on ~150W, but that is only a feature of a few years like 2014-2016). The main changes in frontal position are east of 140W from 2014-2016, and east of 165W from 2018-2020 (based on Fig 1b).

Figures 2 and 3: I think it would be best to use the same projection in the maps shown in Figures 2 and 3.

168-170: Interesting that the zero curl contour at 180 is about 5 deg latitude farther south in your positive phase than negative phase composite, while it is at about the same latitude for both phases on the west coast of N. America ~125W; This seems at odds with the SSH composites in Fig 2

176-179: Fig 3a shows your “positive phase” having the gyre boundary east of 150W shifting northward, not southward, in response to a negative wind stress curl anomaly in the Gulf of Alaska (north of 50N).

180-181: This is confusing. Perhaps you could include the actual wind stress curl anomalies for the composites of positive and negative phases and this would be easier for readers to see and understand than trying to assess anomalies from the composites shown here.

Figure 3 a-b-c and Fig 5 a-b: Wind vectors are really small and difficult to see, even at 200% magnification. Please make them bigger.

211-212: Thermocline deepening is inconsistent with previously published findings (like Amaya et al 2020 that show a shoaling mixed layer depth in summer 2019)?

241-249: I don't find this new definition of event onset to be very useful or convincing. Is there something special about climate dynamics in the month of May that makes you think this event onset criteria leads to a new and better understanding for NEP marine heatwave events? Would you say that Blob 3.0 started in January 2022 by your criteria? Did the 2013-14 event end in 2015 or 2016 by your criteria?

276-277: Did the marine heatwaves “result from” or “include” poleward expansion (contraction) of the subtropical (subarctic) gyre?

281-282: This analysis does not address the issue of “persistence”, but instead focuses on “onset”. What sets the time scale for these events? Is it the persistence and pattern of atmospheric forcing (along the lines of Di Lorenzo and Mantua's analysis of the 2013-2015 period)? Or something else?

295-297: If the basin-scale wind-stress curl is the key driver for the gyre variations, then multi-year persistence of these marine heatwaves is due to persistence in the wind-stress curl forcing. So this then begs the question – what provides persistence in the basin-scale atmospheric forcing of the gyre circulation? As noted, the ocean circulation (and mixed-layer heat content) response integrates atmospheric forcing so it has greater variance at longer time scales than the atmospheric forcing, yet the evolution of the atmospheric forcing has already been shown to play a key role in understanding the evolution and persistence of the 2014-2016 (DiLorenzo and Mantua) and 2019-2020 (Amaya et al) events.

324-328: Why would a ML heat budget fail to capture warm water advection into the “Blob” region? Are the boxes drawn with boundaries that lie outside that of the warm water advection process? Wouldn't frontal shift-induced SSTA be maximized along the axis of the SST front, and zonally confined to the SST front?

342-343: Is the 30 deg latitude extent of the SSTA warm pattern consistent with the 200km (2 deg latitude) gyre boundary excursion? What about the scale of atmospheric forcing anomalies associated with surface heat flux anomalies?

(Remarks on code availability)

Reviewer #2

(Remarks to the Author)

This study tried to explore the potential role of gyre scale ocean dynamics in iconic northeast Pacific marine heatwaves, focusing on the interannual or low-frequency timescales. I think this viewpoint is very important and potentially crucial for understanding the climate variability in the Northeast Pacific. But I think the current presentation is quite preliminary in many aspects. Please see my detailed comments below.

Major comments:

In your mixed-layer heat budget equation, the H is time varying mixed layer depth. Please specify how you calculate the surface heat flux term, that is $Q_{net}/(\rho C_p H)$. In particular, how does the mixed layer depth vary in this term? Moreover, I cannot tell exactly how do the gyre variations work on the MLTA in the blob region, because authors merged the oceanic processes into one term. The physical picture is not clear to me.

In many previous studies, as you also shown, the oceanic processes may not be that important for the warm blobs. How to interpret your results?

In this study, authors did not pay special attention to the differences of mechanisms between seasons.

What is the frontal dynamics? I think authors did not use too much frontal theory in this study.

Figure 2: I think there should be a more objective method to select northernmost and southernmost years. Then, significance test should be added to confirm the robustness of your results. Currently, the sample numbers are small. Moreover, authors can add the climatological +0.47 line to show the changes. Same issue for Fig. 3. Currently, I cannot understand well the texts in Line 135-137.

Line 215-218: Previous studies claimed the shallower mixed layer depth during the Northeast Pacific warm blobs/marine heatwaves (Amaya et al. 2020 NC; Shi et al. 2022 GRL). Could authors give more explanations on the deepening of thermocline here.

The neutral point appears a bit arbitrary. Authors tried to raise a new method to identify the onset of the blob, but they should compare in details with the previous results. Why this method is more reasonable than others? Moreover, did Amaya et al. (2020) propose the onset of blob 2.0 in May 2018? I think they only focused on the summer of 2019, rather than 2018.

Minor comments:

The word “iconic” is a bit hard to clearly understand. Could authors change into another word?

Abstract, Line 43: anomalous positive or negative wind stress curl drives the expansion of the subtropical gyre. Please specify here.

Blob 1.0 and 2.0: It is not appropriate to appear in the abstract without any background here. Readers may not know their similarities and differences.

Abstract, Line 52: It is better to specify the sea surface temperature, rather than temperature.

Line 61: Chen et al. (2021) classified two types of warm blobs and reported different mechanisms of the warm blobs in different seasons.

Chen, Z., Shi, J., & Li, C. (2021). Two types of warm blobs in the Northeast Pacific and their potential effect on the El Niño. *International Journal of Climatology*, 41(4), 2810–2827.

Line 64: For Blob 1.0, due to its multi-year persistence over 2013-2015, the growth months may be different in these years. There are some errors in the reference list, such as the missing page information, etc.

Line 85-86: Why the advection is zeros here? Could you please draw a simple diagram to illustrate? Moreover, why do you give this ideal case? I feel confused to understand. How does advection outside the computational domain explain the temperature tendency in the domain?

Line 98-99: Please cite some related papers to confirm this.

Line 391-392: I am wondering whether +0.47 is reasonable for the identification of flow axis in each year/month.

Fig. 1a: climatological mean? Annual-mean? Please specify.

Fig. 1b: Could authors add significance test for this figure? The white shading to the rightmost area indicates there is no +0.47 value, right?

Fig. 1c: the interannual component should be frequency higher-than 9 years. Why do authors use the 1-year filter here?

Fig. 3: Authors did not make it clear the wind vectors and curl are anomalies or original values.

Line 178: confusing for “In the mean”

Line 182 and 184: west of or east of? Moreover, I cannot distinguish the changes without climatological position superimposed. Also for the right column in terms of stream function. Moreover, why these panels are averaged over a narrow band as indicated in Line 193?

Line 203-204: Please give the figures, maybe in the supplement.

Line 276-277: Confusing sentence for "the increased mixed layer heat loss to the air is the response".

Line 337: what did "residual" refer to here?

Line 350-351: What variable for the spatial extent of approximately 30 ° in latitude?

Fig. 5: What is the difference between left and right figures in each panel? In panel c, why does the warm anomaly correspond to cyclonic wind anomalies? PV dynamics is not included in the main texts.

(Remarks on code availability)

Reviewer #3

(Remarks to the Author)

This study analyzed the causes of the "blobs" that occurred in the Northeastern Pacific through a new perspective on large-scale ocean dynamics. The study argued that the positive-negative wind stress curl anomalies in the Northeastern Pacific drove the formation of the blob by shifting the gyre boundary between the subtropical and subarctic gyres. The study has investigated the cause of blobs through a simple heat budget analysis with respect to the gyre boundary variations. Through the negative wind stress curl anomaly, the subarctic gyre contracted, and the subtropical gyre expanded, leading to oceanic heat gain.

The value of this study lies in its examination of blob formation through the oscillations of subtropical and subarctic gyres, rather than focusing solely on the climate variability perspective of air-sea interactions, which was proposed in previous studies. This new viewpoint could provide insights for future research on marine heatwaves and ecosystems in other regions, making it a relevant contribution to the scope of the journal.

However, there are some limitations in the analysis and interpretation of results, and I believe addressing these will significantly enhance the quality of the paper. Here are some major comments:

1. The heat budget analysis is overly simplified. The study divides surface net heat flux and the residual term, attributing the residual term to the ocean's influence. However, the equation presented in the paper assumes that all shortwave radiation enters and exits the mixed layer, while some of it is likely transferred to the subsurface layer beneath the mixed layer. This could result in an overestimation of surface net heat flux and an underestimation of the ocean's influence, which might align with the paper's intent, but accurate calculations and proposals of values would be more appropriate. Additionally, the entrainment term could be helpful to understand the effect of the upwelling & mixed layer depth changes on the blob formation due to the negative wind stress curl anomaly. Relevant references include:

References:

Qiu, C., Kawamura, H., Mao, H., & Wu, J. (2014). Mechanisms of the disappearance of sea surface temperature fronts in the subtropical North Pacific Ocean. *Journal of Geophysical Research: Oceans*, 119(7), 4389-4398.

Murata, K., Kido, S., & Tozuka, T. (2020). Role of reemergence in the central North Pacific revealed by a mixed layer heat budget analysis. *Geophysical Research Letters*, 47(13), e2020GL088194.

2. The positive-negative wind stress curl anomalies weaken (strengthen) the subarctic (subtropical) gyre. According to the Ekman theory, this implies a reduction (intensification) in divergence (convergence) within the surface mixed layer, leading to heat accumulation at the surface and a weakening (strengthening) of upwelling (downwelling), reducing heat loss. Therefore, in the gyre's interior region, the blob formation might be more due to the weakening (strengthening) of the subarctic (subtropical) gyres affecting mixed layer depth and upwelling, (downwelling) rather than advection of the gyre boundary especially. The analysis in the paper does not sufficiently support the idea that heat is transferred from the subtropical gyre to the subarctic gyre. Additional analysis showing the spatiotemporal transmission of heat from the subtropical gyre across the front to the subarctic gyre is needed. This is the reason for that the entrainment term is included in the heat budget analysis.

Additionally, the schematic diagram in Figure 5 suggests that warm water columns move northward via potential vorticity (PV) conservation from the subtropical gyre to the subarctic gyre across the front. However, the negative wind stress curl occurs in the subarctic gyre, not the subtropical gyre, so the figure does not align with the content of the paper. In PV conservation, an increase in H does not necessarily lead to northward movement to higher latitudes but could instead increase relative vorticity. And it is also questionable whether the water column would be able to cross the front and move northward to conserve PV. Therefore, I believe Figure 5 needs significant revision, and if the content of Figure 5 is correct, sufficient discussion should be included in the manuscript.

3. The positive phase period should include not only the year the blob formed but also the years when the blob existed. Therefore, the years 2015 and 2020 should also be included. Additionally, does the negative wind stress curl anomaly occur in both 2015 & 2020 and does the marine heatwaves appear in 2005?

4. In this paper, it is argued that Blob 1.0 started in May 2013, based on 365-day lowpass filtered data. Marine heatwaves are phenomena typically observed on a daily scale, so it is questionable whether identifying them using low-pass filtered data is appropriate. Additionally, an explanation is needed as to why a cold anomaly lasting 40 days was included in the Blob period, and what advantages this inclusion brings.

Minor comments

- Line 138. 'the gyre boundary shifts southward (eastward),' Please verify the sentence.

- Line 182-184, 'resulting in the gyre boundary east of 150W ~ , resulting in the gyre boundary west of 150W shifting northward.' Please verify the sentence.

(Remarks on code availability)

Version 1:

Reviewer comments:

Reviewer #1

(Remarks to the Author)

[Editorial Note: Reviewer #'s attachments can be found at the end of the file]

Review of revised NEP Blob

I really appreciate the careful and extensive work that was done to respond to reviewer comments. This draft has addressed a number of my concerns, but I find that this revised manuscript is difficult to follow and leaves me with a number of important questions that should be addressed before publication.

First, I'm struck by the remarkable trend in the gyre boundary latitude from 1993-2022, where the trend is +2 to +3 st devs! This deserves an article of its own, and seems to be closely related to the rapid increase in North Pacific SSTs in the period since 2013 (Hu et al 2024). Also see the simple SSTA time series figure I created for the 30-50N, 150-135W region that also shows a large warming trend in addition to the interannual extremes in this region from 1993-2023.

Hu et al 2024: Accelerated warming in the North Pacific since 2013. Nature Climate Change. <https://doi.org/10.1038/s41558-024-02088-x>

If impacts on marine life and fisheries are a key concern for the developing science around marine heatwaves, why focus on what I'd call "persistent warm events" (inter annual time scales) rather than the shorter-term extremes and the longer-term trend? Is this something unique to the two warm events in the NE Pacific that are the focus of this work? Are you trying to broaden an argument about Hobday et al's definition of Marine Heat Waves (based on daily SST > 90th percentile values for at least 5 consecutive days)?

On identifying the timing of "event" onset, it seems to me that the SSTA (or MLTA) tendency is more likely to provide insights into the underlying physical mechanisms. Where SSTA/MLTA crosses the zero line does not seem all that important to me if the interest is in understanding the processes, as the zero-line can be arbitrarily defined based on different reference periods.

On the time history of atmospheric forcing: Why not plot the time series of your wind stress curl forcing pattern, which presumably drives both the surface heat fluxes and the Sverdrup response (with some lag-time). I've plotted a simple time series of 12-month low-pass filtered SLP anomalies for 30-50N, 150-130W (attached), and it shows substantial inter annual variability, multi-decadal variations, and century-long trends. This region is part of the wintertime TNH teleconnection pattern, but these variations are not limited to the winter season or the TNH overall pattern that has centers of action outside the NE Pacific (see the attached annual mean SLP difference map for 2014-2022 - 1993-2002).

In summary, I found a lot of really interesting material in this work, with the major contribution being the clear demonstration of the primary role for ocean dynamics in the warming of the NE Pacific at multiple time scales. I do not like the focus on the "blobs" because that terminology is too vague for my tastes, and it limits the focus to just two persistent warm events that are now a part of a series of warm extremes that are also embedded within a longer-term ocean warming that is still playing out. Finally, the nature of the atmospheric forcing, and how that atmospheric forcing drives both surface fluxes and ocean dynamics, remains hard to follow by the approach taken here. If atmospheric forcing has multi-season and longer time scale periods, wouldn't we expect to see air-sea heat exchanges having those same time scales (assuming that the warming of the ocean doesn't become so great that it comes into equilibrium with surface air temperature and humidity)?

In addition to these over-arching comments I have a number of specific comments following the line numbers below.

Sincerely,
Nate Mantua
NOAA/NMFS Southwest Fisheries Science Center
Santa Cruz, CA

1-2: To the extent that NEP blob = NEP MHW, this title is redundant. Maybe stick with MHW, or "persistent MHW events"?

38: I am not a fan of continuing to use the "blob" terminology because it has no criteria or definition that could be objectively applied to these or other events. Why not say "multi-season" or "persistent" MHW events in 2013-15 and 2019-20 instead?

50: the proximate driver of surface wind-stress curl over the NE Pacific clearly projects onto the TNH pattern, but the TNH

pattern was only identified for winter months and it also has centers of action over N. America ... what about a simpler index of SLP over the NE Pacific to track the local/regional atmospheric signature of the relevant wind stress forcing? What are the timescales of that SLP index variability?

72-83: This paragraph makes an argument for ocean dynamic responses favoring multi-season persistence of warm events, rather than the initiation, of the "marine heatwave" extremes that have been previously defined as daily SSTs exceeding 90th percentile values for more than 5 consecutive days (Hobday et al 2016).

115: are ocean dynamics key to the formation of the 2013 and 2019 MHW events, or to the multi-season persistence?

120-121: The same persistent wind stress curl patterns that drive ocean dynamics are also driving surface heat fluxes and related thermodynamics. Bond et al's focus on the early part of the 2013-14 MHW showed that both ocean dynamics and thermodynamics contributed to the initiation and persistence of the event (Sept 2013-Feb 2014).

142-145: Figure 1 shows a remarkable trend over the 1993-2022 period, where the trend is ~ +2 to +3 st devs! This could be the focus of a related journal article that explored the dynamic contributions to NE Pacific SST warming trends over past 3 decades.

148: As noted above, Figure 1 shows an incredible trend in the NPC latitude that was especially confined to the far eastern Pacific (east of ~145W) from 2014-2017 but expanded to include the entire Pacific from 2018-2022. The period of northward expansion is also characterized by rapid and persistent warming of the entire N. Pacific (Hu et al. 2024).

161-165: while your detrended data are good for identifying inter annual peaks, the unfiltered data shows that the positive phase encompasses the entire 2013-2022 period.

211-225: This paragraph about onset dates from this versus other studies is difficult to follow. I understand that this work argues for the importance of the inter annual time scale over the sub-yearly variations. However, Figure 3 has so much information over so many wiggles and lines that my brain cannot simply look at these panels and extract what the authors are trying to convey. One possible remedy to this would be to not show the continuous time series from 2010-2023. But instead limit the figure to the warm events of interest +/- a year on either end. There is simply too much information presented here for my taste.

235-241: 3d shows that the net sfc heat flux and advection terms were positive in the first half of 2013, during a time that the temperature anomaly increased from negative to positive values. Isn't this an indication that dynamic and thermodynamic processes led to the change from cool to warm SSTA/MLTA in the focal region?

242-245: Again, I'm not a fan of "the blobs" terminology because they are so loosely defined. Your use here looks to be defined as periods of warmer than normal SST/MLT that persist for > 1 year. This is clearly at odds with Hobday et al (2016) definition of MHW events as extreme daily SST anomalies (>90th percentile) that persists for at least 5 consecutive days. In your Introduction you could explicitly state that this work is focused on the inter annual time scales over shorter time scales associated with the Hobday et al definition. And then show your results that support the importance of inter annual time scale processes in the formation and persistence of the 2013-2016 and 2019 warm events in the NE Pacific.

267-269: The +/- phase terminology here is confusing because of the dipole nature of the windstress curl anomaly patterns. Your + phase has an expanding subtropical gyre in the NE Pacific, and a contracting subtropical gyre in the central Pacific. Your negative phase has the subarctic gyre expanding and the subtropical gyre contracting in the east with little change farther to the west.

315-318: Yet the streamfunction anomaly pattern shown in panel e (2014) doesn't look anything like those in panels d and f. I agree that the upper ocean warming is similar, but not the streamfunction anomalies.

325: replace "suggest" with "show"

379-381: Fig 6(a) shows that the net heat flux and latent heat flux are positive for most of 2013 and 2014, consistent with surface fluxes promoting a positive MLTA tendency in those years.

412-413: Does the SLPA pattern propagate northward, or does the SW/NE dipole with centers in the western tropical Pacific and NE Pacific intensify following while the SLP anomalies in the eastern tropical Pacific weaken over the 5-months lag time?

447: The emergence of the 2013-14 warming looks to be the result of persistent atmospheric forcing that generates rapid upper ocean warming due to dynamic and thermodynamic processes. The persistence and warming at depth look to be attributed to the dynamic processes you focus on here.

460-461: I remain unconvinced that ocean advection is the only factor contributing to the emergence of the two warm events examined here. The surface heat flux anomaly maps included in your rebuttal letter show that 2013 had a pattern of positive surface heat flux anomalies that spanned the entire NE Pacific. This was also a key result of Bond et al's heat budget analysis for the Sept 2013-Feb 2014 period.

506-512: You also show an impressive multidecadal trend in the gyre boundary, and others have shown a corresponding trend in NP SSTs and mixed layer depths. The same trend exists in the atmospheric SLP field. So there are long time scales in atmospheric forcing worth considering too. You could create a more targeted atmospheric forcing index based on the SLPA pattern. A spectral analysis of that time series will show the time scales the atmospheric forcing pattern (for wind stress curl and sensible and latent heat fluxes).

614: Do you mean the red line in 3c (MLTA tendency) and red shading (SSTA) in 3d? or do you mean the MLTA tendency and Box TA tendency? There is no red shading in 3c.

Rebuttal letter:

P9, R1-4: I am surprised by this statement: "Consequently, it is proposed that the anomaly in temperature transport due to the northward expansion of the subtropical circulation is counterbalanced by the net heat flux anomaly, leading to a lack of a linear warming trend in local SSTA." There is clearly a strong positive warming trend in the region 30-50N, 150W-135W from 1993-2022. Also see Hu et al 2024: Accelerated warming in the North Pacific since 2013. Nature Climate Change. <https://doi.org/10.1038/s41558-024-02088-x>

Figure R1-7: I'd say that the spatial pattern of surface heat flux anomalies in 2013 encompassed nearly all of the NE Pacific, a pattern that is unique in the 2003-2022 period for the extent and magnitude of positive surface heat flux anomalies in the NE Pacific.

(Remarks on code availability)

Reviewer #2

(Remarks to the Author)

I thank the authors for their great efforts in addressing the comments and concerns raised by the three reviewers. Although some of my comments have been well addressed, I may have more concerns to put forward in this round of review:

1. The motivation or evidence of authors to focus on the interannual timescales is not clear, although interannual time scale/1-year filter is used in many places throughout the manuscript.
2. In the revision, authors used a box model to try to give calculation for the heat budget of the blob marine heatwaves. However, based on the method description from Line 620, this method may not well distinguish horizontal and vertical advection processes. On the other hand, the relationship between the mixed layer heat budget and box model heat budget is not well addressed, which may affect the consistency and readability of this paper. In the following subsection b), authors give detailed discussion on the entrainment term, whose relationship with the box model also needs clear description.
3. Related to comment 2, the response last time mainly addressed from the perspective of vertical processes, however, I think authors tried to explain the warm water transportation from the view of horizontal advection. But authors did not show any advantages of the box model from the view of horizontal advection.
4. As other reviewers also pointed out the concern on the relationship between the shoaling mixed layer depth anomalies and deepened thermocline depth anomalies, I think authors should give the clear figures (spatial pattern and temporal evolution) to better clarify this issue.
5. In terms of the circulation pattern, authors attributed to the TNH, rather than NPO. I think authors should note that the variance of TNH, based on the EOF analysis, is only around 5%, which may not well explain the frequent occurrence of marine heatwaves over the past decade. Moreover, the existence of TNH in the abstract as the main findings may not be good as they have been already reported by other studies.
6. In the Fig. R1-3, I cannot see any obvious trend, although authors have processed the data.

Additional comments:

1. Tracked-changes version: I cannot see any tracked changes. Please make sure that authors upload the correct file.
2. Affiliation 7 did not belong to any author. Please check.
3. Fig. 5e,f: Why did authors show the magenta lines here?
4. Although authors added some statistical significance tests in the revision manuscript, there are other places that should have such test, for example, Figs. 2g, h.

(Remarks on code availability)

Reviewer #3

(Remarks to the Author)

Comments

Most of the points I previously mentioned have been appropriately revised, so I suggest acceptance once the minor comments are addressed.

Minor comments

Line 91: In the phrase "geostrophy influences ~", geostrophy itself cannot induce vertical motion. However, vertical motion

can occur when baroclinicity and the Ekman effect act together. This sentence may lead to misunderstanding, so I suggest revising it to explicitly include this explanation.

Figure 2: Changing the colorbar for the Difference plots to a different color would make the distinction between (a)-(f) and (g)-(i) clearer.

(Remarks on code availability)

Version 2:

Reviewer comments:

Reviewer #2

(Remarks to the Author)

Thanks for authors to carefully address my concerns. I am mostly clear about my concerns. But I still have some minor comments before the manuscript being potentially accepted.

1. Figure R2-1 can be moved to supplementary information, but now I cannot find any supplementary information.
2. In the last paragraph of response to my second comment, it would be great if authors could add figures to show each term and their corresponding correlations.
3. In the last paragraph of response to my third comment, I think authors could re-write it especially the last sentence and try to use the same marks (including consistent upper or lower cases with previous comment). Why is the first term on the right hand side derived from six boundaries of the box?
4. I think the deepening of thermocline should be illustrated explicitly. Please show the figures.

As I was asked to take a look at the points raised by Reviewer #1 as well, I will also include those comments as below:

1. I suggest authors to give specific line numbers for their modifications, which is not clear now.
2. In terms of the response to #1 comment, authors did not prove the causality between the warming trend and displacement of the gyre boundary. Thus, "a major contributor ..." needs to be rephrased.
3. "For the multi-season persistence, shifts in the Sverdrup balance are favored by the low frequency component of the atmospheric forcing, with the ocean acting as a low-pass filter to the atmospheric forcing": By saying these, are ocean dynamics key to the multi-season persistence? I am not clear. Would it be possible to quantify the contribution to the SSTAs of the MHWs?
4. "ocean advection is the only factor contributing to the emergence of the two warm events": How to reconcile with previous studies? I think this issue should be better elaborated in the manuscript.

(Remarks on code availability)

Version 3:

Reviewer comments:

Reviewer #2

(Remarks to the Author)

Thanks for addressing my concerns and comments previously. I am now glad to accept this paper.

(Remarks on code availability)

Reviewer #1 (Remarks to the Author):

This manuscript reports on an investigation of a novel hypothesis that a dynamic ocean response to wind stress curl forcing played a fundamental role in two recent multi-year NE Pacific marine heatwaves. The authors conclude that interannual variations in gyre scale frontal dynamics comprehensively explain the spatial characteristics and multi-year persistence of the two events. The article is interesting for offering up this idea and some provocative interpretations of the results of their analyses. However, I do not believe that the work detailed here supports such grand conclusions. As noted in my specific comments below, simply showing that these events featured meridional excursions of the subarctic frontal boundary falls short of demonstrating cause and effect. Moreover, there is really no analysis or theory offered for linking the proposed mechanisms to the observed time-scales involved in the multi-year marine heatwaves of interest. If the event duration times were determined by the atmospheric forcing, for instance, one would need to explain the observed evolution of the atmospheric forcing field (something that DiLorenzo and Mantua focused on for the 2014-15 period, and Amaya et al. focused on for summer 2019). This work has focused more on a time-varying but spatially invariant interpretation of the atmospheric forcing - for example, creating composite maps (Figs 2 and 3) for sea level, SST, and the surface wind stress fields during what are termed “positive” and “negative” phase years in their detrended and 1-yr filtered gyre boundary index (which has its maximum value in 1993, a year that is not used in the composites).

Thank you very much for your thoughtful review. We have now revised our manuscript based on your comments and those of the other two reviewers. Our detailed responses are provided below, with your original comments in black font and our replies in blue font.

In the revised manuscript, we have endeavored to contextualize our analysis and proposed mechanisms within the framework of existing theories. Overall, the heat budget analysis indicates that the advection term is responsible for the onset of the Blobs, while the increased surface net heat flux from the ocean to the atmosphere accounts for its decay.

We employed the Sverdrup relation to elucidate the oceanic dynamic response to the anomalous wind stress curl. The Sverdrup relation serves as a foundational theory in ocean general circulation, delineating the relationship between meridional mass transport and the thickness of the water column. Although this relationship is effectively time-independent, the composite maps are effective for

interpreting the quasi-steady-state (or equilibrium) responses to differences in the wind stress curl, and has been validated in some previous studies (Jiang et al., 2006; Hill et al., 2008; Thomas 2014). Specifically, during the positive phase, the negative anomaly in wind stress curl over the subarctic gyre results in a weakening of the gyre, leading to its contraction and facilitating the northward intrusion of warm subtropical waters. This explanation contrasts with the previous understanding that reduced ocean heat loss to the atmosphere played a major role in the formation of the Blobs.

We find the anomalous latent heat flux is primarily responsible for the decay of the Blobs. It results from the increase in latent heat flux loss from the ocean to atmosphere, linked to the ocean thermodynamic response to the northwesterly surface wind anomaly that transports cold air.

Due to significant changes in the manuscript compared to the previous submission, we have summarized the revisions here. The Introduction has been completely rewritten to provide a more comprehensive explanation of our concerns about the classic mixed layer heat budget analysis applied previously. The 'Mixed layer heat budget' section has been restructured to follow the 'Gyre boundary variation in the 'Blob' region', highlighting the importance of advection in the heat budget. Furthermore, we have added two new sections: one examining the variations in net heat flux and another addressing the physical drivers of the Blobs. The final section has also been revised to incorporate our latest findings.

References

- Jiang, H., Wang, H., Zhu, J. & Tan, B. Relationship between real meridional volume transport and Sverdrup transport in the North Subtropical Pacific. *Chinese Science Bulletin* **51**, 1757–1760 (2006).
- Hill, K. L., Rintoul, S. R., Coleman, R. & Ridgway, K. R. Wind forced low frequency variability of the East Australia Current. *Geophysical Research Letters* **35**, (2008).
- Thomas, M. D., De Boer, A. M., Johnson, H. L. & Stevens, D. P. Spatial and temporal scales of Sverdrup balance*. *Journal of Physical Oceanography* **44**, 2644–2660 (2014).

This work has raised some interesting questions for me that I think are worth addressing. These include:

- The monthly mean gyre boundary index shows a substantial trend over the 1993-2022 period - is there a corresponding trend in the surface wind stress field (for instance, in the zero-curl contour latitude in the NE Pacific)?

Thank you for your comment. Yes, there is a corresponding trend in the atmospheric field.

In the Alaskan gyre region where the correlation coefficient between the gyre boundary and wind stress curl reaches a maximum (140-152 °W, 50-57 °N), we found a noticeable linear decrease in the absolute values of the surface wind stress curl with -3.9×10^{-9} N/m³ per decade, and in the sea level pressure anomalies with a value of 79 Pa per decade (Figure R1-1; we also added this figure as Fig. 7b in our revised manuscript). This suggests in the region where the wind stress curl anomaly has the greatest influence on the gyre boundary, there exists a corresponding trend in the surface wind stress field.

However, the zero-curl contours in the NE Pacific are winding and with many smaller scale features, even using the annual mean data (Figure R1-2, black contours). Consequently, we selected the longest zero-curl contour (Figure R1-2, black thick contours) and averaged its latitude to a resolution of 1 degree in longitude. Subsequently, we computed the linear trend of the zero-curl contour and identified a northward displacement, characterized by a trend of 1 degree per decade (Figure R1-3).

Figure R1-1: Areal mean wind stress curl (red line) and sea level pressure (blue line) over the region (green boxes) where the correlation coefficient between the gyre boundary latitude and wind stress curl anomaly reaches maximum.

Figure R1-2: The yearly mean wind stress curl. Black contours denote the zero-curl contour and black thick contours denote the wind stress curl selected to calculate the zero-curl contour latitude.

Figure R1-3: The linear trend of the zero-curl contour latitude. The red dots represent the point where the p -value is less than 0.05.

- How would the gyre strengths and positions vary in response to the kind of atmospheric forcing variations described by Di Lorenzo and Mantua (2016)? Would they include the expansion of the subtropical gyre in 2013 through winter 2014, or a slowing of both the subarctic and subtropical gyres in response to the atmospheric forcing in the “North Pacific Oscillation” pattern that DiLorenzo and collaborators have linked with the North Pacific Gyre Oscillation?

Thank you for your comment. Di Lorenzo and Mantua (2016) did not provide a distinct pattern of wind stress curl; rather, they showed the seasonal evolution of sea level pressure anomalies in their Figure 1. Our analysis of the time series of gyre boundary latitude indicates that interannual variations are significantly greater than seasonal variations (Figure 1c, comparison between the red and blue lines). This implies the ocean dynamic response to the atmospheric driving field favors the interannual variation, rather than intra-annual variation.

It has been previously argued that the ocean's response to atmospheric dynamics in the Northeast Pacific has a damping time scale of 1-3 years (Cummins & Lagerloef, 2004; Cummins & Freeland, 2007; Cummins & Masson, 2018). In this context, it would seem less appropriate to discuss changes in ocean circulation based on the atmospheric forcing pattern averaged over a specific month or a few months unless it designates the state of slow variations on the timescale over one-year. If your concern is that the pattern may present the interannual variation, please refer to our response to the following comment.

Di Lorenzo and Mantua (2016) reported, “*We have shown that both the 2014 and 2015 SSTa patterns can be reconstructed independently using direct atmospheric forcing functions. This suggests that ocean internal processes may not have played a key role in persisting and evolving the record amplitude of the SSTa through 2015.*” According to their analysis of the AR1 model, it was concluded previously that ocean dynamics might not have been a significant factor. The previous research does not account for the expansion of the subtropical gyre from 2013 to winter 2014, nor the slowing of both the subarctic and subtropical gyres as far as we can establish here.

References

- Cummins, P. F. & Lagerloef, G. S. E. Wind-driven interannual variability over the northeast Pacific Ocean. *Deep Sea Research Part I Oceanographic Research Papers* **51**, 2105–2121 (2004).
- Cummins, P. F. & Freeland, H. J. Variability of the North Pacific Current and its bifurcation. *Progress in Oceanography* **75**, 253–265 (2007).

Cummins, P. F. & Masson, D. Low-frequency isopycnal variability in the Alaska Gyre from Argo. *Progress in Oceanography* **168**, 310–324 (2018).

- Would the fall 2014/winter of 2015 feature an expansion of the subarctic gyre but contraction of the subtropical gyre? These SSH variations and their links to the gyre circulations and atmospheric forcing are basically those described by Cummins and Lagerlof (2004), your reference 20.

Thank you for your comment.

If we consider the atmospheric forcing pattern from fall 2014 to winter 2015 as a representation of interannual variation, it can influence ocean circulation changes. Previous studies have explored how wind patterns affect circulation in this area (Cummins & Masson, 2018; Hristova et al., 2019), so we will only provide a brief overview. According to ocean general circulation theory, specifically here the Sverdrup relation, the North Pacific subtropical gyre is forced by negative wind stress curl, while the subarctic gyre is influenced by positive wind stress curl. For the interannual variation, a reduction in the negative wind stress curl over the subtropical gyre is expected to result in either a weakening or contraction of the gyre. Conversely, an increase in the positive wind stress curl over the subarctic gyre is anticipated to lead to either a strengthening or expansion of that gyre. This relationship also holds true in the reverse scenario. Furthermore, negative sea level pressure (SLP) anomalies typically align with positive wind stress curl, and vice versa.

The uniformly negative SLP anomaly over the northeast Pacific (atmospheric forcing pattern from fall 2014 to winter 2015) leads to a positive wind stress curl anomaly, which increases the positive wind stress curl in the subarctic gyre and decreases the negative wind stress curl in the subtropical gyre. This results in the expansion of the subarctic gyre and the contraction of the subtropical gyre, effectively shifting the gyre boundary southward. Our findings indicate that there was indeed a southward shift in the gyre boundary during fall 2014/winter 2015 (Fig. 1c, blue solid line).

References

- Cummins, P. F. & Masson, D. Low-frequency isopycnal variability in the Alaska Gyre from Argo. *Progress in Oceanography* **168**, 310–324 (2018).
- Hristova, H. G., Ladd, C. & Stabeno, P. J. Variability and Trends of the Alaska Gyre from Argo and Satellite Altimetry. *Journal of Geophysical Research Oceans* **124**, 5870–5887 (2019).

- Could you better evaluate the relative contributions of basin-scale ocean dynamics versus surface flux forcing through simulations that involved forced ocean responses in a slab-ocean (mixed layer only) versus a dynamically-responding ocean model?

Thank you for your comment and your suggestion, which we acknowledge would be a very nice separate modelling study. However, we consider this to be beyond the scope of the current manuscript. In lieu of this suggested modelling approach, we have instead substantially improved our diagnostic analysis to strengthen the reliability of our conclusions. In the 'Mixed layer heat budget' section, we used a box model to decompose the temperature tendency into the terms representing the net surface heat flux, the net lateral advection solely, and the residual (Figure 3), to more completely explain the advection contribution. The result shows the lateral advection term accounts for the positive temperature tendency during the onset of the Blob and supports our hypothesis that the warm subtropical water displacing into the typical subarctic region is the causal mechanism.

We included a more in-depth examination of the impact of net surface heat flux during the Blobs in the “Anomalous wind direction explains net heat fluxes” section, and found the decrease in SSTA during the decay phase of the Blobs is primarily attributed to an increase in latent heat flux loss from the ocean to the atmosphere, linked to the ocean thermodynamic response to the northwesterly surface wind anomaly that transports cold air into the Blob region.

In summary, I believe there is merit in the research directions this work reports on. However, I do not believe that the research reported here supports the conclusions that are offered. I recommend this manuscript be returned to the authors with a request for major revisions.

Nate Mantua
NOAA/NMFS Southwest Fisheries Science Center
Santa Cruz, CA

Specific comments:

40: I would add “and fisheries” after “marine ecosystems”

Thank you for your comment. We followed your advice by adding “and fisheries” after “marine ecosystems”.

43-44: revise to “meridional expansion (contraction” of the subtropical (subarctic) gyre” ... current text doesn't include subarctic

Thank you for your comment. This sentence is not revised because the northward excursion of the gyre boundary is the representation of the expansion of the subtropical gyre and contraction of the subarctic gyre.

The contraction of the subarctic gyre is reported in the section of “Wind stress curl anomalies drive gyre boundary poleward displacements”. Fig 5. (d) in the revision shows direct evidence of this variation. The red shading over the upper 100m indicates an enhancement of the anticyclonic rotation over the northeast Pacific. If we add the background value, this anomaly value designates a contraction in the subarctic gyre and an expansion of the subtropical gyre.

In the revision, we have reduced the emphasis on changes in gyre boundaries and enhanced the focus on the expansion and contraction of the two gyres.

45-47: this analysis does not isolate the contribution to gyre scale frontal dynamics from other processes, nor does it connect those dynamics to the time scale of the events, so it falls short of “comprehensively explain(ing) the spatial characteristics and multi-year persistence of these ... heatwaves”

We have deleted this sentence and instead followed your suggestions conducting more analysis on the net heat flux and physical drivers, to make our analysis more comprehensive and conclusions more compelling.

60: reasons for multi-year persistence have been investigated and explained by others (e.g., DiLorenzo and Mantua 2016)

Thank you for your comments. We revised "the reason for their different onset months and multi-year persistence remains unclear" to " the reason for their different onset months and multi-year persistence remains the subject of further consideration and exploration." After "multi-year persistence," we cited DiLorenzo and Mantua (2016).

78-80: Isn't a dominant tendency term required to understand the origins of a broad-scale warming of the upper ocean that happens within a single season, hence the focus on atmosphere-ocean interactions?

Thank you for your comment. We don't think so. Our original submission did not clarify this issue clearly, and we have now revised this section in the third paragraph of the Introduction in the revised manuscript.

In particular, the heatwaves are linked to positive extremes in SSTA. Since the ocean and atmosphere influence SSTA at different frequency bands or time scales during air-sea interactions, it is essential to first determine which frequency band or time scale is most influential in the changes in SSTA, and then analyze the dominant tendency term at that specific frequency band or time scale.

Calculating temperature tendencies and identifying the main factors using SSTA filtered on an unsuitable timescale (or unfiltered) may lead to misunderstanding. This is because atmospheric changes occur at a higher frequency than oceanic changes. The temperature tendencies resulting from atmospheric processes are associated with high-value segments, while those from oceanic processes relate to low-value segments. Analyzing unfiltered SSTA without considering the timescales that primarily affect positive SSTA may skew the results towards higher values, which are coincidentally linked to the atmospheric dynamics.

83-86: This may be true in an ideal case, but this real-world case study does not involve a surface mixed layer having top and bottom interfaces that are adiabatic.

Thank you for your comment. We deleted this ideal case for it leads to confusion.

120-125: Causation may go either way, or there may be a common forcing for the SSTa and latitude of the gyre boundary.

Thank you for your comment. The similarity between the latitude of the gyre boundary and SSTA cannot fully establish a causal relationship. However, the latitude of the gyre boundary corresponds to the advection term in the mixed layer heat budget, and the mixed layer heat budget can help us determine that SSTA is indeed influenced by lateral advection.

I also think that the trend over the 1994-2023 period of record is striking. Is there is a similar trend in atmospheric forcing?

Yes, there is a corresponding trend in the atmospheric field. In the region where the correlation coefficient between the gyre boundary and wind stress curl reaches a maximum (140-152 °W, 50-57 °N), we found a noticeable linear increase in the absolute values of the surface wind stress curl with -3.9×10^{-9} N/m³ per decade, and in the sea level pressure anomalies with a value of 79 Pa per decade (Figure R1-1; we also added it as Fig. 7b in our revised manuscript).

Why doesn't the daily SSTA time series share the longer-term trend with the gyre boundary index?

We found that there is a linear trend primarily influenced by net heat flux in the net heat flux term anomaly, indicating a steady increase in ocean heat loss (Figure R1-4). Consequently, it is proposed that the anomaly in temperature transport due to the northward expansion of the subtropical circulation is counterbalanced by the net heat flux anomaly, leading to a lack of a linear warming trend in local SSTA.

Figure R1-4: Decomposition of the net surface heat flux term. The data used to draw the figure are

derived according to the equation: $\widetilde{\left(\frac{Q}{h}\right)} = \frac{\bar{Q}}{\bar{h}} - \frac{\bar{Q}\bar{h}}{\bar{h}^2} + residual$ (Shi et al., 2022). A tilde denotes the climatological anomaly while an overline denotes the climatological daily mean. The climatological daily mean was removed from the result and then a 365-day low-pass filter was applied. The data used in the following figure are from BRAN2020.

Reference

Shi, J. et al. Role of Mixed Layer Depth in the Location and Development of the Northeast Pacific Warm Blobs. *Geophysical Research Letters* **49**, (2022).

131-134: Why do you conclude that the subarctic gyre west of 150W expands southward? I can't see the southward expansion of the subarctic gyre west of 150 from the SLH or SST difference maps (panels e and f) – do you mean the negative SSH differences between 40-50N and 180-170W?. I don't see this in Fig 1b either (I would expect to see a dipole centered on ~150W, but that is only a feature of a few years like 2014-2016). The main changes in frontal position are east of 140W from 2014-2016, and east of 165W from 2018-2020 (based on Fig 1b).

We appreciate that you pointed out this point. We revised the text to 'During the positive phase, the subtropical gyre located east of 175 °W expands northward, leading to a southwest-northeast oriented tilt of the gyre boundary (Fig. 2a). In contrast, during the negative phase, the gyre boundary shifts southward and assumes an almost zonal orientation (Fig. 2d)'.

Figures 2 and 3: I think it would be best to use the same projection in the maps shown in Figures 2 and 3.

Thank you for your comment. We now use the same map projection in the maps shown in Figures 2 and 3 of the revised manuscript.

168-170: Interesting that the zero curl contour at 180 is about 5 deg latitude farther south in your positive phase than negative phase composite, while it is at about the same latitude for both phases on the west coast of N. America ~125W; This seems at odds with the SSH composites in Fig 2

Thank you for your comment. The zero curl contour and the latitude of the gyre boundary serve as indicators of changes in wind stress curl and ocean circulation. However, these two parameters do

not need to be exactly aligned in latitude, but rather in their orientation. The relationship between the ocean circulation and wind stress curl can be confirmed by comparing Fig. 4c with Fig. 5a. The negative anomaly in wind stress curl (shown in Fig. 4c) leads to a contraction of the subarctic gyre (indicated by blue shading in Fig. 5a and red shading in the subarctic region of Fig. 5d) and an expansion of the subtropical gyre (represented by red shading in Fig. 5a and the subtropical region in Fig. 5d). This variation occurs between latitudes 30-60 °N, with the gyre boundary being the area where these changes are most pronounced.

176-179: Fig 3a shows your “positive phase” having the gyre boundary east of 150W shifting northward, not southward, in response to a negative wind stress curl anomaly in the Gulf of Alaska (north of 50N).

We appreciate that you pointed out this point. We revised ‘southward’ to ‘northward.’

180-181: This is confusing. Perhaps you could include the actual wind stress curl anomalies for the composites of positive and negative phases and this would be easier for readers to see and understand than trying to assess anomalies from the composites shown here.

Thank you for your comment. In revision, we added both the absolute wind stress curl and wind stress curl anomaly for the positive and negative phase, and wind stress curl anomaly in 2014 and 2019 in Fig. 4.

Figure 3 a-b-c and Fig 5 a-b: Wind vectors are really small and difficult to see, even at 200% magnification. Please make them bigger.

Thank you for your comment. In revision, we made all the wind vectors bigger.

211-212: Thermocline deepening is inconsistent with previously published findings (like Amaya et al 2020 that show a shoaling mixed layer depth in summer 2019)?

Thank you for your comment. This is not contradictory; we are discussing the variations in the main thermocline, primarily caused by geostrophic processes, while the key factors affecting the mixed layer depth may differ. The mixed layer is influenced by changes in wind-induced mixing, surface net heat flux, stratification of the bottom of the mixed layer driven by changes of temperature and salinity,

as well as advection processes.

Although Amaya et al. (2020) noted a shallower mixed layer depth in the summer of 2019, Bond et al. (2014) also observed a deeper mixed layer in winter. Additionally, Amaya's study area is located in the southeast of the Blob region. Their Figure 1c indicates that the shoaling of the mixed layer identified in the area of 34° – 47° N, 213° – 232° W appears to be a northern extension of the deepening mixed layer found at lower latitudes (south of 34° N).

[REDACTED]

Screenshot of Figure 2c. in Amaya, D. J. et al. Are Long-Term changes in mixed layer depth influencing North Pacific marine heatwaves? *Bull. Amer. Meteor. Soc.* **102**, S59–S66 (2021).

241-249: I don't find this new definition of event onset to be very useful or convincing. Is there something special about climate dynamics in the month of May that makes you think this event onset criteria leads to a new and better understanding for NEP marine heatwave events?

Thank you for your comment. In the revision, by carefully examining the 365-day low-passed SSTA, the neutral point for Blob 1.0 is identified as March 19, 2013, while the neutral point for Blob 2.0 is determined to be May 6, 2018. Revisions have been made in accordance with these recent findings.

Our analysis of the emergence of Blob 1.0 seeks to address the question posed by Amaya et al. (2020) regarding the differences in timing between Blob 1.0 and Blob 2.0. They noted, "*An important distinction between the recent temperature anomalies (referred to as Blob 2.0) and those from 2013/2014 (referred to as Blob 1.0) is that Blob 2.0 primarily intensified during the summer months. In contrast, Blob 1.0 developed in the winter due to a persistent atmospheric ridge in the Northeast Pacific, which weakened the climatological Aleutian Low and associated surface winds.*"

While defining the onset as October is reasonable from an event perspective, it leads to entirely different conclusions when analyzing the mechanisms. A comparison of Fig. 3b and 3c reveals that the classic mixed layer heat budget, after applying a 90-day low-pass filter, indicates that both ocean processes and surface net heat flux contribute to warming at the time marked by the black double-headed arrow. In contrast, the results from a 365-day low-pass filter show that during the Blobs, ocean processes consistently have a positive contribution, while surface net heat flux consistently has a negative contribution (the direction of the flux is from ocean to atmosphere). This raises the question of which explanation to trust.

The primary fluctuations in SSTA in the Blob region occur on an interannual scale, suggesting that understanding the positive phase of the interannual component, rather than higher frequency variations of SSTA, can explain most of the positive anomalies. Therefore, we argue that focusing on the interannual timescale, particularly by analyzing SSTA after a 365-day low-pass filter, is more suitable for finding the casual mechanism.

Apparently, the climate dynamics differs in March and October. We did not examine this because this is outside the scope of this manuscript.

Figure R1-5 (Fig. 3 in the revision). The mixed layer heat budget. | (a) Mixed layer temperature anomalies (MLTA, \square , red shading); also shown are the MLTA low-pass filtered by a 90-day cutoff (blue line) and MLTA low-pass filtered by a 365-day cutoff (black line). (b) Mixed layer temperature tendencies (\square /day) as a function of time, including net surface heat flux and integrated ocean process contributions from the tendency budget, derived from daily resolution data, with the annual cycle removed and 90-day lowpass filtered. (c) Same as (b) but 365-day lowpass filtered. (d) The heat budget for a box model, and 365-day lowpass filtered. The magenta arrows denote the neutral point. The black arrow denotes the cooling event in 2017. The black boxes in (c) denote the periods of ‘Blobs’. The black boxes in (d) denote the period of the positive MLTA tendency that involved the onset of ‘Blobs’.

Would you say that Blob 3.0 started in January 2022 by your criteria?

We are uncertain about this. According to our definition, Blob 3.0 appears to be an extension of Blob 2.0, as the interannual component of SSTA has not dropped below zero, indicating that there is no neutral point separating Blob 2.0 from Blob 3.0.

Did the 2013-14 event end in 2015 or 2016 by your criteria?

It varies. The discrepancy in the timing (onset or decay) of the Blobs arises from the different foci of the disciplinary communities. The physical oceanographers' concern is to explain how the positive SSTA was formed while the ecologists aim to understand how this above threshold SSTA affects the marine ecosystem. Apparently, the timing of SSTA surpassing or dropping below the threshold works more effectively for evaluating the biological impact of the temperature extreme, while the timing of SSTA exceeding or falling below the zero value works for examining the underlying physical mechanisms. These two definitions are not contradictory, as they reflect distinct perspectives from disciplinary communities.

The SSTA dropped below the threshold during the winter of 2015, and it went below zero in the winter of 2016. From a mechanistic perspective, we recommended the winter of 2016 as the endpoint, whereas communities like ocean ecologists, who are more concerned with the impact of SSTA on the ecological environment, the winter of 2015 may be a better choice.

By and large, the neutral point of SSTA (from positive to negative phase, or from negative to positive phase) is a better choice when analyzing the physical mechanisms behind the marine heatwave events.

276-277: Did the marine heatwaves “result from” or “include” poleward expansion (contraction) of the subtropical (subarctic) gyre?

The marine heatwaves are a consequence of the poleward movement (or shrinkage) of the subtropical (or subarctic) gyre. The findings from the mixed layer heat budget attest to this. The mixed layer heat budget analysis performed on the primary frequency band of SSTA indicates that oceanic advection contributes significantly to the fluctuations in the SSTA.

281-282: This analysis does not address the issue of “persistence”, but instead focuses on “onset”. What sets the time scale for these events? Is it the persistence and pattern of atmospheric forcing (along the lines of Di Lorenzo and Mantua's analysis of the 2013-2015 period)? Or something else?

See combined response to this question and the one below in our following response.

295-297: If the basin-scale wind-stress curl is the key driver for the gyre variations, then multi-year persistence of these marine heatwaves is due to persistence in the wind-stress curl forcing. So this then begs the question – **what provides persistence in the basin-scale atmospheric forcing of the gyre circulation?** As noted, the ocean circulation (and mixed-layer heat content) response integrates atmospheric forcing so it has greater variance at longer time scales than the atmospheric forcing, yet the evolution of the atmospheric forcing has already been shown to play a key role in understanding the evolution and persistence of the 2014-2016 (DiLorenzo and Mantua) and 2019-2020 (Amaya et al) events.

We appreciate your motivating comment.

The responses to your two comments were combined into a single reply since they both address the persistence of the Blob.

What sets the time scale for these events? The physics behind the ocean dynamic response to the wind stress curl is essentially the Sverdrup relation, where the gyre-scale ocean dynamics favor the low-frequency variation in the atmosphere. This is supported by the fact that the interannual changes in the gyre boundary are more significant than its seasonal variations (Fig. 1c).

Di Lorenzo and Mantua (2016) noted in their Abstract, “... *teleconnections between the North Pacific and the weak 2014/2015 El Niño linked the atmospheric forcing patterns of this event. These teleconnection dynamics from the extratropics to the tropics during winter 2013/14, and then back to the extratropics during winter 2014/15, are a key source of multi-year persistence of the North Pacific atmosphere.*” In their Fig. 4, the onset of the Blob is explained by the (atmospheric) North Pacific Oscillation. However, from the perspective of the Sverdrup relation, changes driven by the NPO should induce either equally strong or equally weak variations in the subtropical and subarctic gyre, rather than be compensatory in their relationship. We conclude, therefore, that the NPO cannot be the dominant physical driver.

Instead, the question becomes which atmospheric pattern drives the local atmospheric variation? We analyzed the lag-correlation between the sea level pressure anomaly in the area where the wind stress curl anomaly has the highest correlation with the gyre boundary latitude and at each grid point (see Fig. 7) to identify the physical driver of the Blob. The results indicate that the Tropical Northern Hemisphere pattern (as compared in Fig. 7h and the figure found at https://www.cpc.ncep.noaa.gov/data/teledoc/tnh_map.shtml) is likely the driving force behind the

Blob. Interestingly, our results align with the conclusions of Liang et al. (2017), who suggested that the Tropical Northern Hemisphere Pattern is closely linked to the formation of the Blob based on their statistical analysis.

Moreover, Mo and Livezey (1986) noted in Section 6 E, “*The TNH pattern is principally connected to interannual and longer time scales.*” This offers additional support to our interpretation.

Overall, the persistence of the Blob can be attributed to the ocean response to atmospheric conditions at the interannual timescale, with local interannual variations in the atmosphere corresponding to the Tropical Northern Hemisphere Pattern (TNH), and the TNH features with interannual variation. We added this explanation in our revised manuscript.

References

Liang, Y.-C., Yu, J.-Y., Saltzman, E. S. & Wang, F. Linking the tropical Northern hemisphere pattern to the Pacific warm blob and Atlantic cold blob. *J. Clim.* **30**, 9041–9057 (2017).

Mo, K. C. & Livezey, R. E. Tropical-Extratropical Geopotential Height Teleconnections during the Northern Hemisphere Winter. *Monthly Weather Review* **114**, 2488–2515 (1986).

324-328: Why would a ML heat budget fail to capture warm water advection into the “Blob” region? Are the boxes drawn with boundaries that lie outside that of the warm water advection process?

Thank you for your comment. We have now hopefully better explained this in the Methods section of our revised manuscript.

We analyzed the complete form of the classic mixed layer heat budget equation to show why this equation cannot fully capture the advection process.

The complete equation of the classic mixed layer heat budget is:

$$\frac{\partial MLTA}{\partial t} = \frac{Q_{net} - Q_h}{\rho C_p H} - \bar{\mathbf{u}} \cdot \nabla \bar{T} + Mixing - \left(\frac{\bar{T} - T_{-H}}{H} \right) \left(\frac{\partial H}{\partial t} + \mathbf{u}_{-H} \cdot \nabla H + w_{-H} \right)$$

The second term on the right-hand side represents advection, while the fourth term represents entrainment. If we take the entrainment term apart, we can get,

$$- \left(\frac{\bar{T} - T_{-H}}{H} \right) \left(\frac{\partial H}{\partial t} + \mathbf{u}_{-H} \cdot \nabla H + w_{-H} \right) = - \frac{\bar{T}}{H} w_{-H} + \frac{T_{-H}}{H} w_{-H} - \left(\frac{\bar{T} - T_{-H}}{H} \right) \left(\frac{\partial H}{\partial t} + \mathbf{u}_{-H} \cdot \nabla H \right)$$

According to the continuity equation, the vertical velocity, w_{-H} can be written as $H(\nabla \cdot \mathbf{u})$,

which suggests the $-\frac{\bar{T}}{H}w_{-H}$ term can be written as $-\bar{T}\nabla \cdot \mathbf{u}$. This term represents the temperature tendency induced by the divergence or convergence of the horizontal velocity.

Moreover, the entrainment term in the classic mixed layer heat budget is in the form of the effective flux, whereby the temperature difference $(\bar{T} - T_{-H})$ multiplies vertical velocity (the term of $\frac{\partial H}{\partial t} + \mathbf{u}_{-H} \cdot \nabla H + w_{-H}$ has a unit of m/s). Given that the temporal variation of the mixed layer depth $(\frac{\partial H}{\partial t})$ may also be affected by the vertical motion, the entrainment term likely contains more contribution from the lateral advection. Therefore, the advection term in this equation may underestimate the contribution of the advection process.

Another simple way to understand this issue is, when we applied the classic mixed layer heat budget equation to a box (the bottom of the box is flat and does not vary with time), the $\frac{\partial H}{\partial t}$ and ∇H terms vanished. By moving the net heat flux term to the left-hand side, we get:

$$\frac{\partial MLTA}{\partial t} - \frac{Q_{net} - Q_h}{\rho C_p H} = -\bar{\mathbf{u}} \cdot \nabla \bar{T} - \left(\frac{\bar{T} - T_{-H}}{H}\right)w_{-H} + Mixing$$

Expanding the second term in the right-hand side and rearranging, we have:

$$\frac{\partial MLTA}{\partial t} - \frac{Q_{net} - Q_h}{\rho C_p H} = -\bar{\mathbf{u}} \cdot \nabla \bar{T} - \frac{\bar{T}}{H}w_{-H} + \frac{T_{-H}}{H}w_{-H} + Mixing$$

where $-\bar{\mathbf{u}} \cdot \nabla \bar{T} - \frac{\bar{T}}{H}w_{-H}$ is the lateral advection and $\frac{T_{-H}}{H}w_{-H}$ represents the vertical heat advection.

A simple scale analysis could help to evaluate the relative importance of each term. In the Blob region, the average horizontal velocity at the sea surface is 0.05 m/s, the maximum horizontal temperature gradient is 9×10^{-6} K/m ($1 \text{ K} / 110 \text{ km} \times 0.001 \text{ km/m}$), the vertical velocity in the upper ocean is 10^{-4} m/s, the annual mean mixed layer depth is 50 m, SST = 12 K, $T_{-H} = 12 \text{ K} - 0.3 \text{ K} = 11.7 \text{ K}$ (0.3 K is the common criteria for defining the mixed layer depth), and suppose $\bar{T} = \frac{SST + T_{-H}}{2} = 11.85^\circ \text{C}$. On the right-hand side of the above equation, the first term $(\bar{\mathbf{u}} \cdot \nabla \bar{T})$ is on the order of 4.5×10^{-7} K/s, the second term $(\frac{\bar{T}}{H}w_{-H})$ is on the order of 2.37×10^{-5} K/s, and the third term $(\frac{T_{-H}}{H}w_{-H})$ is on the order of 2.34×10^{-5} K/s.

Therefore, both the lateral advection $(\frac{\bar{T}}{H}w_{-H})$ and the vertical advection at the bottom of the mixed

layer ($\frac{T-H}{H} w_{-H}$) plays an important role in the heat budget equation. However, if we write them in the form of the effective flux ($-\frac{\bar{T}}{H} w_{-H} + \frac{T-H}{H} w_{-H}$, temperature difference multiplying vertical velocity), the two terms cancel out and generate a small term that is on the order of $-3 \cdot 10^{-7} \text{ W/s}$ ($-2.37 \cdot 10^{-5} \text{ W/s} + 2.34 \cdot 10^{-5} \text{ W/s}$), which is comparable to the term of $\bar{\mathbf{u}} \cdot \nabla \bar{T}$. This analysis suggests that if the predominance of a particular term is assessed based on its magnitude, both the vertical and lateral processes may be undervalued when the entrainment term is represented as the effective flux.

Wouldn't frontal shift-induced SSTA be maximized along the axis of the SST front, and zonally confined to the SST front?

The variations in SSTA closely align with the ocean circulation (as shown in the plan view: Fig. 2g and h, and sectional view: Fig. 5d with color shading and black line). The gyre boundary displacements serve as a crucial indicator of changes in ocean circulation. This boundary aligns with the SST front, meaning it intersects the area of maximum SSTA (Fig. 2h). However, both the sea level and geostrophic stream function suggest that the changes are not limited to the frontal zone; rather, they extend spatially from 30 to 60 degrees North. Consequently, the SSTA caused by the shift in the front is not restricted to the SST front in a zonal manner.

342-343: Is the 30 deg latitude extent of the SSTA warm pattern consistent with the 200km (2 deg latitude) gyre boundary excursion?

Thank you for reminding us. We examined the thermal displacement and found it increased with latitude (Fig. R1-4). Therefore, we revised this statement to "In this case, the northward thermal displacement caused by the gyre boundary excursion increases with latitude from less than 100 kilometres at about 35 °N and over 200 kilometres at about 52 °N."

Figure R1-6: Comparison of isotherms of SST during the positive phase (black line) and negative phase (blue line).

What about the scale of atmospheric forcing anomalies associated with surface heat flux anomalies?

Focusing on yearly changes in the surface heat flux anomaly (Fig. R1-7), in 2013, the spatial distribution of the annual average surface heat flux anomaly was concentrated in the northeast Pacific region, whereas in other years, the spatial pattern was less uniform.

Figure R1-7: The surface heat flux anomaly referring to an annual mean over 1993 to 2022. A positive value indicates that the ocean is absorbing heat.

Reviewer #2 (Remarks to the Author):

This study tried to explore the potential role of gyre scale ocean dynamics in iconic northeast Pacific marine heatwaves, focusing on the interannual or low-frequency timescales. I think this viewpoint is very important and potentially crucial for understanding the climate variability in the Northeast Pacific. But I think the current presentation is quite preliminary in many aspects. Please see my detailed comments below.

We appreciate your acknowledgment of the importance of this work. Below, you will find our detailed responses to your comments, with your comments in black font and our replies in blue font.

Due to significant changes in the manuscript compared to the previous submission, we want to summarize the main points in the revision. The “Introduction” has been completely rewritten to provide a more comprehensive explanation of our concerns about the conventional mixed layer heat budget analysis. The section headed 'Mixed layer heat budget' has been restructured to follow the 'Gyre boundary variation in the 'Blob' region', highlighting the importance of advection in the heat budget. Furthermore, we have added two new sections: one examining the variations in net heat flux and the other addressing the physical drivers of the Blobs. The final section has also been revised to incorporate our new findings.

Major comments:

In your mixed-layer heat budget equation, the H is time varying mixed layer depth. Please specify how you calculate the surface heat flux term, that is $Q_{net}/(\rho C_p H)$. In particular, how does the mixed layer depth vary in this term? Moreover, I cannot tell exactly how do the gyre variations work on the MLTA in the blob region, because authors merged the oceanic processes into one term. The physical picture is not clear to me. Moreover, I cannot tell exactly how do the gyre variations work on the MLTA in the blob region, because authors merged the oceanic processes into one term.

Thank you for your comment. In our revision, the surface heat flux term is calculated according to $\frac{Q_{net}-Q_h}{\rho C_p H}$, where Q_{net} is the surface net heat flux, calculated by the sum of the latent and sensible heat flux, the longwave and short wave radiation, Q_h is the penetrative shortwave radiation, H is the time-varying mixed layer depth, and other factors are calculated as follows, $\rho C_p = 4.088 \times$

$10^6 \text{ J}^\circ\text{C}^{-1}\text{m}^{-3}$ and $Q_h = Q_{sw}(0.58e^{-\frac{h}{0.35}} + 0.42e^{-\frac{h}{23}})$ (Shi et al., 2022). All the data are daily resolved.

Since the relationship between mixed layer depth and net surface heat flux is not our main focus and has been thoroughly addressed by Shi et al. (2022), and cited in our manuscript, we did not examine its variations. The importance of advection is highlighted by its significant role in the positive Mixed Layer Temperature Anomaly (MLTA) during the onset of the Blobs, as shown in Fig. 3c. We employed a box model to analyze the temperature tendency. The rationale for not utilizing the traditional mixed layer heat budget is explained in response to the next comment. The lateral advection term in our box model effectively captures the contributions from the oceanic advection process (Fig. 3d). This advection corresponds to the Sverdrup transport, which is fundamental to the wind-driven ocean general circulation. Fig. 4 demonstrates that slow variations in the large-scale ocean circulation can be explained by this theory, while Fig. 5 supports this by analyzing the geostrophic stream function in a sectional view. This reason was part of our previous submission, although the figures mentioned are arranged differently in the revision.

Additionally, Fig. 3c indicates that the net heat flux may contribute to the decay of the Blob. Consequently, we investigated the surface heat flux and found that the latent heat flux is the most significant factor (Fig. 6a). Further analysis revealed that the unusual wind direction affects the 2-meter air humidity, which in turn influences the latent heat flux. The persistence of the Blob is briefly discussed through the lag-correlation between sea level pressure anomalies in the region where the wind stress curl anomaly shows the largest correlation coefficient with the gyre boundary latitude and at each geographical point. The result suggests the persistence can be attributed to a Tropical Northern Hemisphere Pattern-like teleconnection pattern (Fig. 7).

References

Shi, J. et al. Role of mixed layer depth in the location and development of the Northeast Pacific warm blobs. *Geophys. Res. Lett.* 49, e2022GL098849 (2022).

In many previous studies, as you also shown, the oceanic processes may not be that important for the warm blobs. How to interpret your results?

Thank you for your comment. In previous studies, oceanic process contributions were found to be

negligible, with conclusions derived from diagnostic analysis of the mixed layer heat budget. However, we find that the classic mixed layer heat budget misleads these interpretations.

In the present study, we analyzed the more complete mixed layer heat budget equation, to show why the advection term in this equation cannot fully capture the advection process.

The complete equation of the classic mixed layer heat budget is:

$$\frac{\partial MLTA}{\partial t} = \frac{Q_{net} - Q_h}{\rho C_p H} - \bar{\mathbf{u}} \cdot \nabla \bar{T} + \text{Mixing} - \left(\frac{\bar{T} - T_{-H}}{H} \right) \left(\frac{\partial H}{\partial t} + \mathbf{u}_{-H} \cdot \nabla H + w_{-H} \right)$$

The second term on the right-hand side represents advection, while the fourth term represents entrainment. If we take the entrainment term apart, we get

$$- \left(\frac{\bar{T} - T_{-H}}{H} \right) \left(\frac{\partial H}{\partial t} + \mathbf{u}_{-H} \cdot \nabla H + w_{-H} \right) = - \frac{\bar{T}}{H} w_{-H} + \frac{T_{-H}}{H} w_{-H} - \left(\frac{\bar{T} - T_{-H}}{H} \right) \left(\frac{\partial H}{\partial t} + \mathbf{u}_{-H} \cdot \nabla H \right)$$

According to the continuity equation, the vertical velocity, w_{-H} can be written as $H(\nabla \cdot \mathbf{u})$, which suggests, the $-\frac{\bar{T}}{H} w_{-H}$ term can be written as $-\bar{T} \nabla \cdot \mathbf{u}$. This term represents the temperature tendency induced by the divergence or convergence of the horizontal velocity. Moreover, the $\frac{T_{-H}}{H} w_{-H}$ term represents the net vertical heat advection through the bottom of the mixed layer.

The entrainment term in the classic mixed layer heat budget is in the form of the effective flux, whereby the temperature difference $(\bar{T} - T_{-H})$ multiplies vertical velocity (the term of $\frac{\partial H}{\partial t} + \mathbf{u}_{-H} \cdot \nabla H + w_{-H}$ has a unit of m/s). Given that the temporal variation of the mixed layer depth $\left(\frac{\partial H}{\partial t}\right)$ may also be affected by the vertical motion, the entrainment term likely contains more contribution from the lateral advection. Therefore, the advection term in this equation may underestimate the contribution of the advection process.

Another simple way to understand this is when we applied the classic mixed layer heat budget equation to a box (the bottom of the box is not time-varying and is flat), the $\frac{\partial H}{\partial t}$ and ∇H terms vanished. By moving the net heat flux term to the left-hand side, we get:

$$\frac{\partial MLTA}{\partial t} - \frac{Q_{net} - Q_h}{\rho C_p H} = -\bar{\mathbf{u}} \cdot \nabla \bar{T} - \left(\frac{\bar{T} - T_{-H}}{H} \right) w_{-H} + \text{Mixing}$$

Expanding the second term on the right-hand side and rearranging, we have:

$$\frac{\partial MLTA}{\partial t} - \frac{Q_{net} - Q_h}{\rho C_p H} = -\bar{\mathbf{u}} \cdot \nabla \bar{T} - \frac{\bar{T}}{H} w_{-H} + \frac{T_{-H}}{H} w_{-H} + \text{Mixing}$$

where $-\bar{\mathbf{u}} \cdot \nabla \bar{T} - \frac{\bar{T}}{H} w_{-H}$ is the lateral advection and $\frac{T_{-H}}{H} w_{-H}$ represents the vertical heat advection.

A simple scale analysis could help to evaluate the relative importance of each term. In the Blob region, the average horizontal velocity at the sea surface is 0.05 m/s, the maximum horizontal temperature gradient is 9×10^{-6} K/m ($1 \text{ K} / 110 \text{ km} \times 0.001 \text{ km/m}$), the vertical velocity in the upper ocean is 10^{-4} m/s, the annual mean mixed layer depth is 50 m, SST = 12 K, $T_{-H} = 12 \text{ K} - 0.3 \text{ K} = 11.7 \text{ K}$ (0.3 K is the common criteria for defining the mixed layer depth), and suppose $\bar{T} = \frac{SST + T_{-H}}{2} = 11.85^\circ \text{C}$. On the right-hand side of the above equation, the first term ($\bar{\mathbf{u}} \cdot \nabla \bar{T}$) is on the order of 4.5×10^{-7} K/s, the second term ($\frac{\bar{T}}{H} w_{-H}$) is on the order of 2.37×10^{-5} K/s, and the third term ($\frac{T_{-H}}{H} w_{-H}$) is on the order of 2.34×10^{-5} K/s.

Therefore, both the lateral advection ($\bar{\mathbf{u}} \cdot \nabla \bar{T}$) and the vertical advection at the bottom of the mixed layer ($\frac{T_{-H}}{H} w_{-H}$) plays an important role in the heat budget equation. However, if we write them in the form of the effective flux ($-\frac{\bar{T}}{H} w_{-H} + \frac{T_{-H}}{H} w_{-H}$, temperature difference multiplying vertical velocity), the two terms cancel out and generate a small term that is on the order of -3×10^{-7} K/s (-2.37×10^{-5} K/s + 2.34×10^{-5} K/s), which is comparable to the term of $\bar{\mathbf{u}} \cdot \nabla \bar{T}$. This analysis suggests that if the predominance of a particular term is assessed based on its magnitude, both the vertical and lateral processes may be undervalued when the entrainment term is represented as the effective flux.

The mixed layer heat budget was not filtered to extract the most influential SSTA component in previous study. Therefore, we assessed the heat budget within the frequency band that primarily influences the positive SSTA. Despite the classic mixed layer heat budget facing challenges in breaking down the oceanic term, we consider its most basic form to be trustworthy. This indicates that the mixed layer temperature anomaly (MLTA), the surface heat flux term, and the oceanic term—defined as the difference between the two—are all reliable. Our analysis reveals that during the Blob period, the oceanic term is positive, whereas the surface heat flux term is negative (the direction of the flux is from ocean to atmosphere). Consequently, we conclude that the oceanic advection plays a key role in the formation of the Blob, while the surface heat flux is more important for its decay.

In this study, authors did not pay special attention to the differences of mechanisms between seasons.

Thank you for your comment. This is not our primary concern. Our analysis focuses on the interannual timescale, and hence here seasonal variations are removed.

What is the frontal dynamics? I think authors did not use too much frontal theory in this study.

Thank you for your comment. The process responsible for the northward movement of the front is based on classic wind-driven ocean circulation theory provided by the Sverdrup relation. The curl of the wind stress causes downward Ekman pumping and stretching of the water column. This results in a northward movement of the water column to maintain potential vorticity conservation ($D((f+\zeta)/H)/Dt=0$, where f and ζ represent the planetary and relative vorticity, and H denotes the thickness of the water column). Based on the premise of geostrophic motion, the Rossby number $\ll 1$, or $\zeta \ll f$, the potential vorticity conservation represents the balance between planetary vorticity (f) and the layer thickness (H). More specifically, it is the shifts in the temperature front's position that are significant, rather than the front itself (for example, its intensity). Thus, it is not necessary to apply frontal theory in this context.

Figure 2: I think there should be a more objective method to select northernmost and southernmost years. Then, significance test should be added to confirm the robustness of your results. Currently, the sample numbers are small. Moreover, authors can add the climatological +0.47 line to show the changes. Same issue for Fig. 3.

Thank you for your comment. The composite map provides a readily comprehensible overview of the changes in ocean circulation and wind patterns. A significance test (p -value) was performed to show the statistical significance of the correlation coefficient between the gyre boundary and SSTA (Fig. 2i), as well as between the gyre boundary and wind stress curl (Fig. 7a). Therefore, our conclusions are not influenced by the selection of the northernmost and southernmost years.

Since the composite map is based on absolute values rather than anomalies, the years chosen for the composite were selected based on specific criteria: first, we used data from entire years for averaging to avoid seasonal cycle aliasing; second, we noted a linear trend indicating a northward shift of the gyre boundary, so we aimed to select years that are evenly spread across each decade to prevent the composite map from reflecting a long-term trend; third, we created anomalies for 2014 and 2019

and compared them with the positive phase to ensure that the spatial patterns were similar.

We included the +0.47 contour line in each panel of Figure 2, and replaced the 0.47 contour line with the zero contour line of the stream function derived from the wind stress curl.

Currently, I cannot understand well the texts in Line 135-137.

We appreciate you mention it. The description in lines 135-137 is problematic. Reviewer #1 also noted this. We have now revised the text to “During the positive phase, the subtropical gyre located east of 175 °W expands northward, leading to a southwest-northeast oriented tilt of the gyre boundary (Fig. 2a). In contrast, during the negative phase, the gyre boundary shifts southward and assumes an almost zonal orientation (Fig. 2d).”

Line 215-218: Previous studies claimed the shallower mixed layer depth during the Northeast Pacific warm blobs/marine heatwaves (Amaya et al. 2020 NC; Shi et al. 2022 GRL). Could authors give more explanations on the deepening of thermocline here.

Thank you for your comment. In practical terms, if the subtropical gyre expands northward and encroaches upon the area occupied by the subarctic gyre, the existing characteristics of low sea level, a shallower thermocline, and cooler sea surface temperatures, will be replaced by those of the subtropical gyre, which include higher sea levels, a deeper thermocline, and warmer SST.

From a theoretical standpoint, the principle of potential vorticity (PV) conservation explains why the northward movement of the water column is typically linked to a deepening of the main thermocline. The equation representing PV conservation is $D((f+\zeta)/H)/Dt=0$, where f is the Coriolis parameter and H is the water column thickness. Based on the premise of geostrophic motion, the Rossby number $\ll 1$, or $\zeta \ll f$, the potential vorticity conservation represents the balance between planetary vorticity (f) and the layer thickness (H). As the water column moves north, it enters a higher latitude region where f increases. To maintain PV conservation, the water column must also be stretched, leading to the increase of H . Conversely, as the water column stretches first (H increases), it also moves northward (f increases) subsequently to satisfy this relationship. In this case, the wind stress curl causes Ekman pumping, which stretches the water column before it moves northward, contributing to the Sverdrup balance.

The main thermocline acts as the boundary between the upper ocean and the deep ocean. The

stretching of the water column in the upper ocean causes the main thermocline to descend, which is the dynamical response outlined here.

The neutral point appears a bit arbitrary. Authors tried to raise a new method to identify the onset of the blob, but they should compare in details with the previous results. Why this method is more reasonable than others?

Thank you for your comment. In the revision, by carefully examining the 365-day low-passed SSTA, the neutral point for Blob 1.0 is identified as March 19, 2013, while the neutral point for Blob 2.0 is determined to be May 6, 2018. Revisions have been made in accordance with these findings.

Our analysis of the emergence of Blob 1.0 seeks to address the question posed by Amaya et al. (2020) regarding the differences in timing between Blob 1.0 and Blob 2.0. They noted, "*An important distinction between the recent temperature anomalies (referred to as Blob 2.0) and those from 2013/2014 (referred to as Blob 1.0) is that Blob 2.0 primarily intensified during the summer months. In contrast, Blob 1.0 developed in the winter due to a persistent atmospheric ridge in the Northeast Pacific, which weakened the climatological Aleutian Low and associated surface winds.*"

While defining the onset as October is reasonable from an event perspective, it leads to entirely different conclusions when analyzing mechanisms. A comparison of Fig. 2b and 2c reveals that the classic mixed layer heat budget, after applying a 90-day low-pass filter, indicates that both ocean processes and surface net heat flux contribute to warming at the time marked by the black double-headed arrow. In contrast, the results from a 365-day low-pass filter show that during the Blobs, ocean processes consistently have a positive contribution, while surface net heat flux consistently has a negative contribution (the direction of the flux is from ocean to atmosphere). This raises the question of which explanation to trust.

The primary fluctuations in SSTA in the Blob region occur at the interannual timescale, suggesting that understanding the positive phase of the interannual component of SSTA can explain most of the positive anomalies. Therefore, we argue that focusing on the interannual timescale, particularly by analyzing SSTA after a 365-day low-pass filter, is more suitable.

Moreover, did Amaya et al. (2020) propose the onset of blob 2.0 in May 2018? I think they only focused on the summer of 2019, rather than 2018.

Amaya et al. (2020) proposed the Blob 2.0 was intensified in May 2019, and did not analyze (or determine) the onset of Blob 2.0. We have corrected the text.

Minor comments:

The word “Iconic” is a bit hard to clearly understand. Could authors change into another word?

Thank you for your comment. We deleted this word.

Abstract, Line 43: anomalous positive or negative wind stress curl drives the expansion of the subtropical gyre. Please specify here.

Thank you for your comment. We revised this sentence to “*The northward expansion (or contraction) of the subtropical (subarctic) gyre is influenced by anomalous wind stress curl and can be understood through wind-driven ocean circulation theory, specifically the Sverdrup relation.*”

Blob 1.0 and 2.0: It is not appropriate to appear in the abstract without any background here. Readers may not know their similarities and differences.

Thank you for your comment. We deleted this sentence.

Abstract, Line 52: It is better to specify the sea surface temperature, rather than temperature.

Thank you for your comment. This sentence is deleted in the revision.

Line 61: Chen et al. (2021) classified two types of warm blobs and reported different mechanisms of the warm blobs in different seasons.

Chen, Z., Shi, J., & Li, C. (2021). Two types of warm blobs in the Northeast Pacific and their potential effect on the El Niño. *International Journal of Climatology*, 41(4), 2810–2827.

Thank you for your comment. We cited this paper in the Line 58 in the revision.

Line 64: For Blob 1.0, due to its multi-year persistence over 2013-2015, the growth months may be different in these years.

Thank you for your comment. We removed "growth" since the two time points are centered on the onset of the Blobs.

There are some errors in the reference list, such as the missing page information, etc.

Thank you for your comment. We have now carefully checked and revised this list.

Line 85-86: Why the advection is zeros here? Could you please draw a simple diagram to illustrate? Moreover, why do you give this ideal case? I feel confused to understand. How does advection outside the computational domain explain the temperature tendency in the domain?

Thank you for your comment. We have now deleted this ideal case as it leads to misunderstanding.

Here we explain the idea of advection that occurs outside the computational domain, a concept first introduced by Lee et al. (2004) and termed "external advection." They incorporated the advection term into the classic mixed layer heat budget, where the dot product of the temperature gradient and horizontal velocity signifies heat redistribution within the computational domain (referred to as internal advection). However, internal advection alone does not explain the overall movement of a water parcel within the computational domain (control volume).

This issue arises from the distinction between the Eulerian and Lagrangian perspectives (Figure R2-1). From the Eulerian viewpoint, the computational domain used to calculate the net heat flux term remains fixed, while the water column beneath it is in motion. From the Lagrangian viewpoint, the water column within the specified area (shown by the black parallelogram, A1) moves from one time point (T1) to another (T2) due to advection, arriving at the position indicated by the red dashed parallelogram (A2).

We will assume that there is a transient net heat flux impacting the time point from T1 to T2. This net heat flux leads to temperature changes between the area represented by the black parallelogram at T1A1 and the area depicted by the red dashed parallelogram at T2A2. In the budget calculation, the temperature change is evaluated based on the temperatures in the black parallelogram at both T1 and T2, i.e., T1A1 and T2A1. The temperature difference utilized to calculate the temperature tendency should accurately be represented as T2A2 - T1A1; however, the value that is computed is T2A1 - T1A1.

Clearly, the results obtained from the two methods differ. This discrepancy results from the overall movement of the water parcel within the computational domain, and is termed external advection. Additionally, internal advection refers to the movement of a single particle from one point in the computational domain to another.

Next, we can evaluate how much this external advection influences the results. Typically, the classic mixed layer heat budget is analyzed on a monthly basis. In the Blob region, the average horizontal velocity at the sea surface is 0.05 m/s, which translates to a movement of 130 km over the course of a month ($0.05 \text{ m/s} * 0.001 \text{ km/m} * 86400 \text{ s/day} * 30 \text{ days/month}$). This suggests that nearly 10% of the water column is displaced beyond the computational domain, and this effect is not accounted for in the classic mixed layer heat budget.

Figure R2-1: Schematic view of the external advection. The black parallelogram (A1) indicates the computational domain and the red dashed parallelogram (A2) indicates the position where the water column arrived at.

Line 98-99: Please cite some related papers to confirm this.

Thank you for your comment. We cited the following papers.

Fig. 2c in *Di Lorenzo, E. & Mantua, N. Multi-year persistence of the 2014/15 North Pacific marine heatwave. Nat. Clim. Chang 6, 1042–1047 (2016).*

Fig. 1b in *Amaya, D. J., Miller, A. J., Xie, S. P. & Kosaka, Y. Physical drivers of the summer 2019 North Pacific marine heatwave. Nat. Commun. 11, 1903 (2020).*

Fig. 1a in *Chen, Z., Shi, J., & Li, C. (2021). Two types of warm blobs in the Northeast Pacific and their potential effect on the El Niño. International Journal of Climatology, 41(4), 2810–2827.*

Fig. 1a: climatological mean? Annual-mean? Please specify.

Thank you for your comment. It is the annual-mean. We revised the figure caption.

Fig. 1b: Could authors add significance test for this figure? The white shading to the rightmost area indicates there is no +0.47 value, right?

Thank you for your comment. The white shading shows that the +0.47 contour shifts northward, while the coastline is located further to the west. A significance test (p -value) was performed to show the statistical significance of the correlation coefficient between the gyre boundary and SSTA (Fig. 2i), as well as between the gyre boundary and wind stress curl (Fig. 7a).

Fig. 1c: the interannual component should be frequency higher-than 9 years. Why do authors use the 1-year filter here?

Thank you for your comment. The interannual time scale is defined by the component that accounts for most of the variation in SSTA (the standard deviation of this interannual component explains 80% of the standard deviation of the unfiltered SSTA). A 1-year filter is applied to eliminate the effects of higher frequency variations, such as those occurring within a year or season. While using a 2-year filter does not significantly alter our findings, the 1-year filter provides greater clarity regarding the persistence of the Blob event.

Fig. 3: Authors did not make it clear the wind vectors and curl are anomalies or original values.

Thank you for your comment. The wind vectors are anomalies. Also, we revised the figure caption.

Line 178: confusing for “In the mean”

Thank you for your comment. “In the mean” is used to describe the general relationship between the wind stress curl and the two gyres. We revised this word to “in general”.

Line 182 and 184: west of or east of? Moreover, I cannot distinguish the changes without climatological position superimposed. Also for the right column in terms of stream function. Moreover, why these panels are averaged over a narrow band as indicated in Line 193?

Thank you for your comment. The information provided was unfortunately incorrect. We have

now corrected this and revised.

Additionally, all climatological locations of the gyre boundary and stream function have been superimposed in Figure 2 and Figure 5.

Our analysis is not affected by the selection of the band used to calculate the zonal mean stream function. In line with your recommendation, we have now chosen a broader longitude band of 145 °W to 135 °W in the revised manuscript.

Line 203-204: Please give the figures, maybe in the supplement.

Thank you for your comment. We added the zero contour of the wind stress curl derived stream function to Figure 4 in the revised manuscript.

Line 276-277: Confusing sentence for “the increased mixed layer heat loss to the air is the response”.

Thank you for your comment. This sentence has been deleted.

Line 337: what did “residual” refer to here?

Thank you for your comment. The “residual” here is the difference between the total temperature tendency and the sum of the other terms in the equation, i.e., $Residual = \frac{\partial MLTA}{\partial t} - \frac{Q_{net} - Q_h}{\rho C_p H}$.

Line 350-351: What variable for the spatial extent of approximately 30 ° in latitude?

Thank you for your comment. This statement is derived from the details shown in Fig. 5d. The changes in the geostrophic stream function and temperature exhibit comparable spatial patterns, with a spatial range of about 30° in latitude.

Fig. 5: What is the difference between left and right figures in each panel? In panel c, why does the warm anomaly correspond to cyclonic wind anomalies? PV dynamics is not included in the main texts.

Thank you for your comment. The schematic has been deleted.

Line 391-392: I am wondering whether +0.47 is reasonable for the identification of flow axis in each year/month.

Thank you for your comment. Since interannual variations are larger than seasonal variations at the gyre boundary, we examined +0.47 as the flow axis based on yearly data. The black line is positioned between the subtropical and subarctic gyres, suggesting that it is a valid identification of the flow axis.

Figure R2-2: The yearly mean sea level. The black lines represent the +0.47 contour for each year, while the magenta lines show the annual average from 1993 to 2022.

Reviewer #3 (Remarks to the Author):

This study analyzed the causes of the "blobs" that occurred in the Northeastern Pacific through a new perspective on large-scale ocean dynamics. The study argued that the positive-negative wind stress curl anomalies in the Northeastern Pacific drove the formation of the blob by shifting the gyre boundary between the subtropical and subarctic gyres. The study has investigated the cause of blobs through a simple heat budget analysis with respect to the gyre boundary variations. Through the negative wind stress curl anomaly, the subarctic gyre contracted, and the subtropical gyre expanded, leading to oceanic heat gain.

The value of this study lies in its examination of blob formation through the oscillations of subtropical and subarctic gyres, rather than focusing solely on the climate variability perspective of air-sea interactions, which was proposed in previous studies. This new viewpoint could provide insights for future research on marine heatwaves and ecosystems in other regions, making it a relevant contribution to the scope of the journal.

However, there are some limitations in the analysis and interpretation of results, and I believe addressing these will significantly enhance the quality of the paper.

Thank you for your positive review and for pointing out the limitations in the analysis and interpretation of our results. Below, you will find our detailed responses to your comments, with your comments in black font and our replies in blue font.

Due to significant changes in the manuscript compared to the previous submission, we want to summarize the points in our revisions. The "Introduction" was completely rewritten to provide a more comprehensive explanation of our concerns about the conventional mixed layer heat budget analysis. The section titled "Mixed layer heat budget" was restructured to follow the section "Gyre boundary variation in the 'Blob' region", highlighting the importance of advection in the heat budget. Furthermore, we added two new sections: one examining the variations in net heat flux and the other addressing the physical drivers of the Blobs. The final section was also revised to include our new findings.

Here are some major comments:

1. The heat budget analysis is overly simplified. The study divides surface net heat flux and the residual term, attributing the residual term to the ocean's influence. However, the equation presented in the paper assumes that all shortwave radiation enters and exits the mixed layer, while some of it is likely transferred to the subsurface layer beneath the mixed layer. This could result in an overestimation of surface net heat flux and an underestimation of the ocean's influence, which might align with the paper's intent, but accurate calculations and proposals of values would be more appropriate. Additionally, the entrainment term could be helpful to understand the effect of the upwelling & mixed layer depth changes on the blob formation due to the negative wind stress curl anomaly. Relevant references include:

References:

Qiu, C., Kawamura, H., Mao, H., & Wu, J. (2014). Mechanisms of the disappearance of sea surface temperature fronts in the subtropical North Pacific Ocean. *Journal of Geophysical Research: Oceans*, 119(7), 4389-4398.

Murata, K., Kido, S., & Tozuka, T. (2020). Role of reemergence in the central North Pacific revealed by a mixed layer heat budget analysis. *Geophysical Research Letters*, 47(13), e2020GL088194.

Thank you for your comments. First, following your suggestion, we added the penetrative shortwave radiation into the net surface heat flux term, and revised all texts and figures based on this mixed layer heat budget.

Second, instead of decomposing the oceanic terms into the entrainment term and other terms, we used a box model for the entrainment term including the variation of horizontal advection. Additionally, we cited those two papers in our revision.

The details confirming the entrainment term including the variation of advection is as follows. We analyzed the complete form of the classic mixed layer heat budget equation, to show why this equation cannot fully capture the advection process.

The complete equation of the classic mixed layer heat budget is:

$$\frac{\partial MLTA}{\partial t} = \frac{Q_{net} - Q_h}{\rho C_p H} - \bar{\mathbf{u}} \cdot \nabla \bar{T} + Mixing - \left(\frac{\bar{T} - T_{-H}}{H} \right) \left(\frac{\partial H}{\partial t} + \mathbf{u}_{-H} \cdot \nabla H + w_{-H} \right)$$

The second term on the right-hand side represents advection, while the fourth term represents entrainment. If we take the entrainment term apart, we get

$$-\left(\frac{\bar{T} - T_{-H}}{H}\right)\left(\frac{\partial H}{\partial t} + \mathbf{u}_{-H} \cdot \nabla H + w_{-H}\right) = -\frac{\bar{T}}{H}w_{-H} + \frac{T_{-H}}{H}w_{-H} - \left(\frac{\bar{T} - T_{-H}}{H}\right)\left(\frac{\partial H}{\partial t} + \mathbf{u}_{-H} \cdot \nabla H\right)$$

According to the continuity equation, the vertical velocity, w_{-H} can be written as $H(\nabla \cdot \mathbf{u})$, which suggests, the $-\frac{\bar{T}}{H}w_{-H}$ term can be written as $-\bar{T}\nabla \cdot \mathbf{u}$. This term represents the temperature tendency induced by the divergence or convergence of the horizontal velocity. Moreover, the $\frac{T_{-H}}{H}w_{-H}$ term represents the net vertical heat advection through the bottom of the mixed layer.

The entrainment term in the classic mixed layer heat budget is in the form of the effective flux, whereby the temperature difference ($\bar{T} - T_{-H}$) multiplies vertical velocity (the term of $\frac{\partial H}{\partial t} + \mathbf{u}_{-H} \cdot \nabla H + w_{-H}$ has a unit of m/s). Given that the temporal variation of the mixed layer depth ($\frac{\partial H}{\partial t}$) may also be affected by the vertical motion, the entrainment term likely contains more contribution from the lateral advection. Therefore, the advection term in this equation may underestimate the contribution of the advection process.

Another simple way to understand this issue is when we applied the classic mixed layer heat budget equation to a box (the bottom of the box is flat without temporal change), the $\frac{\partial H}{\partial t}$ and ∇H terms vanished. By moving the net heat flux term to the left-hand side, we get:

$$\frac{\partial MLTA}{\partial t} - \frac{Q_{net} - Q_h}{\rho C_p H} = -\bar{\mathbf{u}} \cdot \nabla \bar{T} - \left(\frac{\bar{T} - T_{-H}}{H}\right)w_{-H} + Mixing$$

Expanding the second term on the right-hand side and rearranging, we have:

$$\frac{\partial MLTA}{\partial t} - \frac{Q_{net} - Q_h}{\rho C_p H} = -\bar{\mathbf{u}} \cdot \nabla \bar{T} - \frac{\bar{T}}{H}w_{-H} + \frac{T_{-H}}{H}w_{-H} + Mixing$$

where $-\bar{\mathbf{u}} \cdot \nabla \bar{T} - \frac{\bar{T}}{H}w_{-H}$ is the lateral advection and $\frac{T_{-H}}{H}w_{-H}$ represents the vertical heat advection.

A simple scale analysis could help to evaluate the relative importance of each term. In the Blob region, the average horizontal velocity at the sea surface is 0.05 m/s, the maximum horizontal temperature gradient is $9 \times 10^{-6} \text{ }^\circ\text{C}/\text{m}$ ($1^\circ\text{C}/110\text{km} \times 0.001 \text{ km}/\text{m}$), the vertical velocity in the upper ocean is 10^{-4} m/s, the annual mean mixed layer depth is 50 m, SST = 12 $^\circ\text{C}$, $T_{-H} = 12^\circ\text{C} - 0.3^\circ\text{C} = 11.7^\circ\text{C}$ (0.3 $^\circ\text{C}$ is the common criteria for defining the mixed layer depth), and suppose $\bar{T} = \frac{SST + T_{-H}}{2} = 11.85^\circ\text{C}$.

On the right-hand side of the above equation, the first term ($\bar{\mathbf{u}} \cdot \nabla \bar{T}$) is on the order of $4.5 \cdot 10^{-7} \text{ W/s}$, the second term ($\frac{\bar{T}}{H} w_{-H}$) is on the order of $2.37 \cdot 10^{-5} \text{ W/s}$, and the third term ($\frac{T-H}{H} w_{-H}$) is on the order of $2.34 \cdot 10^{-5} \text{ W/s}$.

Therefore, both the lateral advection ($\frac{\bar{T}}{H} w_{-H}$) and the vertical advection at the bottom of the mixed layer ($\frac{T-H}{H} w_{-H}$) play an important role in the heat budget equation. However, if we write them in the form of the effective flux ($-\frac{\bar{T}}{H} w_{-H} + \frac{T-H}{H} w_{-H}$, temperature difference multiplying vertical velocity), the two terms cancel out and generate a small term that is on the order of $-3 \cdot 10^{-7} \text{ W/s}$ ($-2.37 \cdot 10^{-5} \text{ W/s} + 2.34 \cdot 10^{-5} \text{ W/s}$), which is comparable to the term of $\bar{\mathbf{u}} \cdot \nabla \bar{T}$. This analysis suggests that if the predominance of a particular term is assessed based on its magnitude, both the vertical and lateral processes may be undervalued when the entrainment term is represented as the effective flux.

2. The positive-negative wind stress curl anomalies weaken (strengthen) the subarctic (subtropical) gyre. According to the Ekman theory, this implies a reduction (intensification) in divergence (convergence) within the surface mixed layer, leading to heat accumulation at the surface and a weakening (strengthening) of upwelling (downwelling), reducing heat loss. Therefore, in the gyre's interior region, the blob formation might be more due to the weakening (strengthening) of the subarctic (subtropical) gyres affecting mixed layer depth and upwelling, (downwelling) rather than advection of the gyre boundary especially. The analysis in the paper does not sufficiently support the idea that heat is transferred from the subtropical gyre to the subarctic gyre. Additional analysis showing the spatiotemporal transmission of heat from the subtropical gyre across the front to the subarctic gyre is needed. This is the reason for that the entrainment term is included in the heat budget analysis.

Additionally, the schematic diagram in Figure 5 suggests that warm water columns move northward via potential vorticity (PV) conservation from the subtropical gyre to the subarctic gyre across the front. However, the negative wind stress curl occurs in the subarctic gyre, not the subtropical gyre, so the figure does not align with the content of the paper. In PV conservation, an increase in H does not necessarily lead to northward movement to higher latitudes but could instead increase relative vorticity. And it is also questionable whether the water column would be able to cross the front and

move northward to conserve PV. Therefore, I believe Figure 5 needs significant revision, and if the content of Figure 5 is correct, sufficient discussion should be included in the manuscript.

Thank you for your comment. We agree now that Figure 5 is unclear, and have removed it.

The Sverdrup relation clarifies how Ekman pumping is connected to the horizontal movement of the water column. It is based on the premise of geostrophic motion (where the Rossby number $\ll 1$, or specifically, $\zeta \ll f$) supporting conservation of the balance between planetary vorticity (f) and the layer thickness (H), and ignoring relative vorticity (ζ) which is deemed very small relative to f . Hence, the northward shift of the gyre boundary, or the upward movement of the water column, is due to the Sverdrup transport rather than Ekman transport.

3. The positive phase period should include not only the year the blob formed but also the years when the blob existed. Therefore, the years 2015 and 2020 should also be included. Additionally, does the negative wind stress curl anomaly occur in both 2015 & 2020 and does the marine heatwaves appear in 2005?

Thank you for your comment. The year selected for calculating the composite map during the positive phase is determined by the peak latitude of the gyre boundary rather than the SSTA. However, because of the long-term trend of the gyre boundary moving northward, it is important not to include too many recent years, as this would skew the average position of the gyre boundary further north.

As shown in Fig. 7b of our revised manuscript, a negative wind stress curl anomaly was observed in both 2015 and 2020.

Although there was a peak in SSTA in 2005, it was not classified as a marine heatwave event. Moreover, we can find a northward excursion of the gyre boundary in 2005.

Figure R3-1: The gyre boundary in each year. The thick black line denotes the mean value over the period from 1993-2022.

4. In this paper, it is argued that Blob 1.0 started in May 2013, based on 365-day lowpass filtered data. Marine heatwaves are phenomena typically observed on a daily scale, so it is questionable whether identifying them using low-pass filtered data is appropriate.

Thank you for your comment. In the revision, by carefully examining the 365-day low-passed SSTA, the neutral point for Blob 1.0 is identified as March 19, 2013, while the neutral point for Blob 2.0 is determined to be May 6, 2018. Revisions have been made in accordance with these recent findings.

Marine heatwaves at the sea surface are defined as SSTA exceeding the threshold for more than 5 consecutive days (Hobday et al., 2016). In the Blob region, however, the interannual variation in SSTA (blue line in Fig. R3-2) alone exceeded the threshold (green line), indicating that explaining the SSTA extreme can be simplified as analyzing the interannual variation in SSTA, and explaining the persistence of the Blob equals understanding why interannual variation is the dominant component in SSTA. Therefore, although the marine heatwaves are defined on a daily scale, the low-pass filtered data were used in this study.

Second, as we noted in the Introduction section of our manuscript, “The mixed layer heat budget should be conducted over an appropriate frequency band or timescale that significantly influences extreme positive SSTA. However, this aspect has been largely overlooked in current research. Typically, atmospheric processes with durations exceeding 30 days are categorized as low-frequency variations, which are shorter than the mesoscale processes observed in the ocean. Therefore, the high-frequency components of SSTA are associated with a high temperature tendency, aligning with the characteristic time scale of atmospheric processes, while the low-frequency components correspond to a low temperature tendency and the characteristic time scale of oceanic processes. Consequently, conducting an analysis using unfiltered SSTA without accounting for the timescales that predominantly contribute to positive SSTA may bias towards higher values, which coincidentally correspond to the atmospheric dynamics.” Applying the low-pass filter to the data excludes the interference of the high-frequency variation in the SSTA that associated with high temperature tendency.

Figure R3-2: Areal mean SSTA over the Blob region (150 °W – 135 °W, 40 °N – 50 °N, gray shading), the 365-day low-pass filtered SSTA (blue line), and the threshold (green line) that followed Hobday et al. (2016) definition. The red shading indicates the occurrence of the Blob.

References

Hobday, A. J. et al. A hierarchical approach to defining marine heatwaves. *Prog. Oceanogr.* **141**, 227–238 (2016).

Additionally, an explanation is needed as to why a cold anomaly lasting 40 days was included in the Blob period, and what advantages this inclusion brings.

Our analysis of the emergence of Blob 1.0 seeks to address the question posed by Amaya et al. (2020) regarding the differences in timing between Blob 1.0 and Blob 2.0. They noted, "*An important*

distinction between the recent temperature anomalies (referred to as Blob 2.0) and those from 2013/2014 (referred to as Blob 1.0) is that Blob 2.0 primarily intensified during the summer months. In contrast, Blob 1.0 developed in the winter due to a persistent atmospheric ridge in the Northeast Pacific, which weakened the climatological Aleutian Low and associated surface winds."

While defining the onset as October is reasonable from an event perspective, it leads to entirely different conclusions when analyzing mechanisms. A comparison of Fig. 2b and 2c reveals that the classic mixed layer heat budget, after applying a 90-day low-pass filter, indicates that both ocean processes and surface net heat flux contribute to warming at the time marked by the black double-headed arrow. In contrast, the results from a 365-day low-pass filter show that during the Blobs, ocean processes consistently have a positive contribution, while surface net heat flux consistently has a negative contribution (the direction of the flux is from ocean to the atmosphere, or a net heat loss). This raises the question of which explanation can be trusted.

The primary fluctuations in SSTA in the Blob region occur at the interannual timescale, suggesting that understanding the positive phase of the interannual component of SSTA can explain most of the positive anomalies. Therefore, we argue that focusing on the interannual timescale, particularly by analyzing SSTA after a 365-day low-pass filter, is more suitable. This explains why we exclude this 40-day cooling, because it is the high frequency interference when analyzing the interannual variation.

Minor comments

- Line 138. 'the gyre boundary shifts southward (eastward),' Please verify the sentence.
- Line 182-184, 'resulting in the gyre boundary east of 150W ~ , resulting in the gyre boundary west of 150W shifting northward.' Please verify the sentence.

Thank you for your comments. We modified these sentences and carefully checked others in revising our manuscript.

Reviewer #1 (Remarks to the Author):

Review of revised NEP Blob

I really appreciate the careful and extensive work that was done to respond to reviewer comments. This draft has addressed a number of my concerns, but I find that this revised manuscript is difficult to follow and leaves me with a number of important questions that should be addressed before publication.

Thank you very much for your thoughtful review. We provide point-by-point responses to your concerns below, and to those of the other two reviewers, and have revised our manuscript accordingly. Our detailed responses are provided below, with your original comments in black font and our replies in blue font.

1. First, I'm struck by the remarkable trend in the gyre boundary latitude from 1993-2022, where the trend is +2 to +3 st devs! This deserves an article of its own, and seems to be closely related to the rapid increase in North Pacific SSTs in the period since 2013 (Hu et al 2024). Also see the simple SSTA time series figure I created for the 30-50N, 150-135W region that also shows a large warming trend in addition to the interannual extremes in this region from 1993-2023.

Hu et al 2024: Accelerated warming in the North Pacific since 2013. Nature Climate Change. <https://doi.org/10.1038/s41558-024-02088-x>

We appreciate your thoughts on this and have now cited this paper (Ref. 32) in the discussion of our revised manuscript (the third-to-last paragraph). Your insightful comment and plot do indeed suggest that this warming may be due to the trend in the gyre boundary. Thank you very much for highlighting this point.

Figure AR1-1. Attachment provided by Reviewer 1. SSTA time series for the 30-50°N, 150-135°W region.

2. If impacts on marine life and fisheries are a key concern for the developing science around marine heatwaves, why focus on what I'd call "persistent warm events" (inter annual time scales) rather than the shorter-term extremes and the longer-term trend? Is this something unique to the two warm events in the NE Pacific that are the focus of this work? Are you trying to broaden an argument about Hobday et al's definition of Marine Heat Waves (based on daily SST>90th percentile values for at least 5 consecutive days)?

We do not intend to expand upon the definition of marine heatwaves as established by Hobday et al. (2016) – a marine heatwave (MHW) definition which has greater utility for marine biologists than physical oceanographers.

The occurrences of MHW events are intrinsically linked to SSTA exceeding an appropriately selected threshold. SSTA can occur on a variety of time scales, including interdecadal, interannual, interseasonal, and intraseasonal, and interacting across multiple scales. To understand the cause of the SSTA, it is important to understand the dynamics and thermodynamics – not only in terms of the local processes that can be diagnosed by a heat budget analysis, but also the modulating influences by modes of climate variability and their teleconnections. The scope of this analysis can be considerable.

To streamline our analysis, we concentrate on the timescale that is most influential

on the temperature extremes. The northeast Pacific is particularly noteworthy, as interannual climate variability appears to be the dominant cause of variations in northeast Pacific SSTA. Using the Hobday et al. (2016) definition, we can identify the onset and end dates of MHW events and calculate the average (or median) of both the unfiltered SSTA and filtered SSTA for different time scales (Figure R1-1).

Figure R1-1. Boxplot of (a) unfiltered SSTA, (b) its interannual component, (c) intra-annual component, and (d) intraseasonal component when the MHW occurred. The red line in the box denotes the median. (e) SSTA and threshold within 150 °W – 135 °W and 40 °N – 50 °N.

The interannual component contributes dominantly to temperature extremes during marine heatwave events, accounting for 80.7% of the mean value relative to the unfiltered SSTA ($\overline{SSTA_{Interannual-MHW}} / \overline{SSTA_{unfiltered-MHW}}$, Figure R1-1b). On the other hand, the interannual component of the mixed layer temperature anomalies (MLTA) accounted for 80% of the variations observed in the unfiltered MLTA ($STD(MLTA_{Interannual}) / STD(MLTA_{unfiltered})$) (see Methods 5d). These substantial proportions highlight the critical role of interannual-scale variability in driving extreme SSTA.

Additionally, during the two blob events, the interannual component alone surpasses the threshold, demonstrating that understanding the interannual variability of SSTA is sufficient to explain the blob dynamics (Figure R1-1e). Therefore, we have selected the interannual time scale for our analysis.

3. On identifying the timing of “event” onset, it seems to me that the SSTA (or MLTA) tendency is more likely to provide insights into the underlying physical mechanisms. Where SSTA/MLTA crosses the zero line does not seem all that important to me if the interest is in understanding the processes, as the zero-line can be arbitrarily defined based on different reference periods.

Thank you for your comment. We concur with the majority of your statement. While the zero line can be defined arbitrarily based on various reference periods, the zero point (intersection point between SSTA and zero line) still provides useful information as it can be interpreted as a transition boundary between dominant mechanisms. However, we also noticed the transition boundary marked by the zero point may not be explicit if the long-term trend is significant. For example, in the blob region, if we artificially double the long-term trend in SSTA, the SSTA will intersect less with the zero line and thus has less zero point.

In response to your 5th general comment, we will show the zero-point in the period of SSTA increase (positive SSTA tendency) acts as a transition boundary between the two stages, i.e., ‘Wind with warmer air blow over low SSTA region’ and ‘Wind with warmer air blow over high SSTA region’. The air-sea interaction of these two stages is different. Therefore, although the identification of the zero point involves subjectivity, combining the analysis of SSTA tendency allows us to gain a deeper understanding of the temporal variations in the underlying physical mechanisms.

4. On the time history of atmospheric forcing: Why not plot the time series of your wind stress curl forcing pattern, which presumable drives both the surface heat fluxes and the Sverdrup response (with some lag-time).

Thank you very much for this valuable suggestion. The accompanying diagrams

that depict the annual average SLP differences between the periods of 2014-2022 and 1993-2002 have been very beneficial.

Following your suggestion, we analyzed the first (18.6% variance explained) and second (14.3%) EOF modes of climatological monthly anomalies in 1-year low-pass filtered wind stress curl. In accordance with Sverdrup theory, the spatial structure of the first EOF mode (Figure R1-2a) would correspond to a contraction of the subarctic gyre accompanied by subtropical gyre expansion. Conversely, the second EOF mode (Figure R1-2c) exhibits a pattern that would induce simultaneous contraction of both gyres. Notably, both modes display pronounced negative anomaly centers over the subarctic gyre region, suggesting that these patterns contribute to northward displacement of the gyre boundary through their cumulative effects.

Figure R1-2. First (upper panels) and second (low panels) EOF mode of climatological monthly anomaly of 1-year low-pass filtered wind stress curl within the region of 180W-120W and 30-60N.

The spatial characteristics of the second EOF mode (Figure R1-2c) show remarkable consistency with our composite anomaly patterns (Figure 4c, d in revision). Furthermore, the temporal evolution of this mode, as evidenced by positive phase

occurrences in its principal component (Figure R1-2d), demonstrates temporal coherence with the emergence of Blobs. This phase synchronization implies the second EOF mode may represent a key atmospheric driver modulating upper ocean dynamic conditions through wind stress curl anomalies.

However, the surface heat flux is affected by wind direction, and ocean circulation is influenced by changes in the sign of the wind stress curl – with both factors governed by wind patterns. On the other hand, the Sverdrup balance as the dominate air-sea interaction mechanism is valid primarily within the Blob region, which means the WSC may not be used for characterizing the large-scale driving fields associated with the Blob. Given that the strength and direction of the near-surface winds are due to fluctuations in SLPA (Figure R1-3), we have opted to focus our analysis on SLPA rather than wind stress curl.

Figure R1-3. Geographic distribution of point-wise temporal correlations between monthly resolved SLPA and wind stress curl anomalies (with climatology removed and 1-year low pass filtered) for the period of 1993-2022. The black line contours correspond to correlations of -0.7.

I've plotted a simple time series of 12-month low-pass filtered SLP anomalies for 30-50N, 150-130W (attached), and it shows substantial inter annual variability, multi-decadal variations, and century-long trends.

Figure AR1-2. Attachment of Reviewer 1. 12-month low-pass filtered SLP anomalies for 30-50°N, 150-135°W.

This region is part of the wintertime TNH teleconnection pattern, but these variations are not limited to the winter season or the TNH overall pattern that has centers of action outside the NE Pacific (see the attached annual mean SLP difference map for 2014-2022 - 1993-2002).

Figure AR1-3. Attachment of Reviewer 1. Annual mean SLP difference map for 2014-2022 - 1993-2002.

Thank you very much again for your thoughts. Unfortunately, we are unclear

whether there is an issue raised here, or primarily a comment. The elements pertaining to multi-decadal trends fall outside the scope of this research. Nevertheless, we are willing to provide a response grounded in our own understanding.

The primary finding of our research indicates that the interannual variability of SLPA is a significant factor influencing the interannual variability of SSTA. In the context of lower-frequency signals, such as decadal variations in SSTA, our ongoing research indicates that other oceanic responses (Precipitation-stratification enhancement at the base of the mixed layer-decreased downward heat transfer from the mixed layer to the depth-SSTA increase) to the interannual variability of SLPA may also be of considerable importance. Should you express interest in this specific aspect, we would be pleased to engage in a private discussion following the acceptance of this manuscript.

Concerning long-term trends or multi-decadal variations in SLPA, we unfortunately do not have any additional valuable insights at this time.

The activity centers illustrated in Figures AR1-3 primarily represent the long-term trends of SLPA and do not necessarily correlate with variations in the TNH pattern. In essence, the long-term variation of SLPA may not be driven by the TNH.

5. In summary, I found a lot of really interesting material in this work, with the major contribution being the clear demonstration of the primary role for ocean dynamics in the warming of the NE Pacific at multiple time scales. I do not like the focus on the “blobs” because that terminology is too vague for my tastes, and it limits the focus to just two persistent warm events that are now a part of a series of warm extremes that are also embedded within a longer-term ocean warming that is still playing out. Finally, the nature of the atmospheric forcing, and how that atmospheric forcing drives both surface fluxes and ocean dynamics, remains hard to follow by the approach taken here. If atmospheric forcing has multi-season and longer time scale periods, wouldn't we expect to see air-sea heat exchanges having those same time scales (assuming that the warming of the ocean doesn't become so great that it comes into equilibrium with surface air temperature and humidity)?

Thank you for your valuable comment and also endorsement of this research. From the perspective of the atmospheric forcing, there indeed exists multi-season and longer time scale period variations. However, air-sea interaction processes within the northeast Pacific region are non-linear. Essentially, the competition (interaction) between different air-sea interaction mechanisms makes their time series more variable.

We take the transition between two quasi-equilibrium states (positive SSTA in the NE Pacific region-northward excursion of the gyre boundary, and negative SSTA-southward excursion of the gyre boundary) in our study as an example, to elucidate the evolution of the primary air-sea interaction processes that underlie the variations in SSTA (Figure R1-3; new Figure 7 in the revised manuscript).

For clarity, we delineate three temporal reference points: the SSTA trough (SSTA-Trough), the SSTA at zero value (SSTA-0), and the SSTA ridge (SSTA-Ridge).

Figure R1-3. Schematic view of the physical mechanism evolution.

The transition from the SSTA-Trough to SSTA-0 signifies the cessation of negative SSTA anomalies, marking the decay of a cold spell, while the shift from SSTA-0 to the SSTA-Ridge indicates the onset of positive SSTA anomalies, representing the onset, development and maturation of the marine heatwave. These two phases collectively constitute a complete positive phase of temperature tendency. Conversely, the transition from the SSTA-Ridge to the SSTA-0 denotes the end of positive SSTA anomalies, while

the shift from the SSTA-0 to the SSTA-Trough signifies the occurrence of negative SSTA anomalies, indicating the onset of a cold spell. These latter two stages represent the negative phase of the temperature tendency. The entire fluctuation, characterized by the sequence of SSTA-Trough - SSTA-0 - SSTA-Ridge - SSTA-0 - SSTA-Trough, encapsulates the positive-negative phase of the temperature tendency, thereby representing a complete period of SSTA change.

We next outline the initial conditions at the three temporal reference points:

- 1) At the SSTA-Trough, the gyre boundary is positioned at its southernmost extent, the wind stress curl transitions from a positive to a negative anomaly, and the wind direction shifts from a southwest anomaly to a northeast anomaly.
- 2) At the SSTA-Ridge, the gyre boundary reaches its northernmost point, the wind stress curl transitions from a negative to a positive anomaly, and the wind direction changes from a northeast anomaly to a southwest anomaly.
- 3) At the SSTA-0, the gyre boundary is situated at its average position.

To simplify the discussion of heat flux, we represent changes in net heat flux through variations in latent heat flux, as the latter predominantly influences net heat flux changes (Figure 6a in the revised manuscript).

Next, we consider the temporal evolution.

- 1) Positive phase of the temperature tendency (SSTA-Trough - SSTA-0 - SSTA-Ridge): The alteration in wind direction facilitates the rapid influx of warmer, more humid air from the south into the northeast Pacific region. This movement of near-surface air from the warmer southern region to the cooler northern region, leads to a temperature decrease in air parcel and a contraction in volume of air, which in turn results in a reduction in wind speed (Figure 6f).

During the transition from the SSTA-Trough to SSTA-0 (cessation of the cold spell), the negative anomaly of specific humidity remains the predominant factor contributing to the reduction of latent heat flux, while the situation of **"Wind with warmer air blow over negative SSTA region"** is the primary driver of reduced ocean heat loss (this explains the positive latent heat anomaly in 2013). Concurrently, influenced by the negative anomaly of wind stress curl

(Sverdrup balance), warmer seawater from the south gradually intrudes into the northeast Pacific region. During this stage, both advection and latent heat flux play significant roles in positively influencing the temperature tendency.

Given that the ocean's dynamic response to atmospheric changes is slower than its thermodynamic response, as the gyre boundary anomaly traverses the zero-value position and gradually shifts northward, the positive anomaly associated with the northward advection of warm water progressively elevates the SSTA, prompting a transition in air-sea interaction to "**Wind with warmer air blow over positive SSTA region**". This shift is reflected in the dominance of advection during the SSTA-0 to SSTA-Ridge, while latent heat flux remains at a near-zero anomaly (Figure 3d, the second half of 2013 and around 2019).

- 2) Negative phase of the temperature tendency (SSTA-Ridge – SSTA-0 – SSTA-trough): During the transition from the SSTA-Ridge to SSTA-0, the near-surface wind field shifts to a northwest wind anomaly, resulting in the movement of near-surface air from the cooler northern region to the warmer southern region. This movement leads to a temperature increase in the air parcel (diabatic heating) and an expansion in volume, resulting in an increase in wind speed (Figure 6f).

The situation of "**Wind with cooler air blow over positive SSTA region**" contributes to an increase in the turbulent heat flux and enhanced ocean heat loss. Simultaneously, driven by the positive wind stress curl anomaly, subtropical warm water continues its southward movement, exiting the northeast Pacific region. During this phase, advection sustains the heat supply to the northeast Pacific region, while turbulent heat flux predominates in terms of heat loss.

In the transition from SSTA-0 to the SSTA-trough, the condition of "**Wind with cooler air blow over negative SSTA region**" leads to a gradual return of latent heat flux from negative to positive anomaly, as the gyre boundary continues its southward movement past the average position, with advection prevailing in the cooling process.

In summary, within an ideal cycle of SSTA changes primarily driven by gyre boundary and net heat flux variations, the interaction between advection processes and surface net heat flux processes results in a non-stationary temporal change in latent heat flux characterized by the sequence of "**negative anomaly (weakened heat loss) - zero anomaly - positive anomaly (enhanced heat loss) - negative anomaly (weakened heat loss)**". Notably, the magnitude of the positive anomaly exceeds that of the negative anomaly, and the duration of the positive phase is shorter than that of the negative phase. Furthermore, traditional linear filters are inadequate for effectively addressing non-stationary time series, indicating that even an ideal SSTA tendency corresponds to "distorted" surface net heat flux changes. Given the complexities of real-world scenarios, additional mechanisms and competitive or interactive processes of surface net heat flux often drive more high-frequency heat flux responses resulting from low-frequency changes in atmospheric forcing.

In addition to these over-arching comments I have a number of specific comments following the line numbers below.

Sincerely,

Nate Mantua

NOAA/NMFS Southwest Fisheries Science Center

Santa Cruz, CA

Specific comments:

1-2: To the extent that NEP blob = NEP MHW, this title is redundant. Maybe stick with MHW, or “persistent MHW events”?

Thank you for your comment. We follow your suggestion by removing Blob from the title and adding ‘persistent’..

38: I am not a fan of continuing to use the “blob” terminology because it has no criteria or definition that could be objectively applied to these or other events. Why not

say “multi-season” or “persistent” MHW events in 2013-15 and 2019-20 instead?

Done. This is now revised to ‘multi-season MHW events in 2013-15 and 2019-2020’.

50: the proximate driver of surface wind-stress curl over the NE Pacific clearly projects onto the TNH pattern, but the TNH pattern was only identified for winter months and it also has centers of action over N. America ... what about a simpler index of SLP over the NE Pacific to track the local/regional atmospheric signature of the relevant wind stress forcing? What are the timescales of that SLP index variability?

Thank you for your comment. The Blob 1.0, defined by the interannual component of SSTA crossing the zero line, commenced in March, while Blob 2.0 began in May. The delayed response of the ocean to atmospheric dynamical forcing, which has been determined to have a maximum lag-correlation of four months, is indicated by the gyre boundary anomaly and wind stress curl anomaly (Figure R1-4). This delay corresponds to the months of November, December, and January when looking back four months from March, April, and May, aligning with the TNH pattern which was only identified during the winter months. Regarding the assertion that "it also has centers of action over North America," Liang et al. (2017) also reported a corresponding relationship with cold spells in the Atlantic.

In terms of predicting the Blob, the SLPA time series in Figure 8b can serve as a simpler index of SLP over the northeast Pacific to monitor the local and regional atmospheric signature of the relevant wind stress forcing. Both our research and that of Liang et al. (2017) suggest that the TNH is a primary driver of the Blob, yet there is relatively limited research focused on the TNH, which highlights a lack of understanding regarding the atmospheric physical processes underlying it.

The wavelet analysis indicates that the climatological anomalies of monthly mean SLPA variations are concentrated within 1–2.5 years, with characteristic time scales consistent with the multi-season persistence of the Blob and the TNH (Figure R1-5).

Figure R1-4 lag-correlation between wind stress curl anomaly (WSCA) and gyre boundary anomaly (GB). The triangles denote the value that its p-value is smaller than 0.05.

Figure R1-5 Wavelet of the climatological anomalies of monthly mean SLPA within Alaskan gyre region (blue line, Figure 8b in the revision).

72-83: This paragraph makes an argument for ocean dynamic responses favoring multi-season persistence of warm events, rather than the initiation, of the “marine heatwave” extremes that have been previously defined as daily SSTs exceeding 90th percentile values for more than 5 consecutive days (Hobday et al 2016).

Indeed, the definition of heatwave and the analysis of their underlying physical mechanisms are distinct matters. The study of the physical mechanisms of surface

marine heatwaves focuses on SSTA changes. Specifically, when a threshold is reached, the inquiry shifts to understanding why the SSTA exceeds a defined reference line, which extends beyond the scope of the marine heatwave itself.

For instance, as in the Blob region (Figure R1-2), the interannual component of SSTA contributes over 80% to the positive values of SSTA, while the intra-annual signal accounts for less than 20%. To streamline our research and to mitigate the potential for high-frequency signals in the mixed layer heat budget equation to obscure low-frequency signals, we focused our analysis on the interannual component of SSTA.

The initiation and persistence of MHW events are closely linked to the occurrence and development of the positive phase in SSTA, therefore identifying its characteristic time scale is essential. For example, on the interannual time scale, we not only revised the reference month of Blob 1.0 but also confirmed that the oceanic dynamic response to the atmosphere follows the Sverdrup relationship. This response indicates that the ocean dynamic response to the atmosphere represents a low pass filter, which explains that although SLPA has high-frequency variations, the ocean dynamic processes favor the low-frequency components within it.

We have now included an explanation of the interannual component of SSTA and its relation to the definition provided by Hobday et al. (2016) in the Introduction, to help better clarify.

115: are ocean dynamics key to the formation of the 2013 and 2019 MHW events, or to the multi-season persistence?

We have found that the ocean dynamics are key to both the formation and multi-season persistence of these two MHW events. For the formation, please refer to our reply to the reviewer's 5th major comment. The detailed process during the stage of SSTA-0-SSTA ridge confirms the ocean dynamics are key to the formation of MHW events. For the multi-season persistence, shifts in the Sverdrup balance are favored by the low frequency component of the atmospheric forcing, with the ocean acting as a low-pass filter to the atmospheric forcing.

120-121: The same persistent wind stress curl patterns that drive ocean dynamics are also driving surface heat fluxes and related thermodynamics. Bond et al's focus on the early part of the 2013-14 MHW showed that both ocean dynamics and thermodynamics contributed to the initiation and persistence of the event (Sept 2013-Feb 2014).

The detailed dynamics refer to our reply to the 5th general comment. Our results show that the contribution of the net surface heat flux to SSTA is positive but small. Bond et al. (2015) show that the contribution (Figure 4, the red line's fourth point from the right) of the net surface heat flux to the SSTA is negative. However, their figure did not show the mean value as the reference line, which makes it difficult to compare.

[REDACTED]

Screenshot of Figure 4. in Bond, N. A., Cronin, M. F., Freeland, H. & Mantua, N. Causes and impacts of the 2014 warm anomaly in the NE Pacific. *Geophys. Res. Lett.* 42, 3414–3420 (2015).

142-145: Figure 1 shows a remarkable trend over the 1993-2022 period, where the trend is $\sim +2$ to $+3$ st devs! This could be the focus of a related journal article that

explored the dynamic contributions to NE Pacific SST warming trends over past 3 decades.

We appreciate your recommendation and have included the referenced paper in the discussion of our revision.

Hu et al 2024: Accelerated warming in the North Pacific since 2013. Nature Climate Change. <https://doi.org/10.1038/s41558-024-02088-x>

148: As noted above, Figure 1 shows an incredible trend in the NPC latitude that was especially confined to the far eastern Pacific (east of ~145W) from 2014-2017 but expanded to include the entire Pacific from 2018-2022. The period of northward expansion is also characterized by rapid and persistent warming of the entire N. Pacific (Hu et al. 2024).

Thank you for your recommendation. We have incorporated the cited paper into our revision. Nevertheless, the mechanism outlined in our study can only partially elucidate the rapid warming observed in the 30-60° N region, as this area is specifically influenced by the excursion of the gyre boundary.

Our ongoing research into the decadal variations of SSTA in the northeast Pacific region indicates that the tendency of the SSTA is influenced by the stratification at the base of the mixed layer and is linked to interannual variations in the surface freshwater flux. This is the subject of our forthcoming research; we would be happy to share this aspect of our research with you in the near future.

161-165: while your detrended data are good for identifying inter annual peaks, the unfiltered data shows that the positive phase encompasses the entire 2013-2022 period.

We find that the long-term trend and interannual variation of the gyre boundary are comparable in magnitude. Therefore, even though the positive phase spans the entire period from 2013 to 2022, it is important to consider the long-term trend of the gyre boundary to ensure that data after 2010 are not disproportionately emphasized.

211-225: This paragraph about onset dates from this versus other studies is difficult

to follow. I understand that this work argues for the importance of the inter annual time scale over the sub-yearly variations. However, Figure 3 has so much information over so many wiggles and lines that my brain cannot simply look at these panels and extract what the authors are trying to convey. One possible remedy to this would be to not show the continuous time series from 2010-2023. But instead limit the figure to the warm events of interest +/- a year on either end. There is simply too much information presented here for my taste.

We have now revised the figure and limit it to the warm events of interest, from 2012 to 2022.

235-241: 3d shows that the net sfc heat flux and advection terms were positive in the first half of 2013, during a time that the temperature anomaly increased from negative to positive values. Isn't this an indication that dynamic and thermodynamic processes led to the change from cool to warm SSTA/MLTA in the focal region?

For the detailed dynamics, please see our reply to the reviewer's 5th major comment.

242-245: Again, I'm not a fan of "the blobs" terminology because they are so loosely defined. Your use here looks to be defined as periods of warmer than normal SST/MLT that persist for > 1 year. This is clearly at odds with Hobday et al (2016) definition of MHW events as extreme daily SST anomalies (>90th percentile) that persists for at least 5 consecutive days. In your Introduction you could explicitly state that this work is focused on the inter annual time scales over shorter time scales associated with the Hobday et al definition. And then show your results that support the importance of inter annual time scale processes in the formation and persistence of the 2013-2016 and 2019 warm events in the NE Pacific.

Thank you for your comment. In the introduction, we have now emphasized the focus in this paper on interannual variations of SSTA in the northeast Pacific region, while reducing the Blob terminology throughout the manuscript. Additionally, in the reply to 2nd general comment, we clarified why the interannual time scales can account for the heatwaves as defined by Hobday et al. (2016).

267-269: The +/- phase terminology here is confusing because of the dipole nature of the wind stress curl anomaly patterns. Your + phase has an expanding subtropical gyre in the NE Pacific, and a contracting subtropical gyre in the central Pacific. Your negative phase has the subarctic gyre expanding and the subtropical gyre contracting in the east with little change farther to the west.

Thank you for your comment. We have now revised this paragraph to be clearer.

315-318: Yet the streamfunction anomaly pattern shown in panel e (2014) doesn't look anything like those in panels d and f. I agree that the upper ocean warming is similar, but not the streamfunction anomalies.

Thank you for your comment. we revised the description to 'Upper 200m streamfunction anomalies indicate subtropical gyre expansion and subarctic gyre contraction during positive phases (Fig. 5d). The exceptional 2019 northward shift to 43.9°N reflects intensified anticyclonic anomalies across 30-60°N (Fig. 5f).'

325: replace "suggest" with "show"

Done.

379-381: Fig 6(a) shows that the net heat flux and latent heat flux are positive for most of 2013 and 2014, consistent with surface fluxes promoting a positive MLTA tendency in those years.

For the detailed dynamics, please see our reply to the reviewer's 5th major comment.

412-413: Does the SLPA pattern propagate northward, or does the SW/NE dipole with centers in the western tropical Pacific and NE Pacific intensify following while the SLP anomalies in the eastern tropical Pacific weaken over the 5-months lag time?

We do not rule out the possibility of 'the SW/NE dipole with centers in the western tropical Pacific and NE Pacific intensify following while the SLP anomalies in the eastern tropical Pacific weaken over the 5-months lag time', as there is currently no

literature supporting the northern propagation. We have revised this paragraph accordingly.

447: The emergence of the 2013-14 warming looks to be the result of persistent atmospheric forcing that generates rapid upper ocean warming due to dynamic and thermodynamic processes. The persistence and warming at depth look to be attributed to the dynamic processes you focus on here.

Thank you for your comment. Please refer to our reply to the reviewer's 5th major comment. During the onset of the warming event in 2013-14, ocean dynamics facilitated the northward transport of warm water into the northeast Pacific region. In terms of thermodynamic processes, which are primarily governed by latent heat flux, the variations observed are largely determined by changes in SSTA influenced by ocean dynamics. The results from the mixed layer heat budget analysis support the notion that, beginning from the point where SSTA crosses the zero line, the temperature tendency associated with ocean advection predominantly dictate the temperature tendency within the box model or the mixed layer heat budget.

460-461: I remain unconvinced that ocean advection is the only factor contributing to the emergence of the two warm events examined here. The surface heat flux anomaly maps included in your rebuttal letter show that 2013 had a pattern of positive surface heat flux anomalies that spanned the entire NE Pacific. This was also a key result of Bond et al's heat budget analysis for the Sept 2013-Feb 2014 period.

Thank you for your comment. The dominance of the lateral advection term (green line, Figure 3d) over the total temperature tendency (red line, Figure 3d) confirms that ocean advection serves as the primary driver of the Blobs' emergence. Although positive surface heat flux anomalies spanned the entire NE Pacific during this period, the net surface heat flux term exhibited a slight negative contribution (blue lines, Figure 3c for the classical mixed-layer heat budget and Figure 3d for the box model).

The basin-wide positive surface heat flux anomalies around 2013 were caused by the specific humidity difference (red line, Figure 6b in the revision) and can be

attributed to the scenario of 'warmer winds blowing over negative SSTA'. Meanwhile, the peak in surface heat flux anomalies from September 2013 to February 2014 was driven by wind speed variations (blue line, Figure 6b in the revision), consistent with the situation of 'warmer winds blowing over positive SSTA'.

For more details please refer to our reply to the 5th general comment.

506-512: You also show an impressive multidecadal trend in the gyre boundary, and others have shown a corresponding trend in NP SSTs and mixed layer depths. The same trend exists in the atmospheric SLP field. So there are long time scales in atmospheric forcing worth considering too. You could create a more targeted atmospheric forcing index based on the SLPA pattern. A spectral analysis of that time series will show the time scales the atmospheric forcing pattern (for wind stress curl and sensible and latent heat fluxes).

Thank you for your comment. You have referenced four key aspects: MLD, SSTA, SLPA, and the gyre boundary (GB). We feel that variations in the MLD fall outside the scope of our current study, and so don't explicitly address that here. Concerning SSTA, GB, and SLPA, on an interannual basis, the ocean's dynamic response to atmospheric conditions influences the increase in SSTA through heat advection anomalies, while the thermodynamic response contributes to a decrease in SSTA. Although the underlying physical mechanisms for warming and cooling differ, practically, a strong linear regression relationship exists between SLPA and SSTA, as illustrated in Liang et al. (2017, Figure 2a). This suggests that the response of SSTA to SLPA forcing is quasi-linear.

Our ongoing research into decadal variations of SSTA in the northeast Pacific region indicates that the tendency of the SSTA is influenced by the stratification at the base of the mixed layer and is linked to interannual variations in the surface freshwater flux. This suggests that the SLPA also influence the damping time scale of the SSTA, indicating that the SSTA may exhibit lower frequency variations relative to the characteristic time scale of the SLPA forcing. As this is the subject of our forthcoming research, we would be happy to share this with you in the near future.

614: Do you mean the red line in 3c (MLTA tendency) and red shading (SSTA) in 3d? or do you mean the MLTA tendency and Box TA tendency? There is no red shading in 3c.

Thank you for pointing this out. We have now revised 3c to 3d.

Rebuttal letter:

P9, R1-4: I am surprised by this statement: “Consequently, it is proposed that the anomaly in temperature transport due to the northward expansion of the subtropical circulation is counterbalanced by the net heat flux anomaly, leading to a lack of a linear warming trend in local SSTA.” There is clearly a strong positive warming trend in the region 30-50N, 150W-135W from 1993-2022. Also see Hu et al 2024: Accelerated warming in the North Pacific since 2013. Nature Climate Change. <https://doi.org/10.1038/s41558-024-02088-x>

Thank you for your careful attention. Our previous statement indeed lacked thorough consideration. However, the findings related to the heat flux anomaly indicate a notable strengthening of surface heat loss. This suggests that the northward migration of the gyre boundary contributes to a long-term increasing trend in SSTA, while the increase in surface heat flux has, to a large extent, mitigated this trend. The subsequent inquiry should focus on quantifying these two effects.

To begin, we delineate these trends:

1. The northward movement of the circulation boundary is observed at a rate of 1.33° of latitude per decade (from 1993 to 2020). Assuming that the gyre boundary aligns with a series of specific isotherms, it is reasonable to assume that these isotherms should also shift northward at the same rate of 1.33° of latitude per decade.

2. The increase in SSTA within the region of 40-50° N and 150-135° W is 0.17 °C/decade (from 1983 to 2020). Given that the SST gradient in this area ranges from 2°C/3° latitude to 2°C /4° latitude, the movement rate of the SST isotherm is estimated to be between 0.25° and 0.33° of latitude per decade ($0.17 \text{ °C} \div 2\text{°C}/3^\circ \sim 0.17 \text{ °C} \div 2\text{°C}/4^\circ$), or the temperature increase purely induced by the northward

excursion of the gyre boundary is estimated to be between 0.67 °C and 0.89 °C per decade ($1.33^{\circ}\times 2^{\circ}\text{C}/4^{\circ} \sim 1.33^{\circ}\times 2^{\circ}\text{C}/3^{\circ}$). This indicates that the rise in surface heat flux has significantly counterbalanced the temperature increase associated with the northward shift of the gyre boundary.

Figure R1-7: I'd say that the spatial pattern of surface heat flux anomalies in 2013 encompassed nearly all of the NE Pacific, a pattern that is unique in the 2003-2022 period for the extent and magnitude of positive surface heat flux anomalies in the NE Pacific.

We appreciate your reminder. The spatial distribution of surface heat flux anomalies observed in 2013 can be elucidated through the analysis of the latent heat flux (Figure 6 in the revised manuscript). The year 2013 was characterized by an anomaly of reduced wind speed (Figure 6f) and a decline in the humidity difference. This can be linked to the anomalous southeasterly winds that promoted the northward influx of warm, humid air into the cooler regions, resulting in a decrease in the humidity difference.

Reviewer #2 (Remarks to the Author):

I thank the authors for their great efforts in addressing the comments and concerns raised by the three reviewers. Although some of my comments have been well addressed, I may have more concerns to put forward in this round of review:

Thank you very much for your thoughtful review. We have now revised our manuscript based on your comments and those of the other two reviewers. Our detailed responses are provided below, with your original comments in black font and our replies in blue font.

1. The motivation or evidence of authors to focus on the interannual timescales is not clear, although interannual time scale/1-year filter is used in many places throughout the manuscript.

We revised the Introduction to highlight our motivation to focus on the interannual timescales.

The SSTA encompass a range of signals, including interdecadal, interannual, interseasonal, and intraseasonal variations, as well as interactions across different time scales. To determine which specific signal leads to SSTA exceeding the defined threshold, one must analyze which signal exhibits a positive anomaly during the period of the temperature extreme and provide an explanation.

To streamline our analysis, we concentrate on the timescale that is most influential on the temperature extremes. The northeast Pacific is particularly noteworthy, as interannual climate variability appears to be the dominant cause of variations in northeast Pacific SSTA. Using the Hobday et al. (2016) definition, we can identify the onset and end dates of MHW events and calculate the average (or median) of both the unfiltered SSTA and filtered SSTA for different time scales (Figure R2-1).

The interannual component contributes dominantly to temperature extremes during marine heatwave events, accounting for 80.7% of the mean value relative to the unfiltered SSTA ($\frac{\overline{SSTA_{Interannual-MHW}}}{\overline{SSTA_{unfiltered-MHW}}}$). On the other hand, the interannual component of the mixed layer temperature anomalies (MLTA) accounted

for 80% of the variations observed in the unfiltered MLTA ($\frac{STD(MLTA_{interannual})}{STD(MLTA_{unfiltered})}$) (see Methods 5d). These substantial proportions highlight the critical role of interannual-scale variability in driving extreme SSTA.

Alternatively, during the two blob events, the interannual component alone surpasses the threshold, demonstrating that understanding the interannual variability of SSTA is sufficient to explain the blob dynamics. Therefore, we have selected the interannual time scale for our analysis.

Figure R2-1. Boxplot of (a) unfiltered SSTA, (b) its interannual component, (c) intra-annual component, and (d) intraseasonal component when the MHW occurred. The red line in the box denotes the median. (e) SSTA and threshold within 150 °W – 135 °W and 40 °N – 50 °N.

2. In the revision, authors used a box model to try to give calculation for the heat budget of the blob marine heatwaves. However, based on the method description from Line 620, this method may not well distinguish horizontal and vertical advection processes. On the other hand, the relationship between the mixed layer heat budget and box model heat budget is not well addressed, which may affect the consistency and readability of

this paper. In the following subsection b), authors give detailed discussion on the entrainment term, whose relationship with the box model also needs clear description.

Thank you for your comment. The explanation of Method 5b was indeed not clear enough, and so we now revised Lines 652-660 as follows.

Another simple way to understand this issue is by applying the classic mixed layer heat budget equation to a cubic control volume. Consider a cubic control volume with fixed dimensions (width, length, and thickness), similar to a basic grid cell in a numerical model. Since the volume is time-invariant, the $\frac{\partial h}{\partial t}$ and ∇h terms vanish. This leads to:

$$\frac{\partial MLTA}{\partial t} = \frac{Q_{net} - Q_h}{\rho C_p h} - \bar{\mathbf{u}} \cdot \nabla \bar{T} - \left(\frac{\bar{T} - T_{-h}}{h} \right) w_{-h} + \text{Mixing}$$

Expanding the third term (entrainment term) on the right-hand side, we have:

$$\frac{\partial MLTA}{\partial t} = \frac{Q_{net} - Q_h}{\rho C_p h} - \bar{\mathbf{u}} \cdot \nabla \bar{T} - \frac{\bar{T}}{h} w_{-h} + \frac{T_{-h}}{h} w_{-h} + \text{Mixing}$$

where $-\bar{\mathbf{u}} \cdot \nabla \bar{T} - \frac{\bar{T}}{h} w_{-h}$ can be written as $-\nabla(\bar{T}\bar{\mathbf{u}})$. Then we get:

$$\frac{\partial MLTA}{\partial t} = \frac{Q_{net} - Q_h}{\rho C_p h} - \nabla(\bar{T}\bar{\mathbf{u}}) + \frac{T_{-h}}{h} w_{-h} + \text{Mixing}$$

If we write the above equation in the discrete form, we have

$$\frac{\partial MLTA}{\partial t} = \frac{Q_{net} - Q_h}{\rho C_p h} + \frac{\oint \left(\int_{-h}^0 \mathbf{u}_L T_L dz \right) dl}{\iiint_{-h}^0 dx dy dz} + \frac{T_{-h}}{h} w_{-h} + \text{Mixing}$$

Suppose $\frac{T_{-h}}{h} w_{-h} + \text{Mixing} = \text{residual}$, we get the equation of the box model,

$$\frac{\partial BOXTA}{\partial t} = \frac{Q_{net} - Q_H}{\rho C_p H} + \frac{\oint \left(\int_{-H}^0 \mathbf{u}_L T_L dz \right) dl}{\iiint_{-H}^0 dx dy dz} + \text{residual}$$

From the above derivation, we can see that the classic mixed layer heat budget decomposes the divergence of the horizontal temperature flux ($-\nabla(\bar{T}\bar{\mathbf{u}})$) into the product of the horizontal velocity and temperature gradient ($-\bar{\mathbf{u}} \cdot \nabla \bar{T}$), as well as the product of temperature and the divergence of horizontal velocity ($-\frac{\bar{T}}{h} w_{-h} = -\bar{T} \nabla \cdot \bar{\mathbf{u}}$). Then, the product of temperature and the divergence of horizontal velocity ($-\frac{\bar{T}}{h} w_{-h}$)

is combined with the vertical temperature advection flux ($\frac{T-h}{h}w_{-h}$), expressed in the form of effective flux ($-\left(\frac{\bar{T}-T-h}{h}\right)w_{-h}$).

However, separating $-\frac{\bar{T}}{h}w_{-h}$ from the entrainment term of the classic mixed layer heat budget makes the results more difficult to interpret. Based on our doubts about the oceanic terms in the classic mixed layer heat budget, we believe that the most reliable form is its simplest form, i.e., $\frac{\partial MLTA}{\partial t} = \frac{Q_{net}-Q_h}{\rho C_p h} + \text{residual}$, where the mixed layer temperature tendency and surface heat flux are explicitly represented, and the residual term represent the sum of the oceanic processes.

On the other hand, whether the results of the box model can be used to enhance the mixed-layer heat budget analysis can be validated by the correlation coefficients for the total temperature tendencies ($\frac{\partial BOXTA}{\partial t}$ and $\frac{\partial MLTA}{\partial t}$), net heat flux terms ($\frac{Q_{net}-Q_h}{\rho C_p h}$ and $\frac{Q_{net}-Q_H}{\rho C_p H}$), and the respective residual terms that represent the oceanic contribution ($\frac{\partial BOXTA}{\partial t} - \frac{Q_{net}-Q_H}{\rho C_p H}$ and $\frac{\partial MLTA}{\partial t} - \frac{Q_{net}-Q_h}{\rho C_p h}$). When the correlation coefficients of the corresponding terms for both are high, we believe that although the control volumes calculated by the two are different (one changes over time at the bottom, while the other remains constant), there is no statistically significant difference in the dominant physical mechanisms corresponding to both.

3. Related to comment 2, the response last time mainly addressed from the perspective of vertical processes, however, I think authors tried to explain the warm water transportation from the view of horizontal advection. But authors did not show any advantages of the box model from the view of horizontal advection.

The primary advantage of the box model lies in its ability to effectively distinguish the horizontal and vertical processes. In this framework, horizontal processes are depicted as the aggregate of net heat flux at the lateral boundaries of the computational domain, deliberately excluding any terms that manifest as temperature differences times velocity.

We will explain the reason why the inclusion of temperature difference term is counterproductive to diagnostic analysis. The general formulation of the entrainment term $(\bar{T} - T_{-h})w$ incorporates a temperature difference $(\bar{T} - T_{-h})$, with w denoting the terms in forms of vertical velocity $(\frac{\partial h}{\partial t} + \mathbf{u}_{-h} \cdot \nabla h + w_{-h})$. Even in instances where temperature difference is substantial, it does not preclude the possibility that it is primarily influenced by \bar{T} . Such a scenario results in a circular reasoning, wherein the term \bar{T} is utilized to justify itself $(\frac{\partial \bar{T}}{\partial t})$, which is evidently illogical.

However, the box model $(\frac{\partial BOXTA}{\partial t} = \frac{Q_{net} - Q_H}{\rho c_p H} + \frac{\oint (\int_{-H}^0 \mathbf{u}_L T_L dz) dl}{\iiint_{-H}^0 dx dy dz} + \frac{\iint w T_B dx dy}{\iiint_{-H}^0 dx dy dz} + \text{mixing})$ addressed this problem. The temperature tendency is represented as the net heat flux through the six boundaries of the box, i.e., the net heat flux through the upper interface $(\frac{Q_{net} - Q_H}{\rho c_p H})$, the horizontal advection through four lateral interfaces $(\frac{\oint (\int_{-H}^0 \mathbf{u}_L T_L dz) dl}{\iiint_{-H}^0 dx dy dz})$, the vertical advection through lower interface $(\frac{\iint w T_B dx dy}{\iiint_{-H}^0 dx dy dz})$, and mixing. Each term is represented in the form of net heat flux $(\bar{T}\mathbf{v})$, rather than effective heat flux $((\bar{T} - T_{-h})\mathbf{v})$.

4. As other reviewers also pointed out the concern on the relationship between the shoaling mixed layer depth anomalies and deepened thermocline depth anomalies, I think authors should give the clear figures (spatial pattern and temporal evolution) to better clarify this issue.

The processes that dominate the mixed layer depth and the processes that dominate the main thermocline are different. The mixed layer depth is influenced by surface fluxes (heat, freshwater and momentum (surface wind stress induced turbulent mixing)) and fluxes at the bottom of the mixed layer, including stretching or squeezing of the water column and advection. Vertical excursions of the thermocline primarily reflect the effects of internal oceanic dynamic processes. It is therefore not unusual for changes in the mixed layer to differ from those in the main thermocline. Variations in sea level and vertical (or horizontal) movements of the main thermocline are dynamically related, with variations in sea level possible by movements of the gyre boundary.

Here, we analyzed the spatial pattern and temporal evolution of the mixed layer depth using an empirical orthogonal function (EOF) analysis (EOF1 is shown here in Figure R2-2).

Figure R2-2. The first EOF of the monthly mixed layer depth anomaly (with climatological monthly mean removed, from BRAN2020). The upper panel shows the EOF1 spatial pattern and the lower panel shows the corresponding mode-1 principal component time series.

The leading EOF (EOF1) of the mixed layer depth anomaly (MLDA) exhibits in-phase variations, accounting for 47% of the variance. During the Blob period (2013-2015, 2018-2019), this mode is characterized by a shallower mixed layer depth, with negative temporal values multiplied by the positive spatial pattern. This contrasts with the deepening of the main thermocline, indicating that the advection of the layer thickness does not predominantly influence MLDA changes. Furthermore, the spatial pattern of EOF1 closely resembles the net heat flux observed in 2013 (Figure R1-7 in the ‘Response to reviewers’ for the first round of review), leading us to hypothesize that the variations in MLD may be largely influenced by the net heat flux. However,

the factors driving MLD variations extend beyond the scope of this study. We intend to focus on this issue in the future.

5. In terms of the circulation pattern, authors attributed to the TNH, rather than NPO. I think authors should note that the variance of TNH, based on the EOF analysis, is only around 5%, which may not well explain the frequent occurrence of marine heatwaves over the past decade.

Thank you for your comment. Liang et al. (2017) stated “*In the present study, we applied the same REOF method of Barnston and Livezey (1987) to monthly 700-mb geopotential height (Z700) anomalies in the NH (20–90°N) to identify the TNH pattern. The TNH pattern was identified as the ninth REOF mode, explaining 4.96% of the total variance, and its corresponding principal component was used as the TNH index in our study.*” Hence, we agree and acknowledge that the TNH explains 4.96% of the total variance in the NH (20–90°N). However, in the Blob region, we believe it can be the driving force. The spatial pattern of the regression map of 500-mb geopotential height (Z500) anomalies onto the TNH index provides clear evidence of this. Compared to relationships with the PNA and NPO index, the location of the Blob region falls in line with the centers of action of TNH.

Moreover, the existence of TNH in the abstract as the main findings may not be good as they have been already reported by other studies.

We revised the sentence in the abstract to now read “The physical driver is suggested to be the Tropical/Northern Hemisphere (TNH) teleconnection pattern.” to “The physical driver is suggested to be the Tropical/Northern Hemisphere (TNH) teleconnection pattern and consistent with previous studies.”

[REDACTED]

Screenshot of Figure 2 in Liang, Y.-C., Yu, J.-Y., Saltzman, E. S. & Wang, F. Linking the tropical Northern hemisphere pattern to the Pacific warm blob and Atlantic cold blob. *J. Clim.* **30**, 9041–9057 (2017).

6. In the Fig. R1-3, I cannot see any obvious trend, although authors have processed the data.

The magnitude of this figure is the trend.

Additional comments:

1. Tracked-changes version: I cannot see any tracked changes. Please make sure that authors upload the correct file.

Sorry for the inconvenience, this time we take specific care.

2. Affiliation 7 did not belong to any author. Please check.

Now fixed, thank you.

3. Fig. 5e,f: Why did authors show the magenta lines here?

These lines are to highlight the negative peak of the latent heat flux.

4. Although authors added some statistical significance tests in the revision manuscript, there are other places that should have such test, for example, Figs. 2g, h.

Thank you for your comment. Figs. 2g and h are not the mapping of correlation coefficients. Rather, they simply show the phase difference between sea level and SST.

Reviewer #3 (Remarks to the Author):

Comments

Most of the points I previously mentioned have been appropriately revised, so I suggest acceptance once the minor comments are addressed.

Thank you very much for your thoughtful review. We have now revised our manuscript based on your comments and those of the other two reviewers. Our responses are provided below, with your comments in black font and our replies in blue font.

Minor comments

Line 91: In the phrase "geostrophy influences ~", geostrophy itself cannot induce vertical motion. However, vertical motion can occur when baroclinicity and the Ekman effect act together. This sentence may lead to misunderstanding, so I suggest revising it to explicitly include this explanation.

Thank you for your comment. Indeed, we agree that geostrophic balance alone cannot directly drive vertical motion in barotropic flows. However, in the stratified ocean, large-scale circulation follows potential vorticity conservation, which governs the relationship between layer thickness and the latitudinal displacement of water columns. Changes in the latitudinal position of a water column alter its thickness, thereby inducing vertical motion within the column.

Figure 2: Changing the colorbar for the Difference plots to a different color would make the distinction between (a)-(f) and (g)-(i) clearer.

Thank you for your comment. We have now revised this figure accordingly.

Reviewer #2 (Remarks to the Author):

Thanks for authors to carefully address my concerns. I am mostly clear about my concerns. But I still have some minor comments before the manuscript being potentially accepted.

Thank you very much for your thoughtful review. We have now revised our manuscript based on your comments. Our detailed responses are provided below, with your original comments in black font and our replies in blue font.

1. Figure R2-1 can be moved to supplementary information, but now I cannot find any supplementary information.

We have moved Figure R2-1 to a supplementary information document following your suggestion. Previously, there was no supplementary information document.

2. In the last paragraph of response to my second comment, it would be great if authors could add figures to show each term and their corresponding correlations.

In the last paragraph of Section 5a) under "Data and Methods", we now show the correlation coefficient between the box model and the relevant terms in the mixed layer heat budget.

In our revised manuscript, we now include the following text: *“The high correlation coefficients for the total temperature tendencies (+0.93, $p < 0.05$, between the red line in Fig. 3d and red shading in Fig. 3c), net heat flux terms (+0.84, $p < 0.05$, between the blue lines in Fig. 3d and Fig. 3c), and the respective residual terms that represent the oceanic contribution (+0.75, $p < 0.05$) suggest that the decomposition of the box model reflects the same physical processes as that of the classic mixed layer heat budget.”*

In Figures 3c and 3d, we present the total temperature tendency from the classic mixed layer heat budget (MLTA tendency, red line) and the box model (BoxTA tendency, red line), along with the net heat flux (blue line), respectively. We made comparisons of each term in Figure R1-1.

Figure R1-1. Comparisons of each term in the mixed layer heat budget (solid line) and box model (dashed line).

3. In the last paragraph of response to my third comment, I think authors could re-write it especially the last sentence and try to use the same marks (including consistent upper or lower cases with previous comment). Why is the first term on the right hand side derived from six boundaries of the box?

We rewrite the paragraph as follows and added it as the 3rd paragraph of Section 5a) under "Data and Methods" (approximately line 627-654):

The box model quantifies the total temperature tendency ($\frac{\partial BOXTA}{\partial t}$) through six interfaces of a predefined cubic control volume in the upper ocean. The model assumes the heat flux into the box is positive and is expressed as: $\frac{\partial BOXTA}{\partial t} = \frac{Q_{net} - Q_H}{\rho c_p H} + \frac{\oint (\int_{-H}^0 \mathbf{u}_L T_L dz) dl}{\iiint_{-H}^0 dx dy dz} + \frac{\iint w T_B dx dy}{\iiint_{-H}^0 dx dy dz} + mixing$, where H is the thickness of the box (in our case it is the total thickness of the upper 10 grids, with a value of about 61 m), \mathbf{u}_L is the horizontal velocity normal to the boundaries of the computational domain, by assuming inflow is positive, T_L is the seawater temperature at the boundaries. The accumulated horizontal advection through the four lateral interfaces (west, east, north, south) is calculated as:

$\oint \left(\int_{-H}^0 \mathbf{u}_L T_L dz \right) dl = \iint u_w T_w dydz + \iint v_s T_s dydz + \iint -u_e T_e dydz +$
 $\iint -v_n T_n dydz$, with u_w / v_s and u_e / v_n representing zonal (east–west) and meridional (north–south) velocities at the corresponding interfaces, T_w, T_e, T_s, T_n as temperatures. The vertical advection term is derived from wT_B (vertical velocity \times temperature at the bottom interface). Each term in the box model is represented as a net heat flux (velocity \times temperature, e.g., $\left(\frac{\oint \left(\int_{-H}^0 \mathbf{u}_L T_L dz \right) dl}{\iiint_{-H}^0 dx dy dz} \right)$ and $\frac{\iint w T_B dx dy}{\iiint_{-H}^0 dx dy dz}$) at the boundaries, rather than an effective heat flux (e.g., velocity \times temperature gradient, e.g., $-\bar{\mathbf{u}} \cdot \nabla \bar{T}$ and $-\left(\frac{\bar{T} - T_{-H}}{H} \right) \left(\frac{\partial H}{\partial t} + \mathbf{u}_{-H} \cdot \nabla H + w_{-H} \right)$).

4. I think the deepening of thermocline should be illustrated explicitly. Please show the figures.

By comparing zonally averaged temperatures (145°W–135°W) when the gyre boundary is at its northernmost (Figure 5a, revision) and southernmost (Figure 5b, revision) positions, we find: 1. When the boundary shifts north, the thermocline descends in the upper 200 m at 33°N and the upper 120 m at 44°N. 2. Since the maximum mixed layer depth is ~ 100 m, this downward shift of isotherms likely reflects thermocline variability.

Figure R1-2. Zonally averaged (145 °W-135 °W) temperature for the positive phase (red lines) and negative phase (blue lines). Color shading denotes the temperature difference between the two phases, with thick black lines denotes the 0-value. The thick blue and red lines denote isotherms of 6 °C.

As I was asked to take a look at the points raised by Reviewer #1 as well, I will also include those comments as below:

1. I suggest authors to give specific line numbers for their modifications, which is not clear now.

Sorry for the inconvenience. We added line numbers in this revision.

2. In terms of the response to #1 comment, authors did not prove the causality between the warming trend and displacement of the gyre boundary. Thus, “a major contributor ...” needs to be rephrased.

The causality can be established based on the result of the mixed layer heat budget, that is, the oceanic processes contribute positively to the mixed layer temperature tendency, while the net surface heat flux contribute negatively. Second, in our response to the first round of reviewer comments (in response to the first-round review, Figure R1-4), we confirmed a long-term weakening trend in the net surface heat flux term, indicating enhanced oceanic heat loss to the atmosphere. While this enhanced oceanic heat loss cannot explain the linear trend of SSTA increase, it is the consequence of this linear trend.

In the penultimate paragraph of the “Conclusions and implications”, we conducted a simple quantification. First, by analyzing the latitudinal shift of the 0.47 contour of sea level, we estimate a northward migration rate of the gyre boundary of 1.33°/decade. Second, based on the sea surface temperature gradient in this region—and assuming a characteristic SST gradient intensity of 2°C per 3°–4° latitude in the northeast Pacific (the ‘Blob’ region)—we calculated that the SSTA increase solely due to the northward shift of the circulation boundary should amount to be 0.67–0.89°C/decade. In contrast, the actual observed warming trend in the ‘Blob’ region is only 0.17°C/decade.

Two reasons may be responsible for the discrepancy between the estimated and observed warming trend. First, the sea level data span from 1993 to 2022. We notice there are negative anomalies of decadal variation in the earlier stage of the data, and positive anomalies in the recent decade. This means the actual linear trend is also affected by the decadal variation of the gyre boundary and may be overestimated. Second, the enhanced oceanic heat loss to the atmosphere mitigates the warming. Taken together, these two pieces of evidence suggest that the long-term warming trend is associated with the northward shift of the gyre boundary, while the enhanced atmospheric heat loss to the ocean largely offsets this warming trend. We added the explanation of the discrepancy into the penultimate paragraph of the “Conclusions and

implications” (From Line 527).

3. "For the multi-season persistence, shifts in the Sverdrup balance are favored by the low frequency component of the atmospheric forcing, with the ocean acting as a low-pass filter to the atmospheric forcing": By saying these, are ocean dynamics key to the multi-season persistence? I am not clear. Would it be possible to quantify the contribution to the SSTAs of the MHWs?

Yes, from the perspective of SSTA variation, ocean dynamics play a key role in the multi-season persistence. Specifically, there are three key factors contributing to this multi-season persistence:

First, the ocean’s Sverdrup response to the atmosphere dictates that only the low-frequency component of SLP (sea level pressure) can drive the meridional displacement of the circulation boundary. Second, the atmospheric physical driver itself also exhibits low-frequency variability—for example, the TNH (Tropical Northern Hemisphere pattern) is inherently an interannual oscillation of the atmospheric pressure field. Third, westward-propagating atmospheric Rossby waves may also have made an important contribution to the maintenance of the marine heatwave (Shi et al., 2025). We discuss this last point in more detail in the sixth paragraph of the revised Discussion section (lines 504-505).

4. "ocean advection is the only factor contributing to the emergence of the two warm events": How to reconcile with previous studies? I think this issue should be better elaborated in the manuscript.

The first reviewer commented that *"ocean advection is the only factor contributing to the emergence of the two warm events"*, whereas we argue that ocean advection is the primary (or dominant) factor (Refers to line 463-464 *"First, oceanic advection was found to be the principal driver of the formation of the 2013-15 and 2019-20 northeast Pacific marine heatwave events."*). To reconcile with the earlier research, we incorporated an analysis of the net surface heat flux (see Section: Anomalous northwesterly wind explains marine heatwave decay). Our final mechanistic analysis indicates that the interannual variation of the net surface heat flux is determined by different air-sea interaction conditions (Figure 7). We also cited the extratropical atmospheric teleconnections (Shi et al., 2024) as an important factor that may explain the multi-season persistence of the ‘Blob’ (Line 504-505).

Ref:

Shi, J. et al. Northeast Pacific warm blobs sustained via extratropical atmospheric teleconnections. *Nat. Commun.* **15**, 2832 (2024).

NCEP/NCAR Reanalysis
Sea Level Pressure (mb) Composite Anomaly 1991–2020 climo

Jan to Dec: 2014 to 2022 minus 1993 to 2002